# DOMAIN-AGNOSTIC SCALABLE AI SAFETY ENSURING FRAMEWORK

## ABSTRACT

AI safety has emerged as a critical priority as these systems are increasingly deployed in real-world applications. We propose **the first domain-agnostic AI safety ensuring framework** that achieves strong safety guarantees while preserving high performance, grounded in rigorous theoretical foundations. Our framework includes: (1) an optimization component with chance constraints, (2) a safety classification model, (3) internal test data, (4) conservative testing procedures, (5) $\zeta$-informative dataset quality measures, and (6) continuous approximate loss functions with gradient computation. Furthermore, to our knowledge, we mathematically establish **the first scaling law in AI safety research**, relating data quantity to safety-performance trade-offs. Experiments across *reinforcement learning, natural language generation*, and *production planning* validate our framework and demonstrate superior performance. Notably, in reinforcement learning, we achieve **3 collisions during 10M actions, compared with 1,000 - 3,000 for PPO-Lag baselines** at equivalent performance levels—a safety level unattainable by previous AI methods. We believe our framework opens a new foundation for safe AI deployment across safety-critical domains.

## 1 INTRODUCTION

As AI systems are increasingly deployed in safety-critical domains such as healthcare and transportation, ensuring their safety has become a fundamental requirement rather than an optional consideration. While recent works (Zou et al., 2023; Agnihotri et al., 2024; Liu et al., 2024a) mainly focus on specific domains, they have a fundamental limitation: it is hard to enforce uniform safety standards across different applications, especially for regulatory purposes, and they can struggle when immediately applied to new AI systems that lack specialized safety techniques. Overcoming these challenges, we propose a **domain-agnostic AI safety ensuring framework that achieves strong safety guarantees while preserving high performance, grounded in rigorous theoretical foundations.** To our knowledge, this is the first to propose a domain-agnostic AI safety framework and validate it across multiple domains.

In this paper, we define *safety* as satisfying all user-specified constraints with user-specified probability thresholds (*e.g.,* ensuring a language model produces harmful outputs in less than 1% of cases). Our key insight is to formulate AI safety as a constrained optimization problem by adding an **optimization component** (Section 2.2). Given any AI model, we take its output and generate the final action by optimizing the user-specified objective for high performance, while probabilistically satisfying safety constraints. This requires two main steps: (1) estimating the probability that (a group of) action candidates (*e.g.,* the AI model's output, or a fixed set) violate each safety constraint, and (2) solving the subsequent optimization problem to find safe, high-performance actions.

How can we handle safety constraints? The fundamental challenge is that safety cannot be determined solely from AI model outputs—it depends on the true *environment state*. For example, in reinforcement learning, action safety depends on the actual environment state rather than the agent's observations, which may contain sensor errors. We define environment state broadly: given this state, an action's safety and performance can be deterministically assessed.

Since perfect environment state prediction is impossible in most tasks, we formulate safety constraints as **chance constraints** (Section 3.2): keeping *constraint violation probabilities* below user-specified thresholds. To evaluate these chance constraints, our framework uses a **safety classification model**

(Section 3.3) that generates a constraint-related output. In our experiments, these outputs are identical to the environment states as the simplest choice; thus, the safety classification model directly estimates the environment state. This prediction is then processed through a procedure that calculates the *posterior probability* of the actual environment state given the predicted output state. These posterior probability estimates are subsequently used for the optimization problem described above.

It is obvious that we need ground-truth safety-labeled data to run our framework. We refer to this as **internal test data** (Section 3.4), which serves dual purposes: evaluating the chance constraints through posterior probability calculations and eventually training the safety classification model. However, using the same data for both training and evaluation creates statistical validity concerns, particularly the risk of overfitting. Our **conservative testing** procedure (Section 3.6) addresses this by deliberately overestimating safety risks by calculating the upper bound of posterior probability (thus, the upper bound of chance constraint) estimates. The degree of overestimation (*i.e.*, conservativeness) in these estimates depends on what we define $\zeta$**-informative** (Section 3.5): a measure of how well a dataset covers the target probability distribution. Notably, high data quality (= good coverage = low $\zeta$) allows our framework to be less conservative while preserving safety guarantees. If the upper bound of our chance constraints is still lower than user-specified probability thresholds, we can guarantee that the system is safe.

Since we propose an entirely new framework, a natural question arises: Is this framework trainable? This requires a loss function that is continuous with respect to the model outputs (rather than the final actions after optimization) to enable backpropagation. To address this requirement, we propose an **approximate loss function** that maintains continuity with respect to model outputs, along with **computation for tailored (virtual) gradients** for training the safety classification model (and optionally the AI model as well).

Furthermore, we mathematically establish and prove a **scaling law** between the quantity of internal test data and the safety-performance trade-off—the fundamental trade-off where achieving stronger safety guarantees typically requires accepting lower performance. For example, when fixing the performance level, increasing the quantity of internal test data enables safer model behavior. To our knowledge, this scaling law represents the first such theoretical relationship in AI safety research. It demonstrates that our framework's effectiveness scales predictably with the quantity of data, indicating continued improvement potential beyond our experimental demonstrations.

Experiments across *reinforcement learning, natural language generation*, and *production planning* validate our framework across diverse domains, demonstrating superior performance and empirically confirming our scaling law. Notably, in reinforcement learning, we achieve **3 collisions during 10M actions, compared with 1,000 - 3,000 for PPO-Lag (Ray et al., 2019) baselines** at equivalent performance levels. Building on these strong experimental results and rigorous theoretical foundations, we believe our domain-agnostic, safety ensuring framework provides a foundation for deploying AI systems in safety-critical applications and fosters the development of domain-agnostic AI methods.

## 2 PRELIMINARY

### 2.1 DOMAIN-AGNOSTIC AI SAFETY FRAMEWORK

In this paper, we propose a domain-agnostic framework for AI safety. We use the term *domain-agnostic* to refer to a framework's ability to work with any arbitrary AI model from any domain. To our knowledge, this is the first work to propose a domain-agnostic AI safety framework and validate it across multiple domains. We define a concept of AI *safety* that can be applied across various domains—satisfying all user-specified constraints with user-specified probability thresholds. Thus, we focus solely on scenarios where safety is well-defined. In this context, safety can be generally defined in the form of constraints. Note that any form of action safety can be converted to a constraint. For example, for an action $\mathbf{u} \in \mathcal{U} \subset \mathbb{R}^{n_{u1}} \times \mathbb{Z}^{n_{u2}}$[1], we can use the constraint $c(\mathbf{u}) \geq 0$ with defining $c(\mathbf{u}) = -1$ when $\mathbf{u}$ is unsafe and $c(\mathbf{u}) = 1$ otherwise.

### 2.2 PROBLEM SETUP & NOTATIONS

We begin by explaining the term *environment state*, which we denote as $\mathbf{s} \in \mathcal{S} \subset \mathbb{R}^{n_{s1}} \times \mathbb{Z}^{n_{s2}}$. We define the environment state as: given this state, the safety and performance of an AI model's final

---

[1]$\mathbb{R}^{n_\star}$ is the space of $n_\star$-dimensional real vectors, and $\mathbb{Z}^{n_\star}$ is the space of $n_\star$-dimensional integer vectors, where any dimension $n_\star$ may be zero.

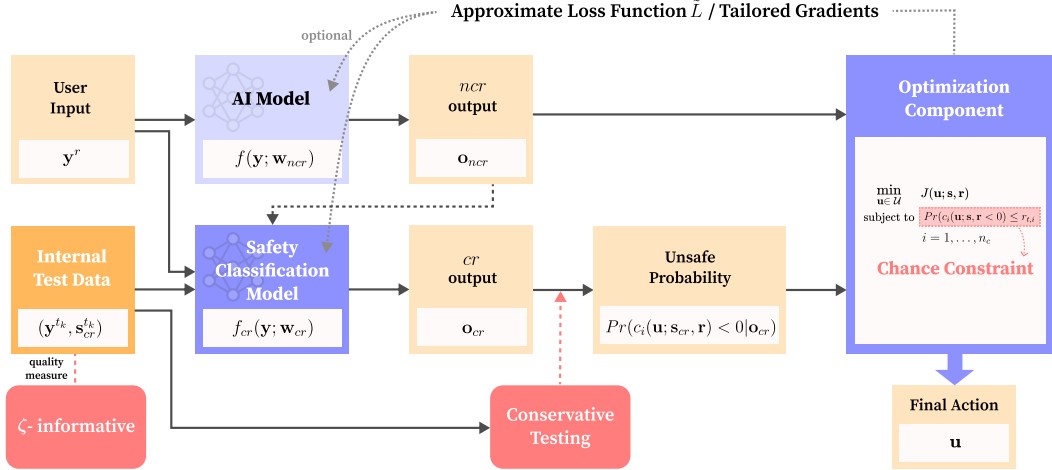

Figure 1: **Our Framework.** We propose a framework that chooses provably safe actions based on any AI model's output, while maintaining high performance.

action can be deterministically assessed. For example, in reinforcement learning, this includes the information of the actual environment in which the agent is placed. Note that this is typically different from the observations that the agent makes, since agent observations often contain errors (*e.g.,* LiDAR sensors in autonomous driving). In natural language generation, the environment state corresponds to the ground-truth safeness of responses by the generator for a given prompt, which we cannot know with certainty. For most tasks, obtaining the actual environment state for all possible cases is fundamentally impossible.

We denote the input as $\mathbf{y} \in \mathcal{Y} \subset \mathbb{R}^{n_y}$. Here, we think of $\mathbf{y}$ as a *measurement* of the environment state $\mathbf{s}$ (*i.e.,* measurement $\mathbf{y}$ follows a sampling-like function $Samp(\mathbf{s})$). Given an AI model $f$ with trained weights $\mathbf{w} \in \mathbb{R}^{n_w}$ and input $\mathbf{y}$, the AI model produces a continuous vector $f(\mathbf{y}; \mathbf{w}) \in \mathbb{R}^{n_f}$ as output. This continuous vector is processed, and then the system outputs the final action $\mathbf{u} \in \mathcal{U} \subset \mathbb{R}^{n_{u1}} \times \mathbb{Z}^{n_{u2}}$. Following the definition of safety in Section 2.1, when user-specified constraints $\mathbf{c} : \mathcal{U} \to \mathbb{R}^{n_c}$ and user-specified parameters (including probability thresholds) $\mathbf{r} \in \mathbb{R}^{n_r}$ are given, we want the system's action $\mathbf{u}$ to satisfy $c_i(\mathbf{u}; \mathbf{s}, \mathbf{r}) \geq 0$ for $i = 1, \ldots, n_c$ under the probabilistic environment state $\mathbf{s}$. This can be formulated as:

$$\min_{\mathbf{u} \in \mathcal{U}} \quad J(\mathbf{u}; \mathbf{s}, \mathbf{r}) \tag{1a}$$

$$\text{subject to} \quad c_i(\mathbf{u}; \mathbf{s}, \mathbf{r}) \geq 0, \quad i = 1, \ldots, n_c \tag{1b}$$

where $J : \mathcal{U} \to \mathbb{R}$ is the user-specified objective minimized to achieve high performance.

## 3 OUR FRAMEWORK: DEALING WITH SAFETY CONSTRAINTS

How can we deal with safety constraints? From Equation 1, both the objective $J$ and constraints $c_i$ are functions of the environment state $\mathbf{s}$. However, as noted in Section 2.2, obtaining the actual environment state is fundamentally impossible, making the calculation of $J(\mathbf{u}; \mathbf{s}, \mathbf{r})$ and $c_i(\mathbf{u}; \mathbf{s}, \mathbf{r})$ challenging. To address this challenge, we utilize a proxy objective and constraints in an optimization component (Section 3.1), including chance constraints (Section 3.2). To evaluate these chance constraints, we employ a safety classification model (Section 3.3) along with internal test data (Section 3.4) and conservative testing (Section 3.6). $\zeta$-informative (Section 3.5) is a concept we introduce to measure the quality of a dataset, used for conservative testing. Comprehensive details are provided in the Appendix (Sections A, B, C).

### 3.1 OPTIMIZATION COMPONENT

Our key insight is to formulate AI safety as a constrained optimization problem. Generalizing the predict-and-optimize framework (Donti et al., 2017; Amos & Kolter, 2017; Kotary et al., 2021), we add an optimization component that is guaranteed to select actions that satisfy safety constraints while maintaining high performance, under the assumption of the existence of a safety-guaranteed default action. Thus, given any AI model, we utilize its output and select the final action $\mathbf{u}$ while satisfying user-specified constraints. We will formalize this optimization component later as Equation 6.

## 3.2 CHANCE CONSTRAINTS

We formulate safety constraints as chance constraints by considering the *probability* of violating each constraint. Thus, the safety constraints (Equation 1b) can be written as:

$$Pr(c_i(\mathbf{u}; \mathbf{s}, \mathbf{r}) < 0) \leq r_{t,i}, \qquad i = 1, \ldots, n_c \tag{2}$$

where $Pr$ indicates probability, and $r_{t,i} \in \mathbb{R}$ is the user-specified probability threshold. Note that chance constraints are optional; for example, they are not required for deterministic constraints.

## 3.3 SAFETY CLASSIFICATION MODEL

We use the subscript $cr$ for 'constraint-related'. Then, we divide the environment state $\mathbf{s}$ into two parts: $\mathbf{s}_{cr} \in \mathcal{S}_{cr}$ is the necessary part to determine the constraints $\mathbf{c}$, and $\mathbf{s}_{ncr} \in \mathcal{S}_{ncr}$ is the remainder.[2] Similarly, since the AI model generates output $\mathbf{o}$ which is not related to the safety constraints, we denote this as $\mathbf{o}_{ncr} \in \mathcal{O}_{ncr}$.

Motivated by the need to estimate $\mathbf{s}_{cr}$, we introduce a *safety classification model* that generates $\mathbf{o}_{cr} \in \mathcal{O}_{cr}$—the prediction that is constraint-related[3]. Since we cannot directly obtain $\mathbf{s}_{cr}$ and our safety constraints are now formulated as probabilities (Equation 2), we aim to calculate the *posterior probability* $p(\mathbf{s}_{cr}|\mathbf{o}_{cr})$. Throughout this paper, for simplicity, we assume $\mathbf{s}_{cr}$ and $\mathbf{o}_{cr}$ take discrete values (*e.g.,* in our natural language generation experiments, $\mathbf{s}_{cr}$ would be either "harmful"= 1 or "harmless"= 0). Let $\mathcal{S}_{cr} = \{\bar{\mathbf{s}}_1, \ldots, \bar{\mathbf{s}}_{n_{scr}}\}$ and $\mathcal{O}_{cr} = \{\bar{\mathbf{o}}_1, \ldots, \bar{\mathbf{o}}_{n_{ocr}}\}$ denote the sets of possible discrete values for $\mathbf{s}_{cr}$ and $\mathbf{o}_{cr}$, respectively. While $\mathcal{O}_{cr}$ does not need to be identical to $\mathcal{S}_{cr}$, for simplicity, we use $\mathcal{O}_{cr} = \mathcal{S}_{cr}$ in our experiments.[4] Thus, our safety classification model directly estimates the environment state.

In practice, the safety classification model offers considerable flexibility in implementation; in our experiments, we use the same structure or a reduced variant of the given AI model $f$. We denote the weights of the AI model as $\mathbf{w}_{ncr}$ and weights of the safety classification model as $\mathbf{w}_{cr}$, giving us $f(\mathbf{y}; \mathbf{w}_{ncr})$ and $f_{cr}(\mathbf{y}; \mathbf{w}_{cr})$, respectively.

## 3.4 INTERNAL TEST DATA

How can we calculate the aforementioned posterior probability $p(\mathbf{s}_{cr} = \bar{\mathbf{s}}_i|\mathbf{o}_{cr} = \bar{\mathbf{o}}_j)$? We introduce *internal test data*, ground-truth safety-labeled data consisting of samples for which we know the constraint-related environment state $\mathbf{s}$. Let us denote the internal test data as measurements $\mathbf{y}^{t_1}, \ldots, \mathbf{y}^{t_{n_t}}$, each associated with environment state labels $\mathbf{s}_{cr}^{t_1}, \ldots, \mathbf{s}_{cr}^{t_{n_t}}$, respectively (we only use the constraint-related part of $\mathbf{s}$ for internal test data, so we write it as $\mathbf{s}_{cr}$). To compute the posterior probability, we add internal test data to the input:

$$\mathbf{y} = (\mathbf{y}^r, \mathbf{y}^{t_1}, \cdots, \mathbf{y}^{t_{n_t}}) \tag{3}$$

where $\mathbf{y}^r$ is the user-given input (what we previously referred to as $\mathbf{y}$). After processing this input $\mathbf{y}$ through our safety classification model, we can count the total number of internal test cases for each environment state as $N_{\mathbf{s}_{cr}=\bar{\mathbf{s}}_i} = \sum_{k=1}^{n_t} \mathbf{1}(\mathbf{s}_{cr}^{t_k}, \bar{\mathbf{s}}_i)$ and for each output and environment state pair as $N_{\mathbf{s}_{cr}=\bar{\mathbf{s}}_i, \mathbf{o}_{cr}=\bar{\mathbf{o}}_j} = \sum_{k=1}^{n_t} \mathbf{1}(\mathbf{s}_{cr}^{t_k}, \bar{\mathbf{s}}_i) \cdot \mathbf{1}(\mathbf{o}_{cr}^{t_k}, \bar{\mathbf{o}}_j)$.[5] Then, we can estimate the following probability:

$$p(\mathbf{o}_{cr} = \bar{\mathbf{o}}_j|\mathbf{s}_{cr} = \bar{\mathbf{s}}_i) \simeq \frac{N_{\mathbf{s}_{cr}=\bar{\mathbf{s}}_i, \mathbf{o}_{cr}=\bar{\mathbf{o}}_j}}{N_{\mathbf{s}_{cr}=\bar{\mathbf{s}}_i}} \tag{4}$$

Assuming that the user specifies the prior knowledge $p(\mathbf{s}_{cr} = \bar{\mathbf{s}}_i)$, we can calculate the posterior probability using Bayes' rule:

$$p(\mathbf{s}_{cr} = \bar{\mathbf{s}}_i|\mathbf{o}_{cr} = \bar{\mathbf{o}}_j) = \frac{p(\mathbf{o}_{cr} = \bar{\mathbf{o}}_j|\mathbf{s}_{cr} = \bar{\mathbf{s}}_i) \cdot p(\mathbf{s}_{cr} = \bar{\mathbf{s}}_i)}{\sum_{k=1}^{n_{scr}} p(\mathbf{o}_{cr} = \bar{\mathbf{o}}_j|\mathbf{s}_{cr} = \bar{\mathbf{s}}_k) \cdot p(\mathbf{s}_{cr} = \bar{\mathbf{s}}_k)} \tag{5}$$

Since our goal is to replace the non-obtainable environment states $\mathbf{s}$ with our estimates $\mathbf{o}$, we show that Equation 1 can be converted into:

$$\min_{\mathbf{u} \in \mathcal{U}} \quad \bar{J}(\mathbf{u}; \mathbf{o}, \mathbf{r}) \tag{6a}$$

$$\text{subject to} \quad \bar{c}_i(\mathbf{u}; \mathbf{o}, \mathbf{r}) \geq 0, \ \ i = 1, \ldots, n_{\bar{c}} \tag{6b}$$

---

[2]$ncr$ stands for non-constraint-related.

[3]Thus, $(\mathbf{o}_{cr}, \mathbf{o}_{ncr}) = \mathbf{o} \in \mathcal{O} \subset \mathbb{R}^{n_{o1}} \times \mathbb{Z}^{n_{o2}}$

[4]Note that our mathematical foundation throughout the paper is not constrained by this choice.

[5]$\mathbf{1}(x_1, x_2)$ is the indicator function that equals 1 when $x_1 = x_2$ and 0 otherwise.

where $\bar{J}$ is $J$ written in terms of $\mathbf{o}$ [6], and $\bar{c}_i$ is $c_i$ written in terms of $\mathbf{o}$ as:

$$\bar{c}_i(\mathbf{u}; \mathbf{o}, \mathbf{r}) := \max_{\substack{q_1,\ldots,q_{N_e} \in \{0,1\} \\ \sum_k p(\mathbf{s}_{cr} = \bar{\mathbf{s}}_k | \mathbf{o}_{cr} = \mathbf{o}_{cr}^r) \cdot q_k \leq r_{t,i}}} \min_k \left( c_i(\mathbf{u}; \bar{\mathbf{s}}_k, \mathbf{r}) + M \cdot q_k \right) \tag{7}$$

Here, $M$ is a sufficiently big constant, following the big-M method (Cococcioni & Fiaschi, 2021). All proofs and derivation details are provided in Section B.2.

During inference, internal test data enables posterior probability calculations while serving as final validation that ensures the trained safety classification model remains reliable. This validation addresses reliability concerns inherent in neural networks, with minimal computational overhead since the inference of the model with respect to the internal test data is performed only once.

### 3.5 $\zeta$-INFORMATIVE

The process of calculating posterior probabilities, described in Section 3.4, utilizes internal test data to evaluate the safety classification model, and this same data will eventually be used to train the framework (Section 4). This implies using the same data for both training and evaluation, creating statistical validity concerns, particularly the risk of overfitting. Before introducing our solution to address this challenge (*conservative testing*: Section 3.6), we will first present a concept $\zeta$-*informative*, which measures the quality of a dataset.

This concept measures how well a dataset covers the entire data space with respect to the target probability distribution. Consider the entire data space, where each data sample (total $n_s$) of a dataset represents a point within this space. We can conceptualize $\zeta$-*balls* centered at each dataset point with radius $\zeta$. We define a dataset as $\zeta$-informative if, for every number $k = 1, \ldots, n_s$, an arbitrary selection of $k$ $\zeta$-balls covers a proportion of the entire data space that is at least $k/n_s$. If the dataset points do not fully cover the probability distribution or are insufficiently dense, a high $\zeta$ value would be required to satisfy this condition. Therefore, a dataset with a small $\zeta$ value (when the dataset is $\zeta$-informative) contains data samples that densely and comprehensively cover the entire probability distribution, meaning the dataset follows the target probability distribution well.

We also prove $Pr(\lim_{|D| \to \infty} \inf(\zeta | D : \zeta - \text{informative}) = 0) = 1$: under mild conditions, sampling sufficient data points from the probability distribution enables us to achieve a sufficiently high quality. The mathematical definition of $\zeta$-informative and all proofs are provided in Section C.1.

### 3.6 CONSERVATIVE TESTING

We introduce *conservative testing* to address the statistical validity concerns arising from using the same data for both training and evaluation. Our approach deliberately overestimates safety risks by adding a penalty term $\xi$ to the intermediate results of the safety classification model (*e.g.,* logit values before the argmax function), making the safety constraints more conservative.

We assume and leverage the Lipschitz continuity property of AI models, which ensures that similar inputs produce similar outputs with bounded differences. Building on the $\zeta$-balls from Section 3.5, we define $\xi$-*balls* $\mathcal{B}(\phi_0, \xi)$, centered at $\phi_0 := f(\mathbf{y}^{t_k}; \mathbf{w}_{cr})$ in the output (logit) space, which includes the (image of) $\zeta$-balls that are passed through the safety classification model. We define two indicator functions $\mathbf{1}^{+\xi}$ and $\mathbf{1}^{-\xi}$ that capture the classification status within these $\xi$-balls: $\mathbf{1}^{+\xi}(\phi_0, \bar{\mathbf{o}}_j)$ equals 1 if any point in the $\xi$-ball $\mathcal{B}(\phi_0, \xi)$ would be classified as $\bar{\mathbf{o}}_j$, while $\mathbf{1}^{-\xi}(\phi_0, \bar{\mathbf{o}}_j)$ equals 1 only if all points in the $\xi$-ball $\mathcal{B}(\phi_0, \xi)$ would be classified as $\bar{\mathbf{o}}_j$.[7] Using these indicator functions, we define $\xi$-versions of the likelihood (Equation 4) by:

$$N_{\mathbf{s}_{cr}=\bar{\mathbf{s}}_i, \mathbf{o}_{cr}=\bar{\mathbf{o}}_j}^{\pm\xi} := \sum_{k=1}^{n_t} \mathbf{1}(\mathbf{s}_{cr}^{t_k}, \bar{\mathbf{s}}_i) \cdot \mathbf{1}^{\pm\xi}(\mathbf{o}_{cr}^{t_k}, \bar{\mathbf{o}}_j), \quad p^{\pm\xi}(\mathbf{o}_{cr} = \bar{\mathbf{o}}_j | \mathbf{s}_{cr} = \bar{\mathbf{s}}_i) := \frac{N_{\mathbf{s}_{cr}=\bar{\mathbf{s}}_i, \mathbf{o}_{cr}=\bar{\mathbf{o}}_j}^{\pm\xi}}{N_{\mathbf{s}_{cr}=\bar{\mathbf{s}}_i}} \tag{8}$$

We prove that under mild conditions, the true likelihood is bounded as:

$$p^{-\xi}(\mathbf{o}_{cr} = \bar{\mathbf{o}}_j | \mathbf{s}_{cr} = \bar{\mathbf{s}}_i) \leq p(\mathbf{o}_{cr} = \bar{\mathbf{o}}_j | \mathbf{s}_{cr} = \bar{\mathbf{s}}_i) \leq p^{+\xi}(\mathbf{o}_{cr} = \bar{\mathbf{o}}_j | \mathbf{s}_{cr} = \bar{\mathbf{s}}_i) \tag{9}$$

---

[6]For example, $\bar{J}(\mathbf{u}; \mathbf{o}, \mathbf{r})$ can be set as $\mathbb{E}_{\mathbf{s}}[J(\mathbf{u}; \mathbf{s}, \mathbf{r})|\mathbf{o}]$

[7]Mathematically, where $\mathrm{argmax}(\phi)$ is the post-processing operator for the safety classification model, $\mathbf{1}^{+\xi}(\phi_0, \bar{\mathbf{o}}_j) := \max_{\phi \in \mathcal{B}(\phi_0, \xi)} \mathbf{1}(\mathrm{argmax}(\phi), \bar{\mathbf{o}}_j)$ and $\mathbf{1}^{-\xi}(\phi_0, \bar{\mathbf{o}}_j) := \min_{\phi \in \mathcal{B}(\phi_0, \xi)} \mathbf{1}(\mathrm{argmax}(\phi), \bar{\mathbf{o}}_j)$.

This allows us to calculate an *upper bound of the posterior probability* (Equation 5):

$$p^\xi(\mathbf{s}_{cr} = \bar{\mathbf{s}}_i | \mathbf{o}_{cr} = \bar{\mathbf{o}}_j) := \frac{p^{+\xi}(\mathbf{o}_{cr} = \bar{\mathbf{o}}_j | \mathbf{s}_{cr} = \bar{\mathbf{s}}_i) \cdot p(\mathbf{s}_{cr} = \bar{\mathbf{s}}_i)}{\sum_{k=1}^{n_{scr}} p^{-\xi}(\mathbf{o}_{cr} = \bar{\mathbf{o}}_j | \mathbf{s}_{cr} = \bar{\mathbf{s}}_k) \cdot p(\mathbf{s}_{cr} = \bar{\mathbf{s}}_k)} \tag{10}$$

We define the conservative constraints $\bar{c}_i^\xi(\mathbf{u}; \mathbf{o}, \mathbf{r})$ by replacing $p$ with $p^\xi$ in Equation 7. Crucially, smaller $\zeta$-balls (= higher dataset quality) lead to smaller $\xi$-balls, enabling less conservative safety constraints. Based on these conservative constraints, we establish and prove the following guarantees under mild conditions: (1) every feasible solution under these conservative constraints also satisfies the actual constraints, and (2) the actual loss is upper-bounded by the loss we achieve when optimizing under conservative constraints. The first property ensures safety, and the second property directly addresses concerns about overfitting. Unlike traditional overfitting, where training performance improves while validation performance degrades, our approach provides an upper bound guarantee: significant improvements in training loss under conservative constraints guarantee improvements in actual loss, thus addressing overfitting. All conditions and proofs are provided in Section C.2.

## 4 TRAINING & RUNNING THE FRAMEWORK

This section details the training and deployment of our framework. The main challenge lies in constructing a continuous loss function suitable for backpropagation. The user-given loss function $L(\mathbf{u}, \mathbf{o}; \mathbf{s}, \mathbf{r})$ depends on the action $\mathbf{u}$, where the process of selecting the optimal $\mathbf{u}$ can be discontinuous with respect to model outputs $\mathbf{o}$, and multiple optimal $\mathbf{u}$ may yield different loss values. These factors make the overall loss likely non-differentiable with respect to model weights. We address this problem by developing a continuous loss approximation (Section 4.1) and corresponding gradient calculation methods (Section 4.2). Additionally, for practical deployment, we introduce a bias correction technique that enables safety threshold adjustments without requiring retraining (Section 4.3).

### 4.1 APPROXIMATE LOSS FUNCTION

We extend prior work (Vlastelica et al., 2020), which presents an approximate loss function for unconstrained problems with linear objective functions. Our contribution extends this approach to *general optimization problems* with *continuous objectives and constraints*.

Our approximation introduces two key parameters: $\boldsymbol{\beta} \in \mathbb{R}^{n_c}$ to merge the constraints into the objective function, and $\lambda \in \mathbb{R}$ to ensure that our approximate loss function $\tilde{L}$ converges to the true loss $L$. This yields:

$$\tilde{L}(\mathbf{o}; \mathbf{s}, \mathbf{r}, \boldsymbol{\beta}, \lambda) = \frac{1}{\lambda} \left( \min_{\mathbf{u} \in \mathcal{U}} \left( \lambda L(\mathbf{u}, \mathbf{o}; \mathbf{s}, \mathbf{r}) + \bar{J}(\mathbf{u}; \mathbf{o}, \mathbf{r}) - \boldsymbol{\beta}^\top \min(\bar{\mathbf{c}}(\mathbf{u}; \mathbf{o}, \mathbf{r}), \mathbf{0}) \right) \right.$$
$$\left. - \min_{\mathbf{u} \in \mathcal{U}} \left( \bar{J}(\mathbf{u}; \mathbf{o}, \mathbf{r}) - \boldsymbol{\beta}^\top \min(\bar{\mathbf{c}}(\mathbf{u}; \mathbf{o}, \mathbf{r}), \mathbf{0}) \right) \right) \tag{11}$$

We prove that under mild conditions, when $\lambda$ is sufficiently small and $\boldsymbol{\beta}$ is sufficiently large, the approximate loss function $\tilde{L}$, which is continuous with respect to $\mathbf{o}$, converges to the true loss function $L^*(\mathbf{o}; \mathbf{s}, \mathbf{r}) := L(\mathbf{u}^*, \mathbf{o}; \mathbf{s}, \mathbf{r})$, where $\mathbf{u}^*$ is the optimal solution of the optimization problem (Equation 6).[8] All conditions and proofs are provided in Section D.

### 4.2 GRADIENT COMPUTATION

We compute the (virtual) gradients of the approximate loss $\tilde{L}$ and backpropagate them to train the safety classification model and, optionally, the AI model. Given the input formulation in Equation 3, we calculate gradients with respect to both the input $\mathbf{y}^r$ and internal test data $\mathbf{y}^t$, then propagate these through the model outputs $\mathbf{o}$ to the model parameters $\mathbf{w}_{cr}$ and $\mathbf{w}_{ncr}$. Gradient computation is especially challenging when outputs are discrete or when $\tilde{L}$ is non-differentiable. We show in Section E that under mild conditions, (virtual) gradients with respect to both $\mathbf{y}^r$ and $\mathbf{y}^t$ can be calculated, despite these challenges.

---

[8]When multiple optimal solutions exist, we select $\mathbf{u}^*$ as the one that minimizes $L(\mathbf{u}^*, \mathbf{o}; \mathbf{s}, \mathbf{r})$.

### 4.3 Bias Correction when Running

Our framework enables threshold adjustments during deployment without requiring retraining, thanks to the bias correction technique. We modify the safety classification model by adding a constant bias vector $\mathbf{v}$ to the final layer: $f'_{cr}(\mathbf{y}; \mathbf{w}_{cr}) = f_{cr}(\mathbf{y}; \mathbf{w}_{cr}) + \mathbf{v}$. This produces adjusted outputs $\mathbf{o}'_{cr}$ while preserving all theoretical properties of our framework. At deployment, we first run inference on internal test data, then compute the bias vector $\mathbf{v}$ that makes the adjusted posterior upper bound $p^\xi(\mathbf{s}_{cr} = \bar{\mathbf{s}}_i | \mathbf{o}'_{cr} = \bar{\mathbf{o}}_j)$ match the desired threshold values. More details could be found in Section G.

## 5 Scaling Law

We mathematically establish a scaling law that characterizes the relationship between the quantity of internal test data and the trade-off between safety and performance. This trade-off represents the fundamental tension, where achieving stronger safety guarantees typically requires accepting lower performance. For example, when the performance level is fixed, increasing the quantity of internal test data enables safer model behavior. Specifically, our scaling law determines the number of internal test data points required to bound both Type I errors (misclassifying unsafe actions as safe) and Type II errors (misclassifying safe actions as unsafe). Under mild conditions, Theorem 3 formalizes this relationship as:

$$N_{reqit} \leq A\alpha^{-2n_y} \tag{12}$$

where $\alpha$ is the upper bound of both Type I and Type II error, $A$ is a constant, $n_y$ is the dimension of measurement space, and $N_{reqit}$ is the expected number of required internal test data. This theorem establishes an inverse power-law relationship between the error bound and the number of internal test data points required. Our scaling law demonstrates that the effectiveness of our framework scales predictably with the quantity of data, indicating continued improvement potential beyond our experimental demonstrations. The formal conditions and proof are provided in Section F.

## 6 Experiments

We validate our framework in three diverse domains. First, we demonstrate effectiveness in *reinforcement learning* using SafetyGym (OpenAI, 2019b), OpenAI's RL safety benchmark (Section 6.1). With abundant internal test data available through simulation, we achieve superior performance that confirms our scaling law. We then demonstrate our applicability and superior performance on *natural language generation* through simple experiments (Section 6.2). Finally, we experiment on *production planning*, a major industrial problem, showcasing our framework's general applicability across various domains and the ability to handle very complex optimization problems (Section 6.3). For each experiment, we present scatter plots that illustrate the performance-safety trade-off. Additional details and computational analysis are provided in the Appendix (Sections H, I, J, K).

### 6.1 Reinforcement Learning (SafetyGym)

**Problem Setup.** We use the 'Safexp-PointGoal1-v0' environment from SafetyGym (OpenAI, 2019b). The agent must navigate to a designated goal location while avoiding hazardous regions. The measurement $\mathbf{y}$ consists of simulated LiDAR and IMU outputs, and the action $\mathbf{u}$ comprises agent acceleration (range $[-1, 1]$) and angular velocity (range $[-1, 1]$). We define an action as *unsafe* if it leads the agent to enter a hazard region (= collision) within 60 subsequent actions, yielding $\mathcal{S}_{cr} = \mathcal{O}_{cr} = \{0, 1\}$, where 0 represents "safe" and 1 represents "unsafe".

**Baselines.** We compare against three baselines: PPO (Schulman et al., 2017), PPO-Lag (Ray et al., 2019), and PPO-Barrier (Yang et al., 2023). PPO serves as an unconstrained baseline, trained solely for goal achievement without safety considerations. PPO-Lag represents the standard approach for constrained RL with safety requirements. PPO-Barrier is one of the current state-of-the-art methods for SafetyGym environments.

**Implementation.** We integrate both PPO and PPO-Lag into our framework. These methods output mean and standard deviation parameters, so $\mathbf{o}_{ncr} = (\mu, \sigma)$, modeling the action distribution as $\mathcal{N}(\mu, \sigma^2)$. We effectively discretize this continuous action space, remove unsafe candidates through our framework, and sample the final action from the remaining safe options. To generate internal test data, we pre-train PPO and PPO-Lag agents for 10,000 epochs each (one epoch equals a scenario consisting of 1,000 actions), and collect internal test data through simulations, obtaining 5M unsafe

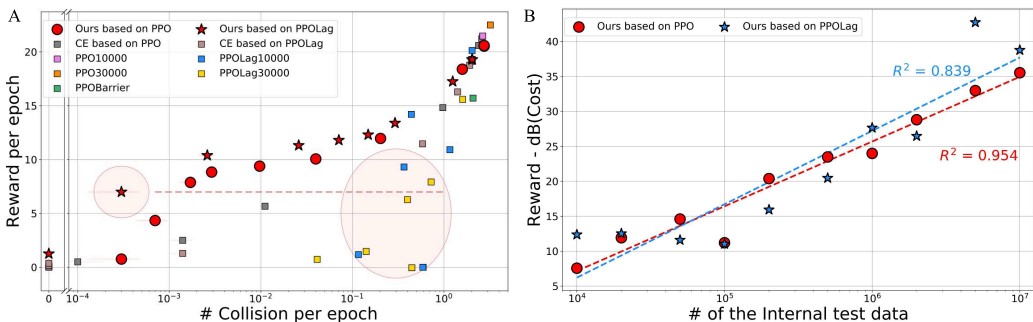

Figure 2: **Reinforcement learning.** (A) Our framework achieves dramatically low collision rates with competitive performance. (B) Our framework demonstrates clear scaling properties.

and 5M safe data points each. We define a default action $\hat{\mathbf{d}}$ for cases when all action candidates are classified as unsafe[9]. The optimization objective $\bar{J}$ is set as $\bar{J}(\hat{\mathbf{d}}) > 0$ and $\bar{J}(\mathbf{u}) = 0$ for other actions, encouraging the agent to avoid default action usage and thus improve performance. We train our framework starting from the $10,000$-epoch pre-trained PPO and PPO-Lag models as initial AI models, which are fine-tuned jointly with the safety classification model using a safety probability threshold of $10^{-4}$. For the safety classification model, we use a variant of the AI model architecture that is around one-third the size.

**Results.** Figure 2-A illustrates the safety-performance trade-off. Each point represents the results of simulating trained agents for 10,000 epochs[10]. Multiple points are shown for methods that handle different safety probability thresholds (our framework and its ablations) or cost limit levels (PPO-Lag). Lower probability thresholds yield fewer collisions but reduced performance, demonstrating the fundamental safety-performance trade-off. Our method efficiently explores different thresholds ($10^{-5} - 1.0$) using the bias correction technique from Section 4.3. Compared baselines include PPO and PPO-Lag trained for 10,000 or 30,000 epochs (to ensure fair comparison in terms of data exposure), and PPO-Barrier trained for 10,000 epochs.[11] Using extremely low thresholds, our PPO-Lag-based framework (red-star points) achieves **only 3 collisions during 10,000 epochs (10M actions)**. To our knowledge, this represents a **safety level unattainable by previous AI methods**, opening possibilities for safety-critical applications where even rare failures can be catastrophic. Notably, at equivalent reward (performance) levels, standard PPO-Lag (yellow points) experiences 1,000 - 3,000 collisions. Figure 2-B validates our scaling law.

**Ablation Studies.** We conduct ablation studies (silver, gold points) by disabling AI model fine-tuning and conservative testing. The safety classification model is trained using standard Cross-Entropy loss instead of our approximate loss from Section 4.1. These ablations essentially reduce to *rejection sampling* methods (von Neumann, 1951; Srinivasan et al., 2020). Our complete framework outperforms these ablations, demonstrating the importance of components and the potential for superior performance compared to rejection sampling-based methods.

## 6.2 NATURAL LANGUAGE GENERATION

**Problem Setup and Implementation.** We generate harmless responses $\mathbf{u}$ given input prompts $\mathbf{y}$. The AI model generates 16 candidate responses, where the environment state indicates the harmlessness of each, yielding $\mathcal{S}_{cr} = \mathcal{O}_{cr} = \{0, 1\}^{16}$. The AI model is OPT-1.3B (Zhang et al., 2022) fine-tuned with PPO-Lag on the SafeRLHF dataset (Dai et al., 2024). Each generated response is concatenated with the input prompt $\mathbf{y}$ and processed through our safety classification model, for which we LoRA fine-tune (Hu et al., 2022) OPT-350M[12]. Internal test data is generated using our fine-tuned OPT-1.3B with a pre-trained safety cost model from SafeRLHF (Dai et al., 2024). We define a default response $\hat{\mathbf{d}}$ for cases when all candidates are classified as unsafe[13]. Note that we only trained the safety classification model for this experiment, with a threshold of $0.001$.

---

[9]Default action policy: if current speed is high, decelerate; if current speed is low, stop.

[10]During each epoch: when the agent enters a hazardous region, the environment resets; when the agent reaches the goal, only the goal location resets.

[11]PPO-Barrier trained for 30,000 epochs obtained negative rewards.

[12]We use LoRA fine-tuning and a smaller LLM to prevent overfitting and ensure stable training, following standard practice of using less expressive networks for reward models (Ouyang et al., 2022).

[13]Default response: "I'm sorry, I regret I cannot respond to this question. ..."

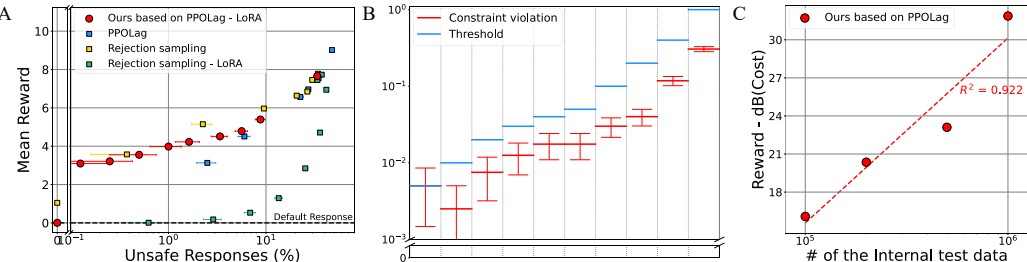

Figure 3: **Natural language generation.** (A) Our framework outperforms our baseline and abla-tions (rejection sampling). (B) Our framework successfully achieves constraint violations lower than designated thresholds. (C) Our framework demonstrates clear scaling properties.

**Results.** Figure 3-A illustrates the safety-performance trade-off. As in Figure 2-A, multiple points are achieved using different safety probability thresholds. Our framework (red points) outperforms the baseline AI model (OPT-1.3B fine-tuned with PPO-Lag, blue points). Similar to Section 6.1, we conduct ablation studies (yellow, green points) by disabling conservative testing and training the safety classification model with standard Cross-Entropy loss instead of our approximate loss, effectively reducing to *rejection sampling* methods. Notably, our framework outperforms the ablation with LoRA fine-tuning and achieves competitive performance compared to the ablation *without LoRA*. This demonstrates the potential of our framework for superior performance compared to recent rejection sampling methods.

### 6.3 Production Planning with Demand Prediction

To demonstrate the general applicability and ability of our framework to handle complex optimization problems, we experiment with *production planning*—a major industrial problem. The task involves optimizing production decisions based on predicted demands from historical demand data. We define "unsafe" as planning production (instead of stopping) when actual demand falls below a specified threshold. Unlike most optimization problems that handle linear constraints, ours involves challenging second-order cone constraints—which, to our knowledge, represents the *first attempt to jointly utilize such constraints with AI*. We demonstrate that our framework achieves higher revenue than baseline methods. Due to space limitations, detailed results are provided in Section H.

## 7 Conclusion

In this paper, we propose a domain-agnostic framework that ensures action safety while maintaining high performance across arbitrary AI models and domains. We combine an optimization component with the AI model and formulate safety constraints as chance constraints. We utilize a safety classification model to evaluate chance constraints, along with internal test data and conservative testing procedures. We introduce an approximate loss function and corresponding tailored gradient computation for end-to-end training. Finally, we mathematically establish and prove the first scaling law between the quantity of data and safety-performance trade-offs.

While our approach demonstrates broad effectiveness, several considerations merit discussion. The framework requires sufficient data to achieve high performance, though it scales effectively with increased data availability. The framework requires additional computational resources as discussed in Section K, though the overhead is modest (*e.g.,* only $18\%$ for natural language generation infer-ence). Adversarial attacks on unseen data may pose potential threats to safety guarantees; however, continuous updates to internal test data, combined with user-provided and continuously updated prior information, could enable rapid system adaptation and efficient attack mitigation. Performance may also be constrained when applying bias correction with thresholds that are substantially different from the training values.

Nevertheless, experimental validation across reinforcement learning, natural language generation, and production planning demonstrates the framework's broad applicability. The unprecedented safety levels achieved while maintaining competitive performance suggest promising transferability to other safety-critical domains. We expect this method to serve as a key milestone for safe and human-aligned deployment of AI applications.

ETHICS STATEMENT

Our domain-agnostic AI safety framework is designed to enhance safety across critical applications by providing mathematical guarantees for constraint satisfaction. We believe that advancing rigorous, mathematically grounded approaches to AI safety is essential for the responsible deployment of AI systems. However, we emphasize that proper validation with sufficient internal test data is crucial before deployment, as safety guarantees depend on the quality and representativeness of this data. We encourage researchers to apply these techniques responsibly and recommend thorough testing in controlled environments before considering real-world deployment in safety-critical applications.

REPRODUCIBILITY STATEMENT

To ensure reproducibility, we provide complete mathematical formulations, proofs, and algorithmic details in the Appendix sections referenced throughout the paper. Implementation specifics for all three experimental domains are detailed in the Appendix as well (reinforcement learning on SafetyGym: Section I, natural language generation: Section J, production planning: Section H), including hyperparameters, training procedures, data pre-processing steps, and experiment details. We include anonymous source code as supplementary materials containing our framework implementation and experimental scripts for generating the reported results. Note that a portion of the code for natural language generation experiments has been omitted due to licensing issues.

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

APPENDIX

## A  A GENERAL CONTROLLABLE PREDICT+OPTIMIZE FRAMEWORK

This section addresses decision-making problems that rely on *perception* performed by an *information processing module*, considering user-specified parameters. The information processing module encompasses any computational system, including random forests, regression models, genetic algorithms, neural networks, AI models, and other algorithmic approaches. The perception process includes any methodology for acquiring and processing information from the environment.

Within this paper's scope, we define these concepts as follows:

- **Information processing module**: The integrated computational system comprising both the AI model and the safety classification model described in the main text.
- **Perception**: The complete process encompassing environment state measurement and all subsequent computational steps required to generate the output $\mathbf{o}$.

The user-specified parameters can be regarded as knobs for users and used for customization. Since these parameters are a part of the optimization problem, which is more directly interpretable than an AI-based information processing module, this method can offer intuitive customization of AI. Moreover, our framework is a *general* form of an optimization problem, including continuous and discrete variables, prediction results of the perception outputs, and user-given customization parameters.

This section contains the following: First, we present our general problem setup. Second, we formulate our general framework, in which user-given customization parameters are included in the optimization problem.

### A.1  PROBLEM SETUP

We want to minimize $J(\mathbf{u}; \mathbf{s}, \mathbf{r})$ with variable $\mathbf{u} \in \mathcal{U} \subset \mathbb{R}^{n_{u1}} \times \mathbb{Z}^{n_{u2}}$, which correspond to the final action of our system, under user-given parameters $\mathbf{r} \in \mathbb{R}^{n_r}$ and environment state $\mathbf{s} \in \mathcal{S} \subset \mathbb{R}^{n_{s1}} \times \mathbb{Z}^{n_{s2}}$. Note that bold notations denote vectors. The real objective $J$ can be unknown, and we cannot directly handle it even if it is known, because it is a function of the environment state for which we do not have full information. We can measure[14] $\mathbf{y}$ that follows the function $Samp(\mathbf{s})$, which is the probability distribution of measurement from the environment state, and use it to obtain information about the environment state $\mathbf{s}$. Constraints $c_i(\mathbf{u}; \mathbf{s}, \mathbf{r}) \geq 0, \; i = 1, \ldots, n_c$ can also exist. Note that some constraints may depend on the environment state $\mathbf{s}$ and others may not. Without loss of generality, we assume that $c_i$ for $i = 1, \ldots, n_{cg}$ does not depend on $\mathbf{s}$ and $c_i$ for $i = n_{cg} + 1, \ldots, n_c$ depend on $\mathbf{s}$.

### A.2  PREDICT+OPTIMIZE FRAMEWORK FORMULATION

Our approach to the problem in Section A.1 consists of two parts: an information processing module that processes measurement results and an optimization stage. The information processing module takes the measurement result $\mathbf{y}$ as input and generates output $\mathbf{o} \in \mathcal{O}$. The processing information processing module may consider the user-given parameters $\mathbf{r}$ for customization as input to obtain a tailored output $\mathbf{o}$, thereby achieving the best performance under $\mathbf{r}$.

Then, we calculate optimal $\mathbf{u}$ according to the following optimization problem. Here, $\bar{J}(\mathbf{u}; \mathbf{o}, \mathbf{r})$ is the objective of the optimization stage, which is normally similar to the real objective $J(\mathbf{u}; \mathbf{s}, \mathbf{r})$ provided that the output $\mathbf{o}$ is well processed. Similarly, we have constraints of the optimization stage $\bar{c}_i(\mathbf{u}; \mathbf{o}, \mathbf{r}) \geq 0, \; i = 1, \ldots, n_{\bar{c}}$, which may partially or fully reflect the real constraints $c_i(\mathbf{u}; \mathbf{s}, \mathbf{r}) \geq 0, \; i = 1, \ldots, n_c$.

$$\min_{\mathbf{u} \in \mathcal{U}} \quad \bar{J}(\mathbf{u}; \mathbf{o}, \mathbf{r}) \tag{13a}$$

$$\text{subject to} \quad \bar{c}_i(\mathbf{u}; \mathbf{o}, \mathbf{r}) \geq 0, \; i = 1, \ldots, n_{\bar{c}} \tag{13b}$$

Note that the final output of the whole system is the action $\mathbf{u}$, and the performance of the entire system depends only on $\mathbf{u}$ in general. Nevertheless, we deal with a highly general loss function that can include the optimal objective of the optimization stage or the prediction accuracy (Section D).

---

[14]The input $\mathbf{y}$ could also be thought of as a measurement from the environment.

# B    DEALING CONSTRAINTS: CHANCE-CONSTRAINED METHOD

Safety is one of the most significant concerns associated with AI-based decision-making. Pure machine-learning-based algorithms are trained by minimizing loss functions only, and it is not straightforward to directly enforce constraint satisfaction. Hence, their safety is difficult to guarantee entirely. Some studies (García & Fernández, 2015; Gros et al., 2020) have been conducted to ensure safety by projecting the results into a safe area to overcome this. However, they also have the disadvantages of being suboptimal and failing to guarantee the satisfaction of nondeterministic constraints. This highlights why pure deep learning-based methods may face challenges when deployed in safety-critical applications such as autonomous driving of passenger cars.

This section presents the method that adopts chance-constrained optimization into our general formulation to guarantee probabilistic constraint satisfaction. To ensure that the constraints are satisfied above the given probability, the performance of the information processing module is evaluated with labeled sample data before the optimization stage. The probability of each actual situation (environment state) for each result of the information processing module (= posterior probability) is calculated based on the prior probability of each situation, as specified by the custom parameters, and the evaluation result. Reflecting on this, we determine how conservatively we decide our action (control output) in the optimization stage to satisfy the constraint with a probability above a given value. In the learning stage, the loss function is defined and learned based on the performance of these conservative actions to ensure constraint satisfaction.

This section contains the following content: First, we provide a general discussion about dealing with constraints in decision-making with perception. Second, we formulate our chance-constrained method to guarantee safety in the framework presented in the previous section. Third, we present the technique for obtaining conservative actions and prove the conditions under which it can be a valid approach for training the chance-constrained method within this framework.

## B.1    CONSTRAINTS: HOW TO DEAL?

In general, we have some constraints that our actions should satisfy. These constraints can be divided into two groups according to whether they are *deterministic*. Deterministic constraints do not depend on the environment state ($\mathbf{s}$ in the problem setup from the former section), thus we do not need to measure $\mathbf{s}$ to know the constraint. Constraints in this group are generally easy to deal with because they can be directly considered in the decision-making stage (optimization stage in our formulation). Even though a method that cannot deal with constraints is used, such as pure machine learning, the satisfaction of constraints in this group can be guaranteed by post-processing the action (*e.g.,* projecting it to the constraint-satisfying region).

Conversely, some constraints can be *nondeterministic*. These constraints may depend on $\mathbf{s}$; thus, we should obtain the information about the environment state to satisfy the constraint. Note that constraints that include pure randomness, such as random variables following a standard normal distribution, are also classified in this group since such pure randomness can also be a part of $\mathbf{s}$. Nondeterministic constraints cannot be considered directly in the decision-making stage. Instead, we can pursue satisfying them for probabilities larger than given *probability values*. The central concept of this section is obtaining the posterior distribution of $\mathbf{s}$ and taking our action conservatively to guarantee constraint satisfaction for these constraints over the given probabilities.

## B.2    CHANCE-CONSTRAINED METHOD

In this subsection, we present the chance-constrained method to guarantee the satisfaction of environment state-dependent constraints ($c_i$ for $i = n_{cg} + 1, \ldots, n_c$). We divide the environment state into two parts: one containing information needed to satisfy some constraints, and the other part containing no constraint-relevant information:

$$\mathbf{s} = (\mathbf{s}_{cr}(\in \mathcal{S}_{cr}), \mathbf{s}_{ncr}(\in \mathcal{S}_{ncr})) \tag{14}$$

where $c_i$ for $i = n_{cg} + 1, \ldots, n_c$ are functions only of $\mathbf{s}_{cr}$. Then, the chance-constraint can be described as follows. Note that the total probability of the dissatisfaction of the $i$-th original constraint (*i.e.,* the probability that our system's action violates this constraint) is upper-limited as $r_{t,i}$. We assume that a $(n_c - n_{cg})$-dimensional vector $(r_{t,n_{cg}+1}, \ldots, r_{t,n_c})$ is also included in $\mathbf{r}$.

$$\forall i, \quad Pr(c_i(\mathbf{u}; \mathbf{s}_{cr}, \mathbf{r}) < 0) \leq r_{t,i} \tag{15}$$

To clarify and calculate the left-hand side, we also divide the information processing module output $\mathbf{o}$ into:

$$\mathbf{o} = (\mathbf{o}_{cr}(\in \mathcal{O}_{cr}), \mathbf{o}_{ncr}(\in \mathcal{O}_{ncr})) \tag{16}$$

such that the former part includes information about $\mathbf{s}_{cr}$ and the latter part does not. We call the part of the information processing module regarding safety classification as *safety classification model*. Moreover, the measurement $\mathbf{y}$ is also divided into:

$$\mathbf{y} = (\mathbf{y}_{cr}, \mathbf{y}_{ncr}) \tag{17}$$

$\mathbf{y}_{cr}$ is defined as the part needed to obtain $\mathbf{o}_{cr}$, and $\mathbf{y}_{ncr}$ is defined as the remaining part. Thus, $\mathbf{y}_{cr}$ can include more information than $\mathbf{s}_{cr}$. In this paper, for simplicity, we only deal with the case that both $\mathcal{S}_{cr}$ and $\mathcal{O}_{cr}$ are *finite*. This clearly implies that neither $\mathbf{s}_{cr}$ nor $\mathbf{o}_{cr}$ has any continuous part.

We apply prior work (Kim & Ahn, 2023) to our framework (presented in the former section) to ensure constraint satisfaction over the given probability. Specifically, we conduct perception for not only the given environment state but also internal test data with which we already know the environment state. Since it presents a challenge that internal test data will not be changed during training, and thus can lead to overfitting, we will address this problem in Section C.2. Then, we use Bayes' rule to merge the results from internal test data with the user-given prior probabilities for each possible environment state, obtaining the posterior probability distribution given the perception of the real (unknown) environment state. Using these results, we solve the chance-constrained optimization problem under the user-given threshold.

To avoid confusion, we use superscript $r$ to denote real quantities (environment state, measurement, etc.) and superscript $t_i$ to denote the $i$-th (internal) test data. For example, $\mathbf{s}^r$ denotes the real environment state of the control situation, and $\mathbf{s}^{t_i}$ denotes the known environment state of the $i$-th test data.

In addition to the measurement of the real environment state, we also measure the part that includes information about constraints for each internal test data. Thus, letting $n_t$ denote the number of internal test data, we define $\mathbf{y}$ as Equation 18:

$$\mathbf{y} = (\mathbf{y}^r, \mathbf{y}_{cr}^{t_1}, \dots, \mathbf{y}_{cr}^{t_{n_t}}), \quad \mathbf{y}^r \sim Samp(\mathbf{s}^r), \quad \mathbf{y}_{cr}^{t_i} \sim Samp_{cr}(\mathbf{s}_{cr}^{t_i}) \tag{18}$$

Note that $Samp(\mathbf{s})$ denotes the measurement process, which can be treated as sampling from the probability distribution given by the real environment state $\mathbf{s}$. Since we only require the constraint-related ($cr$) parts of the internal test data, we use the subscript $cr$ for $\mathbf{y}^t$, where $Samp_{cr}$ denotes applying $Samp$ to these $cr$ parts. Given that $\mathbf{s}_{cr}^{t_i}$ are fixed, $\mathbf{y}$ still depends only on $\mathbf{s}^r$.

Then, we forward all measurement results in parallel into the information processing module, as shown in Equation 19. Here, $f_{cr}(\mathbf{y}_{cr}^{t_i}; \mathbf{r})$ denotes the safety classification model, which is part of the information processing module responsible for generating outputs that contain information about $\mathbf{s}_{cr}$. Thus, Equation 19 can be interpreted as a large information processing module made by merging one original information processing module and $n_t$ safety classification models in parallel.

$$\mathbf{F}((\mathbf{y}^r, \mathbf{y}_{cr}^{t_1}, \dots, \mathbf{y}_{cr}^{t_{n_t}}); \mathbf{r}) := (\mathbf{o}^r, \mathbf{o}_{cr}^{t_1} \dots, \mathbf{o}_{cr}^{t_{n_t}}) := (f(\mathbf{y}^r; \mathbf{r}), f_{cr}(\mathbf{y}_{cr}^{t_1}; \mathbf{r}) \dots, f_{cr}(\mathbf{y}_{cr}^{t_{n_t}}; \mathbf{r})) \tag{19}$$

As the next step, we estimate the probability for each output $\mathbf{o}_{cr}$ under each environment state $\mathbf{s}_{cr}$. We denote the elements of $\mathcal{S}_{cr}$ and $\mathcal{O}_{cr}$ as:

$$\mathcal{S}_{cr} = \{\bar{\mathbf{s}}_1, \dots, \bar{\mathbf{s}}_{N_e}\}, \quad \mathcal{O}_{cr} = \{\bar{\mathbf{o}}_1, \dots, \bar{\mathbf{o}}_{N_o}\} \tag{20}$$

We can enumerate the elements as Equation 20 because we assumed that $\mathcal{S}_{cr}$ and $\mathcal{O}_{cr}$ are finite. We also let $\mathbf{1}(a, b)$ as the indicator function that outputs 1 if $a$ is the same as $b$ and 0 otherwise. Then, the total number of internal test data for each environment state is:

$$N_{\mathbf{s}_{cr} = \bar{\mathbf{s}}_i} = \sum_{k=1}^{n_t} \mathbf{1}(\mathbf{s}_{cr}^{t_k}, \bar{\mathbf{s}}_i) \tag{21}$$

The number of internal test data for each output and environment state is:

$$N_{\mathbf{s}_{cr}=\bar{\mathbf{s}}_i, \mathbf{o}_{cr}=\bar{\mathbf{o}}_j} = \sum_{k=1}^{n_t} \mathbf{1}(\mathbf{s}_{cr}^{t_k}, \bar{\mathbf{s}}_i) \cdot \mathbf{1}(\mathbf{o}_{cr}^{t_k}, \bar{\mathbf{o}}_j) \tag{22}$$

Then, the probability for each output under each environment state can be estimated using sample proportions, as Equation 23.

$$p(\mathbf{o}_{cr} = \bar{\mathbf{o}}_j | \mathbf{s}_{cr} = \bar{\mathbf{s}}_i) \simeq \frac{N_{\mathbf{s}_{cr}=\bar{\mathbf{s}}_i, \mathbf{o}_{cr}=\bar{\mathbf{o}}_j}}{N_{\mathbf{s}_{cr}=\bar{\mathbf{s}}_i}} \tag{23}$$

We assume that $\mathbf{r}$ includes a $N_e$-dimensional probability vector $(r_{ep,1}, \ldots, r_{ep,N_e})$ with $0 \leq r_{ep,k} \leq 1$ and $\sum_{k=1}^{N_e} r_{ep,k} = 1$, representing knowledge about the prior probability of constraint-relevant environment states. Using Bayes' rule, we can compute the posterior probability of each environment state given each safety classification model output as Equation 24:

$$\begin{aligned} p(\mathbf{s}_{cr} = \bar{\mathbf{s}}_i | \mathbf{o}_{cr} = \bar{\mathbf{o}}_j) &= \frac{p(\mathbf{o}_{cr} = \bar{\mathbf{o}}_j | \mathbf{s}_{cr} = \bar{\mathbf{s}}_i) p(\mathbf{s}_{cr} = \bar{\mathbf{s}}_i)}{p(\mathbf{o}_{cr} = \bar{\mathbf{o}}_j)} \\ &= \frac{p(\mathbf{o}_{cr} = \bar{\mathbf{o}}_j | \mathbf{s}_{cr} = \bar{\mathbf{s}}_i) p(\mathbf{s}_{cr} = \bar{\mathbf{s}}_i)}{\sum_{k=1}^{N_e} p(\mathbf{o}_{cr} = \bar{\mathbf{o}}_j | \mathbf{s}_{cr} = \bar{\mathbf{s}}_k) p(\mathbf{s}_{cr} = \bar{\mathbf{s}}_k)} \\ &= \frac{p(\mathbf{o}_{cr} = \bar{\mathbf{o}}_j | \mathbf{s}_{cr} = \bar{\mathbf{s}}_i) r_{ep,i}}{\sum_{k=1}^{N_e} p(\mathbf{o}_{cr} = \bar{\mathbf{o}}_j | \mathbf{s}_{cr} = \bar{\mathbf{s}}_k) r_{ep,k}} \end{aligned} \tag{24}$$

We can now obtain the left-hand side of Equation 15 as shown in Equation 25 to replace an environment state-dependent constraint, following the method used in (Kim & Ahn, 2023). Since the measurement and processing for the real environment state are done and the resulting output $\mathbf{o}^r$ is obtained, we can get the post-perception probability of each environment state based on Equation 24 and the test results. We ensure that the sum of probabilities for the neglected possible environment states does not exceed the given threshold for the specific constraints.

$$\sum_{k:c(\mathbf{u};\bar{\mathbf{s}}_k, \mathbf{r})<0} p(\mathbf{s}_{cr} = \bar{\mathbf{s}}_k | \mathbf{o}_{cr} = \mathbf{o}_{cr}^r) \leq r_{t,i} \tag{25}$$

We can modify Equation 25 based on the big-M method (Cococcioni & Fiaschi, 2021) to solve it with a solver. With a sufficiently large number $M$, we adopt integer variables $\{q_i\}$ to identify the neglected environment states (in our case, $q_i = 0$ or $q_i = 1$). Then, we can formulate a set of constraints to replace the $i$-th ($i = n_{cg} + 1, \ldots, n_c$) original constraint as:

$$c(\mathbf{u}; \bar{\mathbf{s}}_k, \mathbf{r}) + M \cdot q_k \geq 0, \qquad k = 1, \ldots, N_e \tag{26a}$$

$$\sum_k p(\mathbf{s}_{cr} = \bar{\mathbf{s}}_k | \mathbf{o}_{cr} = \mathbf{o}_{cr}^r) \cdot q_k \leq r_{t,i} \tag{26b}$$

Finally, we construct a replacement of the environment state-dependent constraints $c_i$ for $i = n_{cg} + 1, \ldots, n_c$ by compressing the chance-constraint formulation (Equation 26). Then, the replacement of the $i$-th original constraint ($n_{cg} + 1 \leq i \leq n_c$) can be written as:

$$\bar{c}_i(\mathbf{u}; \mathbf{o}, \mathbf{r}) := \max_{\substack{q_1, \ldots, q_{N_e} \in \{0,1\} \\ \sum_k p(\mathbf{s}_{cr}=\bar{\mathbf{s}}_k|\mathbf{o}_{cr}=\mathbf{o}_{cr}^r) \cdot q_k \leq r_{t,i}}} \min_k \left( c_i(\mathbf{u}; \bar{\mathbf{s}}_k, \mathbf{r}) + M \cdot q_k \right) \tag{27}$$

Detailed explanation for our chance-constrained formulation (Equations 18, 19, 23, 24, and 27) can be found in (Kim & Ahn, 2023). We conclude this subsection by noting that this formulation is a case of the general framework presented in Section A.

**Proposition 1.** When both $\mathcal{S}_{cr}$ and $\mathcal{O}_{cr}$ are *finite*, our compressed constraint replacement (Equation 27) based on modified chance-constrained formulation (Equation 15) is included into our general optimization problem (Equation 13). In addition, the whole procedure, including the processing of the concatenated measurement result and calculation process (Equations 18, 19, 23, and 24), is included in the general framework presented in Section A. Moreover, replacements of constraints are continuous with respect to the continuous part of $\mathbf{u}$ and $\mathbf{o}$, provided that the original constraints to be replaced are continuous with respect to the continuous part of $\mathbf{u}$.

*Proof.* First, since the environment states of internal test data are given, the entire measurement depends solely on the environment state of the real situation and chance. Then, the measurement can be treated as measuring the real environment state. In Equation 19, the information processing module can be understood as a new large model to process the concatenated vector. Moreover, the results of Equation 23 and 24 can be substituted in Equation 27. Since we assume that the constraint-related part of the output is discrete, continuous outputs have no effect on Equation 27 and the replaced constraints are stationary with respect to the continuous part of the information processing module output. Furthermore, considering that Equation 27 is constructed by the maximum and minimum of finite continuous functions with respect to the continuous part of $\mathbf{u}$, it is continuous with respect to the continuous part of $\mathbf{u}$. □

Note that constraints affected by the continuous part of the environment state $\mathbf{s}$ or related to the continuous part of the information processing module output $\mathbf{o}$ can be handled as constraints in the general framework Equation 13. In such cases, where our assumption of finite $\mathcal{S}_{cr}$ no longer holds, mathematical guarantees for constraint satisfaction cannot be provided. However, these constraints can still be addressed based on the information processing module output, and we can reasonably expect practical satisfaction when the perception capability of the information processing module is sufficiently good.

## C CONSERVATIVE TESTING WITH INTERNAL TEST DATA

In this section, we present our novel method for training our safety classification model without overfitting issues when using internal test data. We then prove its feasibility and validity under several assumptions. When we train our safety classification model, it begins to depend on internal test data. This may lead to statistical validity concerns. To theoretically guarantee improved safety classification model performance when the loss sufficiently decreases, we make the chance-constrained optimization in training conservative. This ensures that training loss becomes an upper bound of the loss when we know the real performance of our safety classification model $p(\mathbf{o}_{cr} = \bar{\mathbf{o}}_j | \mathbf{s}_{cr} = \bar{\mathbf{s}}_i)$. To achieve this, we introduce a positive real number $\xi$ as a conservativeness parameter that controls and guarantees the conservativeness of chance constraints during training relative to reality. This parameter plays a crucial role in maintaining or verifying the validity of the training technique. In practice (including our experiments), this $\xi$ can be treated as a hyperparameter with a value that efficiently inhibits overfitting without considerable performance degradation, even though the value may be smaller than required for theoretical guarantees.

### C.1 DEFINING REAL PERFORMANCE

First, we mathematically define the real performance. We start at a general metric space $\Omega$ endowed with metric $M : \Omega \times \Omega \to [0, \infty]$ and $\sigma$-algebra $\Sigma$. Then, to treat the measurement of the environment state, we define a probability measure $P : \Sigma \to [0, 1]$. Let $\mathcal{Y}_{cr}$ denote the set of possible $\mathbf{y}_{cr}$ (thus, for both $\mathbf{y}^r$ and $\mathbf{y}^t$). Then, the conditions for defining the real performance of the safety classification model can be summarized as Condition 1.

**Condition 1.** 1. We can implement a metric $M_{\mathcal{Y}_{cr}}$ in $\mathcal{Y}_{cr}$.

2. We can choose a Borel $\sigma$-algebra $\Sigma_{\mathcal{Y}_{cr}}$.

3. For each $\mathbf{s}_{cr}$, according to $Samp_{cr}(\mathbf{s}_{cr})$, a probability measure $P_{\mathbf{s}_{cr}} : \Sigma_{\mathcal{Y}_{cr}} \to [0, 1]$ can be defined.

4. The safety classification model classifies the data into finite potential outputs; that is, the model can be defined as a finite partition of $\mathcal{Y}_{cr}$ denoted as $Y_{\bar{\mathbf{o}}_1}, \ldots, Y_{\bar{\mathbf{o}}_{N_o}}$ [15]. Moreover, $Y_{\bar{\mathbf{o}}_j} \in \Sigma_{\mathcal{Y}_{cr}}$ for each $j$.

Condition 1 is the essential property needed for mathematical analysis and holds in general. Under Condition 1, the real performance of the safety classification model (*i.e.,* the real probability of the output $\bar{\mathbf{o}}_j$ under environment state $\bar{\mathbf{s}}_i$) can be defined as follows:

$$p^*(\mathbf{o}_{cr} = \bar{\mathbf{o}}_j | \mathbf{s}_{cr} = \bar{\mathbf{s}}_i) := P_{\bar{\mathbf{s}}_i}(Y_{\bar{\mathbf{o}}_j}) \tag{28}$$

We can also obtain the replacement of constraints based on Equation 28, similar to Equations 26 and 27, as follows. Note that compared to Equation 27, $p(\mathbf{s}_{cr} = \bar{\mathbf{s}}_i | \mathbf{o}_{cr} = \bar{\mathbf{o}}_j)$ is replaced with $p^*(\mathbf{s}_{cr} = \bar{\mathbf{s}}_i | \mathbf{o}_{cr} = \bar{\mathbf{o}}_j)$.

$$p^*(\mathbf{s}_{cr} = \bar{\mathbf{s}}_i | \mathbf{o}_{cr} = \bar{\mathbf{o}}_j) := \frac{p^*(\mathbf{o}_{cr} = \bar{\mathbf{o}}_j | \mathbf{s}_{cr} = \bar{\mathbf{s}}_i) r_{ep,i}}{\sum_{k=1}^{N_e} p^*(\mathbf{o}_{cr} = \bar{\mathbf{o}}_j | \mathbf{s}_{cr} = \bar{\mathbf{s}}_k) r_{ep,k}} \tag{29}$$

$$\bar{c}_i^*(\mathbf{u}; \mathbf{o}, \mathbf{r}) := \max_{\substack{q_1, \ldots, q_{N_e} \in \{0,1\} \\ \sum_k p^*(\mathbf{s}_{cr} = \bar{\mathbf{s}}_k | \mathbf{o}_{cr} = \mathbf{o}_{cr}^r) \cdot q_k \leq r_{t,i}}} \min_k \left( c(\mathbf{u}; \bar{\mathbf{s}}_k, \mathbf{r}) + M \cdot q_k \right) \tag{30}$$

During training, we assume that the measurements of the same set of internal test data remain constant, thus we reuse the measurement results $(\mathbf{y}_{cr}^{t_1}, \ldots, \mathbf{y}_{cr}^{t_{n_t}})$ after the initial measurement. Since $(\mathbf{y}_{cr}^{t_1}, \ldots, \mathbf{y}_{cr}^{t_{n_t}})$ does not depend on the information processing module, it is natural to define conditions for good internal test data for training. The following definition describes how well the internal test data covers the probability distribution over a given set:

---

[15] The safety classification model takes $\mathbf{y}$ as input and produces $\mathbf{o}$ as output. $Y_{\bar{\mathbf{o}}_j}$ represents the set of all $\mathbf{y}$ values that result in the output $\bar{\mathbf{o}}_j$.

**Definition 1.** Given a compact metric space $\Omega$ endowed with metric $M : \Omega \times \Omega \to [0, \infty]$, $\sigma$-algebra $\Sigma$ that includes all open balls $B$, and a probability measure $P : \Sigma \to [0, 1]$, a finite sequence[16] $(\omega_1, \ldots, \omega_{n_s})$ of elements in $\Omega$ is called $\zeta$-*informative* if and only if the following statement holds for all subsets $X$ of $\{1, \ldots, n_s\}$.

$$P(\bigcup_{i \in X} B(\omega_i, \zeta)) \geq \frac{|X|}{n_s} \tag{31}$$

This implies that the margin $\zeta$ is robust to the selection of the subset of data. The safety classification model with discrete outputs classifies data, where each classification can be viewed as selecting a subset. Thus, this definition provides robustness for any model or parameters.

Now, we introduce a proposition to guarantee that we can achieve $\zeta$-informativeness for arbitrary $\zeta$ by sampling sufficiently many internal test data from the probability distribution.

**Proposition 2.** If the support of each $P$ is connected, when we sample dataset $D$ following $P$, then $Pr\left(\lim_{|D| \to \infty} \inf(\{\zeta | D : \zeta - \text{informative}\}) = 0\right) = 1$.

*Proof.* Without loss of generality, for simplicity, we assume $Diam(\Omega) = 1$.[17] For any $\delta > 0$, $\{B(\omega, \frac{\delta}{8}) | \omega \in \Omega\}$ is obviously an open cover of $\Omega$. Thus, due to the definition of compact sets, the following finite subcover exists:

$$\left\{ B\left(\omega_1, \frac{\delta}{8}\right), \ldots, B\left(\omega_{n_h}, \frac{\delta}{8}\right) \right\} \tag{32}$$

We define $H_i$ as:

$$H_i := \bigcup_{j=1}^{i} B\left(\omega_i, \frac{\delta}{8}\right) \setminus \bigcup_{j=1}^{i-1} B\left(\omega_j, \frac{\delta}{8}\right) \tag{33}$$

$\{H_i\}$ is a partition of $\Omega$ whose elements have a diameter of at most $\delta/4$. Since we assume that each $P$ is well-defined on a $\sigma$-algebra that includes all open sets, each $P$ is well-defined for $H_i$. Since $n_h < \infty$, we can define $\rho$ as:

$$\rho := \min_{i:P(H_i)>0} P(H_i) \tag{34}$$

Due to the strong law of large numbers, provided that $D \neq \phi$:

$$Pr\left(\lim_{|D| \to \infty} \frac{|D \cap H_i|}{|D|} = P(H_i)\right) = 1 \tag{35}$$

This means, provided that $D \neq \phi$:

$$Pr\left(\exists N, |D| > N \to \frac{|D \cap H_i|}{|D|} - \frac{\rho}{n_h} \leq P(H_i)\right) = 1 \tag{36}$$

For any nonempty $X \subset D$, following the definition of $H_i$:

$$\bigcup_{x \in X} B\left(x, \frac{\delta}{4}\right) \supset \bigcup_{x \in X} \bigsqcup_{i:x \in H_i} H_i = \bigsqcup_{i:X \cap H_i \neq \phi} H_i.$$

Moreover, it is obvious that:

$$\overline{\bigcup_{x \in X} B\left(x, \frac{3\delta}{4}\right)} \supset \bigcup_{x \in X} B\left(x, \frac{\delta}{2}\right) \tag{37}$$

---

[16]This definition can be also applied to a set.

[17]Thus, we set the maximum distance between any two points in $\Omega$ to be 1.

$$\overline{\Omega \setminus \overline{\bigcup_{x \in X} B\left(x, \frac{3\delta}{4}\right)}} \cap \overline{\bigcup_{x \in X} B\left(x, \frac{\delta}{2}\right)} = \phi. \tag{38}$$

where overline indicates a closure including its boundary (thus $\overline{B}$ is a closed ball). Then, $\bigcup_{x \in X} B(x, \frac{\delta}{2})$ and $\Omega \setminus \overline{\bigcup_{x \in X} B(x, \frac{3\delta}{4})}$ are disjoint open sets of $\Omega$. By the assumption that the support is connected, we show that at least one of the following statements holds:

$$S_1. \quad P\left(\overline{\bigcup_{x \in X} B\left(x, \frac{3\delta}{4}\right)} \setminus \bigcup_{x \in X} B\left(x, \frac{\delta}{2}\right)\right) > 0 \tag{39}$$

$$S_2. \quad P\left(\bigcup_{x \in X} B\left(x, \frac{\delta}{2}\right)\right) = 0 \tag{40}$$

$$S_3. \quad P\left(\Omega \setminus \overline{\bigcup_{x \in X} B\left(x, \frac{3\delta}{4}\right)}\right) = 0 \tag{41}$$

For proof, let's assume that all $S_1$, $S_2$, $S_3$ don't hold. Then,

$$\text{int}\left(\overline{\bigcup_{x \in X} B\left(x, \frac{3\delta}{4}\right)} \setminus \bigcup_{x \in X} B\left(x, \frac{\delta}{2}\right)\right) \cap \text{supp}(P) = \phi \tag{42}$$

Thus,

$$\text{supp}(P) \subset \overline{\Omega \setminus \overline{\bigcup_{x \in X} B\left(x, \frac{3\delta}{4}\right)}} \cup \overline{\bigcup_{x \in X} B\left(x, \frac{\delta}{2}\right)} \tag{43}$$

where $\text{int}(\cdot)$ is the interior of the set and $\text{supp}(\cdot)$ is the support of measure. Let

$$J := \overline{\bigcup_{x \in X} B\left(x, \frac{\delta}{2}\right)} \cap \text{supp}(P) \tag{44}$$

and

$$K := \overline{\Omega \setminus \overline{\bigcup_{x \in X} B\left(x, \frac{3\delta}{4}\right)}} \cap \text{supp}(P) \tag{45}$$

Then, $J \cap K = \phi$, $J \cup K = \text{supp}(P)$, and both are nonempty since their measure $P$ is nonzero (since we are assuming that $S_2$ and $S_3$ do not hold). Since $\overline{\bigcup_{x \in X} B\left(x, \frac{\delta}{2}\right)}$ and $\Omega \setminus \overline{\bigcup_{x \in X} B\left(x, \frac{3\delta}{4}\right)}$ are open sets in $\Omega \setminus \overline{\bigcup_{x \in X} B\left(x, \frac{3\delta}{4}\right)} \cup \overline{\bigcup_{x \in X} B\left(x, \frac{\delta}{2}\right)}$ (because the complements, which is each other, are closed), $J$ and $K$ are open sets in $\text{supp}(P)$ by definition. Thus, it contradicts the assumption that $\text{supp}(P)$ is connected, and at least one of $S_1$, $S_2$, $S_3$ should hold.

Since we assume that $D$ is sampled according to $P$, the probability of $S_2$ to hold is $0$.[18] Thus, we neglect it. $S_3$ implies:

$$P\left(\bigcup_{x \in X} B(x, \delta)\right) \geq P\left(\overline{\bigcup_{x \in X} B\left(x, \frac{3\delta}{4}\right)}\right)$$
$$= 1 - P\left(\Omega \setminus \overline{\bigcup_{x \in X} B\left(x, \frac{3\delta}{4}\right)}\right) \tag{46}$$
$$= 1 \tag{$S_3$}$$

---

[18] $S_2$ implies that data can be sampled from balls $B(x, \frac{\delta}{2})$ with probability $0$, contradicting the fact that $x$ was sampled.

If $S_1$ holds, let

$$T := \overline{\bigcup_{x \in X} B\left(x, \frac{3\delta}{4}\right)} \setminus \bigcup_{x \in X} B\left(x, \frac{\delta}{2}\right). \tag{47}$$

Since $\{H_i | T \cap H_i \neq \phi\}$ is a cover of $T$,

$$\bigsqcup_{i:H_i \cap T \neq \phi} H_i \supset T \tag{48}$$

Considering that $H_i$ are all disjoint, we obtain

$$\begin{aligned}
\sum_{i:H_i \cap T \neq \phi} P(H_i) &= P\left(\bigsqcup_{i:H_i \cap T \neq \phi} H_i\right) \\
&\geq P(T) \\
&> 0 \qquad\qquad (S_1)
\end{aligned} \tag{49}$$

which implies that there exists $H_k$ that satisfies $H_k \cap T \neq \phi$ and $P(H_k) > 0$. Considering that $Diam(H_i) \leq \frac{\delta}{4}$, $H_k \cap T \neq \phi$ implies $H_k \cap (\bigcup_{x \in X} B(x, \frac{\delta}{4})) = \phi$ and $H_k \subset \bigcup_{x \in X} B(x, \delta)$. Thus,

$$\begin{aligned}
P\left(\bigcup_{x \in X} B(x, \delta)\right) &\geq P(H_k) + P\left(\bigcup_{x \in X} B\left(x, \frac{\delta}{4}\right)\right) \quad \text{(monotonicity \& additivity)} \\
&\geq \rho + P\left(\bigcup_{x \in X} \bigsqcup_{i:x \in H_i} H_i\right) \qquad \text{(monotonicity)} \\
&= \rho + P\left(\bigsqcup_{i:X \cap H_i \neq \phi} H_i\right) \qquad \text{(set algebra)} \\
&= \rho + \sum_{i:X \cap H_i \neq \phi} P(H_i) \qquad \text{(additivity)}
\end{aligned} \tag{50}$$

Moreover, we have

$$\sum_{i:X \cap H_i \neq \phi} \frac{|D \cap H_i|}{|D|} \geq \frac{|X|}{|D|} \tag{51}$$

provided that $D \neq \phi$ since $\{H_i | X \cap H_i \neq \phi\}$ is a cover of $X$. Thus, by Equation 36, Equation 50, and Equation 51, we obtain

$$Pr\left(\exists N, |D| > N \to P(\bigcup_{x \in X} B(x, \delta)) \geq \frac{|X|}{|D|}\right) = 1. \tag{52}$$

Considering that the probability of satisfaction of Equation 40 is 0 and Equation 41 implies Equation 46, we can conclude that

$$Pr\left(\exists N, |D| > N \to P(\bigcup_{x \in X} B(x, \delta)) \geq \frac{|X|}{|D|}\right) = 1, \tag{53}$$

holds for any $X$, equivalently,

$$Pr(\exists N, |D| > N \to D : \delta - \text{informative}) = 1. \tag{54}$$

Moreover, this implies that

$$Pr(\exists N, |D| > N \to \inf(\{\zeta | D : \zeta-\text{informative}\}) \leq \delta) = 1. \tag{55}$$

Therefore, considering that $\delta$ is an arbitrary positive real number, we obtain

$$Pr\left(\lim_{|D| \to \infty} \inf(\{\zeta | D : \zeta-\text{informative}\}) = 0\right) = 1. \tag{56}$$

$\square$

## C.2 CONSERVATIVE TESTING

Since both $\mathcal{S}_{cr}$ and $\mathcal{O}_{cr}$ are finite, we can use the idea of *conservative testing*—introducing a penalty term in the continuous intermediate result of the safety classification model just before the discrete output to obtain a robust evaluation result. We can let

$$\mathbf{o}_{cr} = g_{cr}(f_{cr}(\mathbf{y}; \mathbf{w}_{cr})), \quad f_{cr}(\mathbf{y}; \mathbf{w}_{cr}) \in \mathbb{R}^{m_{cr}} \tag{57}$$

and

$$\mathbf{1}(\mathbf{o}_{cr}^{t_k}, \bar{\mathbf{o}}_j) = \mathbf{1}(g_{cr}(f_{cr}(\mathbf{y}; \mathbf{w}_{cr})^{t_k}), \bar{\mathbf{o}}_j) \tag{58}$$

when $f_{cr}(\mathbf{y}; \mathbf{w}_{cr})^{t_k}$ is $f_{cr}(\mathbf{y}; \mathbf{w}_{cr})$ for the $k$-th (internal) test data. Note that the intermediate output of the safety classification model $f_{cr}(\mathbf{y}; \mathbf{w}_{cr})$ is a function of the input of the safety classification model $\mathbf{y}_{cr}$. Here, we do not separate $\mathbf{o}_{cr}$ ($= f_{cr}(\mathbf{y}; \mathbf{w}_{cr})$) to their elements for simplicity.

To obtain the conservative result of chance-constrained optimization, we also calculate:

$$\mathbf{1}^{+\xi}(\mathbf{o}_{cr}^{t_k}, \bar{\mathbf{o}}_j) = \max_{||\boldsymbol{\omega}_{cr} - f_{cr}(\mathbf{y}; \mathbf{w}_{cr})^{t_k}|| \leq \xi} \mathbf{1}(g_{cr}(\boldsymbol{\omega}_{cr}), \bar{\mathbf{o}}_j) \tag{59}$$

and

$$\mathbf{1}^{-\xi}(\mathbf{o}_{cr}^{t_k}, \bar{\mathbf{o}}_j) = \min_{||\boldsymbol{\omega}_{cr} - f_{cr}(\mathbf{y}; \mathbf{w}_{cr})^{t_k}|| \leq \xi} \mathbf{1}(g_{cr}(\boldsymbol{\omega}_{cr}), \bar{\mathbf{o}}_j) \tag{60}$$

The intuition is to check whether all points in the $\xi$-ball centered on $\boldsymbol{\omega}_{cr}$ consistently lead or consistently do not lead to the output value $\bar{\mathbf{o}}_j$. Note that calculating $\mathbf{1}^{+\xi}(\mathbf{o}_{cr}^{t_k}, \bar{\mathbf{o}}_j)$ and $\mathbf{1}^{-\xi}(\mathbf{o}_{cr}^{t_k}, \bar{\mathbf{o}}_j)$ is easy since $f_{cr}$ is a simple function in general. Then, we also define the upper bound and the lower bound of the number of internal test data for each output and environment state as

$$N_{\mathbf{s}_{cr}=\bar{\mathbf{s}}_i, \mathbf{o}_{cr}=\bar{\mathbf{o}}_j}^{+\xi} = \sum_{k=1}^{n_t} \mathbf{1}(\mathbf{s}_{cr}^{t_k}, \bar{\mathbf{s}}_i) \mathbf{1}^{+\xi}(\mathbf{o}_{cr}^{t_k}, \bar{\mathbf{o}}_j) \tag{61}$$

and

$$N_{\mathbf{s}_{cr}=\bar{\mathbf{s}}_i, \mathbf{o}_{cr}=\bar{\mathbf{o}}_j}^{-\xi} = \sum_{k=1}^{n_t} \mathbf{1}(\mathbf{s}_{cr}^{t_k}, \bar{\mathbf{s}}_i) \mathbf{1}^{-\xi}(\mathbf{o}_{cr}^{t_k}, \bar{\mathbf{o}}_j) \tag{62}$$

Using $N_{\mathbf{s}_{cr}=\bar{\mathbf{s}}_i, \mathbf{o}_{cr}=\bar{\mathbf{o}}_j}^{+\xi}$ and $N_{\mathbf{s}_{cr}=\bar{\mathbf{s}}_i, \mathbf{o}_{cr}=\bar{\mathbf{o}}_j}^{-\xi}$, we can obtain the conservative replacement of the $i$-th original constraint, as in the previous subsection:

$$p^{+\xi}(\mathbf{o}_{cr} = \bar{\mathbf{o}}_j | \mathbf{s}_{cr} = \bar{\mathbf{s}}_i) := \frac{N_{\mathbf{s}_{cr}=\bar{\mathbf{s}}_i, \mathbf{o}_{cr}=\bar{\mathbf{o}}_j}^{+\xi}}{N_{\mathbf{s}_{cr}=\bar{\mathbf{s}}_i}}, \quad p^{-\xi}(\mathbf{o}_{cr} = \bar{\mathbf{o}}_j | \mathbf{s}_{cr} = \bar{\mathbf{s}}_i) := \frac{N_{\mathbf{s}_{cr}=\bar{\mathbf{s}}_i, \mathbf{o}_{cr}=\bar{\mathbf{o}}_j}^{-\xi}}{N_{\mathbf{s}_{cr}=\bar{\mathbf{s}}_i}} \tag{63}$$

$$p^{\xi}(\mathbf{s}_{cr} = \bar{\mathbf{s}}_i | \mathbf{o}_{cr} = \bar{\mathbf{o}}_j) := \frac{p^{+\xi}(\mathbf{o}_{cr} = \bar{\mathbf{o}}_j | \mathbf{s}_{cr} = \bar{\mathbf{s}}_i) r_{ep,i}}{\sum_{k=1}^{N_e} p^{-\xi}(\mathbf{o}_{cr} = \bar{\mathbf{o}}_j | \mathbf{s}_{cr} = \bar{\mathbf{s}}_k) r_{ep,k}} \tag{64}$$

$$\bar{c}_i^{\xi}(\mathbf{u}; \mathbf{o}, \mathbf{r}) := \max_{\substack{q_1, \dots, q_{N_e} \in \{0,1\} \\ \sum_k p^{\xi}(\mathbf{s}_{cr}=\bar{\mathbf{s}}_k | \mathbf{o}_{cr}=\mathbf{o}_{cr}^r) \cdot q_k \leq r_{t,i}}} \min_k \left(c_i(\mathbf{u}; \bar{\mathbf{s}}_k, \mathbf{r}) + M \cdot q_k\right) \tag{65}$$

Note that Equation 65 is the replacement of $c_i(\mathbf{u}; \bar{\mathbf{s}}_k, \mathbf{r})$ to obtain action based on output $\mathbf{o}$ rather than environment state $\mathbf{s}$, and thus, we use $c_i(\mathbf{u}; \bar{\mathbf{s}}_k, \mathbf{r})$ with a minimum function in Equation 65. Since the following is obvious:

$$\mathbf{1}^{-\xi}(\mathbf{o}_{cr}^{t_k}, \bar{\mathbf{o}}_j) \leq \mathbf{1}(\mathbf{o}_{cr}^{t_k}, \bar{\mathbf{o}}_j) \leq \mathbf{1}^{+\xi}(\mathbf{o}_{cr}^{t_k}, \bar{\mathbf{o}}_j) \tag{66}$$

it is straightforward that:

$$p^{\xi}(\mathbf{s}_{cr} = \bar{\mathbf{s}}_k | \mathbf{o}_{cr} = \mathbf{o}_{cr}^r) \geq p(\mathbf{s}_{cr} = \bar{\mathbf{s}}_k | \mathbf{o}_{cr} = \mathbf{o}_{cr}^r) \tag{67}$$

and

$$\bar{c}_i^{\xi}(\mathbf{u}; \mathbf{o}, \mathbf{r}) \leq \bar{c}_i(\mathbf{u}; \mathbf{o}, \mathbf{r}) \tag{68}$$

We can then characterize the additional condition beyond Condition 1 that guarantees the conservativeness of Equation 65 as follows:

**Condition 2.** For some $(\zeta, \xi)$,

1. $\Sigma_{\mathcal{Y}_{cr}}$ includes all open sets (based on $M_{\mathcal{Y}_{cr}}$).

2. For each $\bar{\mathbf{s}}_i$, the subsequence $(\mathbf{y}_{cr}^{t_{a_1}}, \ldots, \mathbf{y}_{cr}^{t_{a_{Nat}}})$ where $\{a_j\} = \{k | \mathbf{s}_{cr}^{t_k} = \bar{\mathbf{s}}_i\}$, that is, the subsequence consisting of sampled measurement results of environment state $\bar{\mathbf{s}}_i$ from the sequence $(\mathbf{y}_{cr}^{t_1}, \ldots, \mathbf{y}_{cr}^{t_{n_t}})$, is $\zeta$-informative in $(\mathcal{Y}_{cr}, M_{\mathcal{Y}_{cr}}, \Sigma_{\mathcal{Y}_{cr}}, P_{\bar{\mathbf{s}}_i})$.

3. For any $v_1, v_2 \in \mathcal{Y}_{cr}$, if $M_{\mathcal{Y}_{cr}}(v_1, v_2) < \zeta$, $f_{cr}(\mathbf{y}_{cr}; \mathbf{w}_{cr})$ satisfies $||f_{cr}(v_1; \mathbf{w}_{cr}) - f_{cr}(v_2; \mathbf{w}_{cr})|| \leq \xi$.

Condition 2 is about the required quality of internal test data and the required property of the safety classification model. The larger $\zeta$ is, the easier it is for sequences to be $\zeta$-informative. Large $\xi$ allows many neural networks and large $\zeta$ to satisfy the condition. However, a large $\xi$ makes our system much more conservative. In contrast, only high-quality samples can be $\zeta$-informative for small $\zeta$, and the information processing module must have low stiffness (*i.e.,* be smooth enough that a small $\zeta$-ball can pass through the model and become a small $\xi$-ball) to satisfy the condition with small $\xi$. However, a small $\xi$ allows less conservative actions.

Now, we can prove the conservativeness of Equation 65 based on the conditions.

**Theorem 1.** *Under Condition 1 and Condition 2,*

$$p^{-\xi}(\mathbf{o}_{cr} = \bar{\mathbf{o}}_j | \mathbf{s}_{cr} = \bar{\mathbf{s}}_i) \leq p^*(\mathbf{o}_{cr} = \bar{\mathbf{o}}_j | \mathbf{s}_{cr} = \bar{\mathbf{s}}_i) \leq p^{+\xi}(\mathbf{o}_{cr} = \bar{\mathbf{o}}_j | \mathbf{s}_{cr} = \bar{\mathbf{s}}_i). \tag{69}$$

*Proof.* For the first inequality, by definition, we obtain

$$p^{-\xi}(\mathbf{o}_{cr} = \bar{\mathbf{o}}_j | \mathbf{s}_{cr} = \bar{\mathbf{s}}_i) = \frac{|\{k | \mathbf{s}_{cr}^{t_k} = \bar{\mathbf{s}}_i, g_{cr}(\bar{B}(f_{cr}(\mathbf{y}_{cr}^{t_k}; \mathbf{w}_{cr}), \xi)) = \{\bar{\mathbf{o}}_j\}|}{N_{\mathbf{s}_{cr} = \bar{\mathbf{s}}_i}} \tag{70}$$

when $\bar{B}(\omega, \xi)$ denotes the closed ball centered on $\omega$ with radius $\xi$. Meanwhile, by the third statement of Condition 2, $g_{cr}(\bar{B}(f_{cr}(\mathbf{y}_{cr}^{t_k}; \mathbf{w}_{cr}), \xi)) = \{\bar{\mathbf{o}}_j\}$ implies $B(\mathbf{y}_{cr}^{t_k}, \zeta) \subset Y_{\bar{\mathbf{o}}_j}$. By substituting this in Equation 70 and using the second statement of Condition 2, we obtain:

$$\begin{aligned} p^{-\xi}(\mathbf{o}_{cr} = \bar{\mathbf{o}}_j | \mathbf{s}_{cr} = \bar{\mathbf{s}}_i) &= \frac{|\{k | \mathbf{s}_{cr}^{t_k} = \bar{\mathbf{s}}_i, g_{cr}(\bar{B}(f_{cr}(\mathbf{y}_{cr}^{t_k}; \mathbf{w}_{cr}), \xi)) = \{\bar{\mathbf{o}}_j\}|}{N_{\mathbf{s}_{cr} = \bar{\mathbf{s}}_i}} \\ &\leq \frac{|\{k | \mathbf{s}_{cr}^{t_k} = \bar{\mathbf{s}}_i, B(\mathbf{y}_{cr}^{t_k}, \zeta) \subset Y_{\bar{\mathbf{o}}_j}\}|}{N_{\mathbf{s}_{cr} = \bar{\mathbf{s}}_i}} \\ &\leq P_{\bar{\mathbf{s}}_i}\left( \bigcup_{\mathbf{s}_{cr}^{t_k} = \bar{\mathbf{s}}_i, B(\mathbf{y}_{cr}^{t_k}, \zeta) \subset Y_{\bar{\mathbf{o}}_j}} B(\mathbf{y}_{cr}^{t_k}, \zeta) \right) \\ &\leq P_{\bar{\mathbf{s}}_i}(Y_{\bar{\mathbf{o}}_j}) = p^*(\mathbf{o}_{cr} = \bar{\mathbf{o}}_j | \mathbf{s}_{cr} = \bar{\mathbf{s}}_i) \end{aligned} \tag{71}$$

For the second inequality, by definition, we obtain:

$$
\begin{aligned}
p^{+\xi}(\mathbf{o}_{cr} = \bar{\mathbf{o}}_j | \mathbf{s}_{cr} = \bar{\mathbf{s}}_i) &= \frac{|\{k | \mathbf{s}_{cr}^{t_k} = \bar{\mathbf{s}}_i, g_{cr}(\bar{B}(f_{cr}(\mathbf{y}_{cr}^{t_k}; \mathbf{w}_{cr}), \xi)) \supset \{\bar{\mathbf{o}}_j\}|}{N_{\mathbf{s}_{cr} = \bar{\mathbf{s}}_i}} \\
&= 1 - \frac{|\{k | \mathbf{s}_{cr}^{t_k} = \bar{\mathbf{s}}_i, g_{cr}(\bar{B}(f_{cr}(\mathbf{y}_{cr}^{t_k}; \mathbf{w}_{cr}), \xi)) \subset \{\bar{\mathbf{o}}_{-\mathbf{j}}\}|}{N_{\mathbf{s}_{cr} = \bar{\mathbf{s}}_i}}
\end{aligned}
\tag{72}
$$

when $\{\bar{\mathbf{o}}_{-\mathbf{j}}\}$ denotes $\{\bar{\mathbf{o}}_1, \ldots, \bar{\mathbf{o}}_{j-1}, \bar{\mathbf{o}}_{j+1}, \ldots, \bar{\mathbf{o}}_{N_o}\}$. Similar to the first inequality, we can prove the second inequality based on statements of Condition 2 as follows:

$$
\begin{aligned}
p^{+\xi}(\mathbf{o}_{cr} = \bar{\mathbf{o}}_j | \mathbf{s}_{cr} = \bar{\mathbf{s}}_i) &= 1 - \frac{|\{k | \mathbf{s}_{cr}^{t_k} = \bar{\mathbf{s}}_i, g_{cr}(\bar{B}(f_{cr}(\mathbf{y}_{cr}^{t_k}; \mathbf{w}_{cr}), \xi)) \subset \{\bar{\mathbf{o}}_{-\mathbf{j}}\}|}{N_{\mathbf{s}_{cr} = \bar{\mathbf{s}}_i}} \\
&\geq 1 - \frac{|\{k | \mathbf{s}_{cr}^{t_k} = \bar{\mathbf{s}}_i, B(\mathbf{y}_{cr}^{t_k}, \zeta) \subset \bigcup_{l \neq j} Y_{\bar{\mathbf{o}}_l}\}|}{N_{\mathbf{s}_{cr} = \bar{\mathbf{s}}_i}} \\
&\geq 1 - P_{\bar{\mathbf{s}}_i}\left( \bigcup_{\mathbf{s}_{cr}^{t_k} = \bar{\mathbf{s}}_i, B(\mathbf{y}_{cr}^{t_k}, \zeta) \subset \bigcup_{l \neq j} Y_{\bar{\mathbf{o}}_l}} B(\mathbf{y}_{cr}^{t_k}, \zeta) \right) \\
&\geq 1 - P_{\bar{\mathbf{s}}_i}\left( \bigcup_{l \neq j} Y_{\bar{\mathbf{o}}_l} \right) = P_{\bar{\mathbf{s}}_i}(Y_{\bar{\mathbf{o}}_j}) = p^*(\mathbf{o}_{cr} = \bar{\mathbf{o}}_j | \mathbf{s}_{cr} = \bar{\mathbf{s}}_i)
\end{aligned}
\tag{73}
$$

$\square$

Theorem 1 implies that we can use fixed internal test data within a safety classification model, and conservative testing allows it to restrain the overfitting problem. Internal test data—a fixed dataset separated from the training batch—can be used to design novel information processing module architectures or loss functions based on comparisons or other calculations involving both types of data (training batch and internal test data). We expect that this novel concept will be an ingredient in a variety of new developments related to computational systems.

Corollary 1 is straightforward from Theorem 1 and the definitions of the replacements of the constraints (Equations 65 and 30):

**Corollary 1.** Under Condition 1 and Condition 2,

$$
\bar{c}_i^{\xi}(\mathbf{u}; \mathbf{o}, \mathbf{r}) \leq \bar{c}_i^*(\mathbf{u}; \mathbf{o}, \mathbf{r}).
\tag{74}
$$

**Corollary 2.** When the optimization problem Equation 13 with conservative testing (where the constraints are replaced with $\bar{c}_i^{\xi}$) has a feasible solution, it outputs the optimal solution that satisfies the original chance-constraints (Equation 15). That is, the satisfaction of the user-provided safety constraints is guaranteed for the user-specified probability thresholds.

We now present the condition under which the conservative replacement of constraints results in a higher or equal loss. Since our loss function is defined separately from the optimization stage, there can be some cases where an action based on conservative testing results in a smaller loss. This will be especially natural when the loss is designed to encourage actions based on conservative testing. Definition 2 describes how the loss function aligns well with the objective of the optimization stage under a constraint.

**Definition 2.** For three functions $\pi, \chi, \psi : \mathbb{R}^{n_{real}} \times \mathbb{Z}^{n_{inte}} \to \mathbb{R}$, we say $\pi$ *aligns well* with $\chi$ under constraint $\psi \geq 0$ in set $\mathcal{S} \subset \mathbb{R}^{n_{real}} \times \mathbb{Z}^{n_{inte}}$ when the following statement holds:

For all $\alpha_1, \alpha_2 \in \mathcal{S}$, when both $\chi(\alpha_1) \leq \chi(\alpha_2)$ and $\psi(\alpha_1) < 0 \leq \psi(\alpha_2)$ hold, then $\pi(\alpha_1) \leq \pi(\alpha_2)$.

We can then configure the condition that guarantees a higher loss for the conservative testing technique.

**Condition 3.** As functions of $\mathbf{u}$, for any $c_i$ that needs to be replaced by the chance-constrained method and any $\bar{\mathbf{s}}_k \in \mathcal{S}_{cr}$, $L(\mathbf{u}, \mathbf{o}; \bar{\mathbf{s}}_k, \mathbf{s}_{ncr}, \mathbf{r})$ aligns well with $\bar{J}(\mathbf{u}; \mathbf{o}, \mathbf{r})$ under constraint $c_i(\mathbf{u}; \bar{\mathbf{s}}_k, \mathbf{r}) \geq 0$ in $\mathcal{U}$.

Since $\bar{J}$ is set to obtain high control performance and $L$ is set to evaluate the performance, it is natural to assume Condition 3. Note that Condition 3 is automatically satisfied when $L$ is a non-decreasing function of $\bar{J}$ or conversely. Now, we can prove that the conservative testing technique results in a loss greater than or equal to that obtained using the real performance value of the information processing module under some conditions.

**Proposition 3.** Under Conditions 1, 2, and 3, the minimum of $L$ with the optimal action[19] is higher or equal when the constraints are replaced with $\bar{c}_i^\xi$ (conservative testing) than when the constraints are replaced with $\bar{c}_i^*$ (control based on the real performance).

*Proof.* We prove by contradiction. Let $\mathbf{u}_\xi^*$ and $\mathbf{u}_*^*$ denote the optimal solutions under conservative testing (constraints $\bar{c}_i^\xi$) and real performance (constraints $\bar{c}_i^*$), respectively. Assume, for contradiction, that $L(\mathbf{u}_\xi^*, \mathbf{o}; \bar{\mathbf{s}}_k, \mathbf{s}_{ncr}, \mathbf{r}) < L(\mathbf{u}_*^*, \mathbf{o}; \bar{\mathbf{s}}_k, \mathbf{s}_{ncr}, \mathbf{r})$.

Since $\mathbf{u}_\xi^*$ is optimal under stricter constraints (by Corollary 1), we have:

$$\bar{J}(\mathbf{u}_\xi^*; \mathbf{o}, \mathbf{r}) \geq \bar{J}(\mathbf{u}_*^*; \mathbf{o}, \mathbf{r}) \tag{75}$$

By Condition 3, for $L(\mathbf{u}_\xi^*) < L(\mathbf{u}_*^*)$ to hold when $\bar{J}(\mathbf{u}_\xi^*) \geq \bar{J}(\mathbf{u}_*^*)$, we must have $c_i(\mathbf{u}_*^*; \bar{\mathbf{s}}_k, \mathbf{r}) \geq 0$ for every $c_i$ that satisfies $c_i(\mathbf{u}_\xi^*; \bar{\mathbf{s}}_k, \mathbf{r}) \geq 0$ and is needed to be replaced. However, by Corollary 1, the reverse of this also holds. Therefore, both $\mathbf{u}_\xi^*$ and $\mathbf{u}_*^*$ satisfy exactly the same set of constraints $c_i$.

Since both solutions are optimal for the same objective function $\bar{J}$ under identical constraints, they must achieve the same optimal value: $\bar{J}(\mathbf{u}_\xi^*) = \bar{J}(\mathbf{u}_*^*)$. When multiple solutions achieve the same optimal $\bar{J}$ value, our selection rule chooses the one minimizing $L$. However, we have $L(\mathbf{u}_\xi^*) < L(\mathbf{u}_*^*)$ by our assumption, which contradicts that both $\mathbf{u}_\xi^*$ and $\mathbf{u}_*^*$ are optimal solutions.

$\square$

To conclude the theoretical analysis of conservative testing and chance-constraints, we finally show that this technique can also be incorporated into the general framework presented in Section A.

**Proposition 4.** When both $\mathcal{S}_{cr}$ and $\mathcal{O}_{cr}$ are *finite*, our conservative testing technique for chance-constrained formulation (Equations 63, 64, and 65) is included in our general framework presented in Section A. Moreover, replacements of constraints are continuous with respect to the continuous part of $\mathbf{u}$ and $\mathbf{o}$ provided that the original constraints to be replaced are continuous with respect to the continuous part of $\mathbf{u}$.

*Proof.* The part for obtaining $\mathbf{1}^{+\xi}(f_{cr}(\mathbf{y}_{cr}^{t_k}; \mathbf{w}_{cr}), \bar{\mathbf{o}}_j))$ or $\mathbf{1}^{-\xi}(f_{cr}(\mathbf{y}_{cr}^{t_k}; \mathbf{w}_{cr}), \bar{\mathbf{o}}_j))$ can be treated as the change of the last layer of the safety classification model. The remaining steps follow the same approach as in the proof of Proposition 1. $\square$

---

[19]If there are multiple optimal actions, we choose the one with the minimum $L$ among them.

## D  CONSTRUCTION OF THE LOSS FUNCTION FOR TRAINING

We need a *loss function* to train the information processing module in our framework. First, we denote it as $L(\mathbf{u}, \mathbf{o}; \mathbf{s}, \mathbf{r})$. We define our loss function as a general function of $\mathbf{u}, \mathbf{o}, \mathbf{s}$, and $\mathbf{r}$ that can include any kind of functions regarding them. While $J(\mathbf{u}; \mathbf{s}, \mathbf{r})$ is commonly used as $L(\mathbf{u}, \mathbf{o}; \mathbf{s}, \mathbf{r})$, our framework permits any function of $\mathbf{u}, \mathbf{o}, \mathbf{s}$, and $\mathbf{r}$. For example, it can include the objective of the optimization stage $\bar{J}$ or traditional loss functions based on $\mathbf{o}$ and $\mathbf{s}$ that are used for perception, such as cross-entropy loss or root mean square loss. This can be useful when the user of our framework needs to explicitly improve the accuracy of the information processing module output for reasons other than the system performance.

We need to obtain the gradient of our loss function to use it for training. However, since the loss function is a function of $\mathbf{u}$, the loss function with the actions $L(\mathbf{u}^*(\mathbf{o}, \mathbf{r}), \mathbf{o}; \mathbf{s}, \mathbf{r})$ depends on the optimal solutions of the optimization stage $\mathbf{u}^*(\mathbf{o}, \mathbf{r})$ that are not generally continuous functions of information processing module output $\mathbf{o}$. This typically causes the gradient to diverge or disappear (See (Vlastelica et al., 2020) for further description). Moreover, when there are multiple optimal solutions in the optimization stage, the corresponding values of the loss function can be different. One fatal problem with this is that the loss cannot be well-defined as a function of only $\mathbf{o}, \mathbf{s}$, and $\mathbf{r}$. Even though we construct a well-defined function that has a minimum value of the loss function among $\mathbf{u}$, which is an optimal solution of the optimization stage, a similar problem remains because it requires solving a two-stage minimization (*i.e.,* minimization in the $\arg\min$ set of another problem) that is computationally hard. Since we deal with a general optimization problem, we cannot guarantee that a closed-form solution to the optimal set of the problem exists. To our knowledge, no literature explicitly addresses the possibility of multiple optimal solutions for the optimization stage.

As an alternative to the loss function $L(\mathbf{u}^*(\mathbf{o}, \mathbf{r}), \mathbf{o}; \mathbf{s}, \mathbf{r})$ that depends on actions, we use the *approximated general loss function* $\tilde{L}$, which is well-defined and continuous with respect to $\mathbf{o}$ and satisfies $\tilde{L} \simeq L(\mathbf{u}^*(\mathbf{o}, \mathbf{r}), \mathbf{o}; \mathbf{s}, \mathbf{r})$ with an action chosen by the optimization stage. Note that $\tilde{L}$ need not be expressible in closed form and may involve some optimization problems. The only requirement is that it be continuous with respect to the continuous part of $\mathbf{o}$ and final intermediate results just before the discrete part of $\mathbf{o}$.

To avoid two-stage minimization—where we minimize the loss function within the optimal solution set of the optimization stage—we combine both stages into one. This is necessary because we assume no specific relationship between the optimization problem and loss function, and the first-stage minimization may not even have a closed-form solution. Before combining them later, we should solve the continuity issue. To construct an approximate loss function that is continuous, we utilize the property that the optimal objective of an optimization problem is continuous with respect to parameters when the objective function is continuous with respect to the parameters and the constraints are independent of the parameters. Thus, converting the optimization stage to a virtually unconstrained optimization problem (constraints do not have any meaningful effect on the solution) can be a good approach for the first stage of constructing such an approximated loss function. This ensures the continuity with respect to output $\mathbf{o}$ before combining it with the loss function. As a result, we transform the optimization problem into a virtually unconstrained one and prove that this transformation is valid.

We convert the problem to an optimization problem with a compact feasible region that does not depend on any parameters by merging constraints with coefficient $\boldsymbol{\beta} > 0$ into the objective. We need the compact feasible region to prevent $\bar{J}$ from diverging to $-\infty$. We can choose region $\mathcal{U}$ with simple constraints such as $||\mathbf{u}|| \leq M$ with a large constant $M$. Note that we can use any norm paired with other mathematical concepts, such as compactness and continuity. The resulting converted optimization problem can be written as Equation 76:

$$\min_{\mathbf{u} \in \mathcal{U}} \bar{J}(\mathbf{u}; \mathbf{o}, \mathbf{r}) - \boldsymbol{\beta}^{\top} \min(\bar{\mathbf{c}}(\mathbf{u}; \mathbf{o}, \mathbf{r}), \mathbf{0}) \tag{76}$$

This conversion cannot guarantee equivalence between the original and converted optimization problems. The constraints may be nearly stationary (*e.g.,* at a local maximum or minimum point) at the boundary of the feasible region, which can lead to a solution that is infeasible in the original optimization problem, even when $\beta_i$ is extremely large for all $i$.

Instead, since the output of the optimization layer is the optimal solution $\mathbf{u}$, we check the distance between the solutions of these two optimization problems. Considering that the loss function or real performance mainly depends on actions rather than the optimal value in the optimization stage, the conversion is valid if and only if it leads to similar actions (optimal solutions). We aim to make the conversion have solutions sufficiently close to the original solutions under the given $\mathcal{U}$ by choosing appropriate $\beta_i$ for all $i$.

To guarantee this, we need both $\bar{J}(\mathbf{u}; \mathbf{o}, \mathbf{r})$ and $\bar{c}_i(\mathbf{u}; \mathbf{o}, \mathbf{r})$ for all $i$ to be continuous with respect to $\mathbf{x}$. Moreover, in Equation 76, $\mathcal{U}$ should be a subset of $\mathcal{X} \times \mathcal{Z}^{20}$ that only consists of a finite number of $\mathbf{z}$ with paired compact subsets ($\mathcal{X}_i \subset \mathcal{X}$ for each $z_i$) of $\mathcal{X}$, and $\bigcup_i \mathcal{X}_i \times \{z_i\}$ includes the region of interest. Our assumptions can be summarized as Assumption 1. Note that from now on, we denote the continuous part and the discrete part of $\mathbf{u}$ as $\mathbf{x}$ and $\mathbf{z}$, respectively.

**Assumption 1.** $\bar{J}(\mathbf{u}; \mathbf{o}, \mathbf{r})$, $\bar{c}_i(\mathbf{u}; \mathbf{o}, \mathbf{r})$, and $\mathcal{U}$ satisfy the following statements.

1. $\bar{J}(\mathbf{u}; \mathbf{o}, \mathbf{r})$ is continuous with respect to $\mathbf{x}$.

2. For all $i$, $\bar{c}_i(\mathbf{u}; \mathbf{o}, \mathbf{r})$ is continuous with respect to $\mathbf{x}$.

3. $\mathcal{U}$ is a finite union of multiplications of different elements of $\mathcal{Z}$ and paired compact subsets of $\mathcal{X}$. That is, $\mathcal{U}$ can be written as $\bigcup_1^{m_u(<\infty)} \mathcal{X}_i \times \{\mathbf{z}_i\}$, $\quad \forall i, \mathcal{X}_i(\subset \mathcal{X})$ is compact, $\forall i, j, \mathbf{z}_i \neq \mathbf{z}_j$.

4. $\mathcal{U}$ includes all optimal solutions of Equation 13.

For notational convenience, we denote the optimal solution space of the non-converted optimization problem as:

$$S(\mathbf{o}, \mathbf{r}) := \arg\min_{\mathbf{u}} \ \bar{J}(\mathbf{u}; \mathbf{o}, \mathbf{r}) \quad \text{subject to} \quad \bar{\mathbf{c}}(\mathbf{u}; \mathbf{o}, \mathbf{r}) \geq \mathbf{0} \tag{77}$$

and the optimal solution space of $\mathbf{x}$ paired with a specific $\mathbf{z}$ as:

$$S_{\mathbf{z}}(\mathbf{o}, \mathbf{r}) := \{\mathbf{x} : \mathbf{u} \in S(\mathbf{o}, \mathbf{r})\} \tag{78}$$

Then, Proposition 5 states that we can arbitrarily reduce the distance between solutions of the original optimization problem and the converted problem by choosing sufficiently large $\boldsymbol{\beta}$ under given $\mathbf{o}$ and $\mathbf{r}$.

**Proposition 5.** When $\bar{J}(\mathbf{u}; \mathbf{o}, \mathbf{r})$, $\bar{c}_i(\mathbf{u}; \mathbf{o}, \mathbf{r})$, and $\mathcal{U}$ satisfy Assumption 1, for any $\epsilon_1 > 0$, there exists $\underline{\boldsymbol{\beta}}(\mathbf{o}, \mathbf{r}, \epsilon_1) > \mathbf{0}$ such that any $\boldsymbol{\beta} > \underline{\boldsymbol{\beta}}$ makes all solutions of the converted problem lies within distance $\epsilon_1$ from a solution of the original problem with the same $\mathbf{z}$. Specifically,

$$\forall \boldsymbol{\beta} > \underline{\boldsymbol{\beta}}, \quad \forall(\tilde{\mathbf{x}}, \mathbf{z}) \in \arg\min_{\mathbf{u} \in \mathcal{U}} \left\{ \bar{J}(\mathbf{u}; \mathbf{o}, \mathbf{r}) - \boldsymbol{\beta}^\top \min(\bar{\mathbf{c}}(\mathbf{u}; \mathbf{o}, \mathbf{r}), \mathbf{0}) \right\},$$
$$\exists \mathbf{x}^* \in S_{\mathbf{z}}(\mathbf{o}, \mathbf{r}) \text{ such that } \|\mathbf{x}^* - \tilde{\mathbf{x}}\| < \epsilon_1 \tag{79}$$

*Proof.* For all $\mathbf{z}$, we denote the union of $\epsilon_1$-balls[21] centered on elements of $S_{\mathbf{z}}(\mathbf{o}, \mathbf{r})$ as:

$$\tilde{S}_{\mathbf{z}}(\mathbf{o}, \mathbf{r}, \epsilon_1) = \bigcup_{\mathbf{x} \in S_{\mathbf{z}}(\mathbf{o}, \mathbf{r})} B(\mathbf{x}, \epsilon_1) \tag{80}$$

For each $\mathbf{z}$, $\tilde{S}_{\mathbf{z}}(\mathbf{o}, \mathbf{r}, \epsilon_1)$ is open because it is a union of open balls. We also define:

$$\tilde{S}(\mathbf{o}, \mathbf{r}, \epsilon_1) = \bigcup_{\mathbf{z}} \tilde{S}_{\mathbf{z}}(\mathbf{o}, \mathbf{r}, \epsilon_1) \times \{\mathbf{z}\} \tag{81}$$

---

[20] $\mathcal{X}$ and $\mathcal{Z}$ corresponds to the $\mathbb{R}$ and $\mathbb{Z}$ part, respectively.

[21] '$a$-ball' indicates an open ball.

and let:

$$\mathcal{U}_{\mathbf{z}} := \mathcal{U} \cap (\mathcal{X} \times \{\mathbf{z}\}) \tag{82}$$

Then, it is straightforward that $\mathcal{U}_{\mathbf{z}} \setminus \tilde{S}(\mathbf{o}, \mathbf{r}, \epsilon_1) = \mathcal{U}_{\mathbf{z}} \setminus \tilde{S}_{\mathbf{z}}(\mathbf{o}, \mathbf{r}, \epsilon_1)$ is compact.

We denote the optimal value of the non-converted optimization problem as $\bar{J}^*(\mathbf{o}; \mathbf{r})$ and the set of points that lead to a smaller or equal objective than $\bar{J}^*(\mathbf{o}; \mathbf{r})$ as:

$$H(\mathbf{o}, \mathbf{r}) = \{\mathbf{u} \in \mathcal{U} | \bar{J}(\mathbf{u}; \mathbf{o}, \mathbf{r}) \le \bar{J}^*(\mathbf{o}; \mathbf{r})\} \tag{83}$$

Note that $\bar{J}^*$ is the optimal value under constraints, while $\bar{J}$ itself does not consider constraints. All solutions of the converted optimization problem are included in $H(\mathbf{o}, \mathbf{r})$ regardless of $\boldsymbol{\beta}$. Otherwise, if some $\mathbf{u}$ with $\bar{J}(\mathbf{u}) > \bar{J}^*$ were optimal in the converted problem, then any feasible $\mathbf{u}^*$ achieving $\bar{J}^*$ would have a strictly better converted objective value since $\bar{J}(\mathbf{u}^*) - \boldsymbol{\beta}^\top \min(\bar{\mathbf{c}}(\mathbf{u}^*), \mathbf{0}) = \bar{J}(\mathbf{u}^*) = \bar{J}^* < \bar{J}(\mathbf{u})$, yielding a contradiction.

Since $\bar{J}$ is continuous, ensuring that small perturbations in the input produce only small changes in the function value, $H(\mathbf{o}, \mathbf{r})$ is closed as the inverse image of a closed set under this continuous mapping. Thus, $\mathcal{U}_{\mathbf{z}} \cap H(\mathbf{o}, \mathbf{r})$ is an intersection of a closed set with a compact set and is thus compact. As a result, $(\mathcal{U}_{\mathbf{z}} \cap H(\mathbf{o}, \mathbf{r})) \setminus \tilde{S}(\mathbf{o}, \mathbf{r}, \epsilon_1)$ is compact for any $\mathbf{z}$.

Considering that all feasible solutions (satisfying constraints of the original problem) that are included in $H(\mathbf{o}, \mathbf{r})$ are optimal, they are elements of $S(\mathbf{o}, \mathbf{r})$. It implies that all elements of $(\mathcal{U}_{\mathbf{z}} \cap H(\mathbf{o}, \mathbf{r})) \setminus \tilde{S}(\mathbf{o}, \mathbf{r}, \epsilon_1)$ make at least one of the constraints violated. Then, we can construct a function $\phi(\mathbf{u}; \mathbf{o}, \mathbf{r})$ as:

$$\phi(\mathbf{u}; \mathbf{o}, \mathbf{r}) : (\mathcal{U}_{\mathbf{z}} \cap H(\mathbf{o}, \mathbf{r})) \setminus \tilde{S}(\mathbf{o}, \mathbf{r}, \epsilon_1) \to \mathbb{R}^+, \quad \phi(\mathbf{u}; \mathbf{o}, \mathbf{r}) = \frac{\bar{J}(\mathbf{u}; \mathbf{o}, \mathbf{r}) - \bar{J}^*(\mathbf{o}; \mathbf{r})}{\min_i \bar{c}_i(\mathbf{u}; \mathbf{o}, \mathbf{r})} \tag{84}$$

Note that $\phi(\mathbf{u}; \mathbf{o}, \mathbf{r})$ is not a function of $\epsilon_1$, and since its domain does not include any optimal solutions, $\phi(\mathbf{u}; \mathbf{o}, \mathbf{r}) \ne 0$. Since $\phi(\mathbf{u}; \mathbf{o}, \mathbf{r})$ is the ratio of two continuous functions and thus is continuous, we can choose the maximum value of $\phi(\mathbf{u}; \mathbf{o}, \mathbf{r})$ due to the compactness of $(\mathcal{U}_{\mathbf{z}} \cap H(\mathbf{o}, \mathbf{r})) \setminus \tilde{S}(\mathbf{o}, \mathbf{r}, \epsilon_1)$. Let us denote the maximum value of $\phi(\mathbf{u}; \mathbf{o}, \mathbf{r})$ with respect to $\mathbf{x}$ as $\mu(\mathbf{z}; \mathbf{o}, \mathbf{r}, \epsilon_1)$ and define $\underline{\boldsymbol{\beta}}$ as:

$$\underline{\boldsymbol{\beta}}(\mathbf{o}, \mathbf{r}, \epsilon_1) := \max_z \mu(\mathbf{z}; \mathbf{o}, \mathbf{r}, \epsilon_1) \cdot \mathbf{1} \tag{85}$$

Then $\forall \boldsymbol{\beta} > \underline{\boldsymbol{\beta}}$ and $\forall \mathbf{z} \in \mathcal{Z}$, for any elements of $(\mathcal{U}_{\mathbf{z}} \cap H(\mathbf{o}, \mathbf{r})) \setminus \tilde{S}(\mathbf{o}, \mathbf{r}, \epsilon_1)$, we have:

$$
\begin{aligned}
\bar{J}(\mathbf{u}; \mathbf{o}, \mathbf{r}) &- \boldsymbol{\beta}^\top \min(\bar{\mathbf{c}}(\mathbf{u}; \mathbf{o}, \mathbf{r}), \mathbf{0}) \\
&> \bar{J}(\mathbf{u}; \mathbf{o}, \mathbf{r}) - \max_{\mathbf{z}'} \mu(\mathbf{z}'; \mathbf{o}, \mathbf{r}, \epsilon_1) \mathbf{1}^\top \min(\bar{\mathbf{c}}(\mathbf{u}; \mathbf{o}, \mathbf{r}), \mathbf{0}) \\
&\ge \bar{J}(\mathbf{u}; \mathbf{o}, \mathbf{r}) - \mu(\mathbf{z}; \mathbf{o}, \mathbf{r}, \epsilon_1) \mathbf{1}^\top \min(\bar{\mathbf{c}}(\mathbf{u}; \mathbf{o}, \mathbf{r}), \mathbf{0}) \\
&\ge \bar{J}(\mathbf{u}; \mathbf{o}, \mathbf{r}) - \frac{\bar{J}(\mathbf{u}; \mathbf{o}, \mathbf{r}) - \bar{J}^*(\mathbf{o}, \mathbf{r})}{\min_i \bar{c}_i(\mathbf{u}; \mathbf{o}, \mathbf{r})} \mathbf{1}^\top \min(\bar{\mathbf{c}}(\mathbf{u}; \mathbf{o}, \mathbf{r}), \mathbf{0}) \\
&= \bar{J}(\mathbf{u}; \mathbf{o}, \mathbf{r}) - (\bar{J}(\mathbf{u}; \mathbf{o}, \mathbf{r}) - \bar{J}^*(\mathbf{o}, \mathbf{r})) \sum_j \frac{\min(\bar{c}_j(\mathbf{u}; \mathbf{o}, \mathbf{r}), 0)}{\min_i \bar{c}_i(\mathbf{u}; \mathbf{o}, \mathbf{r})} \\
&\ge \bar{J}^*(\mathbf{o}, \mathbf{r})
\end{aligned}
\tag{86}
$$

Note that $\boldsymbol{\beta}^\top \min(\bar{\mathbf{c}}(\mathbf{u}; \mathbf{o}, \mathbf{r}), \mathbf{0})$ is always non-positive. The first inequality is based on the definition of $\underline{\boldsymbol{\beta}}$. The third inequality is based on the definition of $\mu$. The last inequality holds because $\bar{J}(\mathbf{u}; \mathbf{o}, \mathbf{r}) - \bar{J}^*(\mathbf{o}; \mathbf{r}) \le 0$ in $H(\mathbf{o}, \mathbf{r})$ and $\min_i \bar{c}_i(\mathbf{u}; \mathbf{o}, \mathbf{r}) < 0$ due to the definition of $H(\mathbf{o}, \mathbf{r})$ and the optimality of $\bar{J}^*(\mathbf{o}; \mathbf{r})$.

Therefore, all elements of $(\mathcal{U}_{\mathbf{z}} \cap H(\mathbf{o}, \mathbf{r})) \setminus \tilde{S}(\mathbf{o}, \mathbf{r}, \epsilon_1)$ cannot be a solution of the converted optimization problem. Considering that all solutions of the converted optimization problem are included in $H(\mathbf{o}, \mathbf{r})$, all solutions of the converted optimization problem are included in $\tilde{S}(\mathbf{o}, \mathbf{r}, \epsilon_1)$ and this implies Equation 79. $\qquad \square$

This conversion allows us to use the continuity of the optimal objective, provided that the objective function is continuous for both the variables and parameters. Now, similar to (Vlastelica et al., 2020), we can construct the general approximate loss function $\tilde{L}(\mathbf{o}; \mathbf{s}, \mathbf{r}, \boldsymbol{\beta}, \lambda)$ with the converted optimization problem as follows.

$$\tilde{L}(\mathbf{o}; \mathbf{s}, \mathbf{r}, \boldsymbol{\beta}, \lambda) = \frac{1}{\lambda} \left( \min_{\mathbf{u} \in \mathcal{U}} \left( \lambda L(\mathbf{u}, \mathbf{o}; \mathbf{s}, \mathbf{r}) + \bar{J}(\mathbf{u}; \mathbf{o}, \mathbf{r}) - \boldsymbol{\beta}^\top \min(\bar{\mathbf{c}}(\mathbf{u}; \mathbf{o}, \mathbf{r}), \mathbf{0}) \right) \right.$$
$$\left. - \min_{\mathbf{u} \in \mathcal{U}} \left( \bar{J}(\mathbf{u}; \mathbf{o}, \mathbf{r}) - \boldsymbol{\beta}^\top \min(\bar{\mathbf{c}}(\mathbf{u}; \mathbf{o}, \mathbf{r}), \mathbf{0}) \right) \right) \tag{87}$$

We need to assume that the objective, the constraints, and the loss are continuous with respect to the variable $\mathbf{x}$ and the parameters conveyed by the information processing module $\mathbf{o}$. Assumption 2 covers statements we need to assume but are not covered in Assumption 1.

**Assumption 2.** $\bar{J}(\mathbf{u}; \mathbf{o}, \mathbf{r})$, $\bar{\mathbf{c}}(\mathbf{u}; \mathbf{o}, \mathbf{r})$, and $L(\mathbf{u}, \mathbf{o}; \mathbf{s}, \mathbf{r})$ satisfy the following statements:

1. $\bar{J}(\mathbf{u}; \mathbf{o}, \mathbf{r})$ is continuous with respect to $\mathbf{o}$.

2. For all $i$, $\bar{c}_i(\mathbf{u}; \mathbf{o}, \mathbf{r})$ is continuous with respect to $\mathbf{o}$.

3. $L(\mathbf{u}, \mathbf{o}; \mathbf{s}, \mathbf{r})$ is continuous with respect to $\mathbf{x}$ and $\mathbf{o}$.

4. $\mathcal{U}$ includes all optimal solutions of the following equation (identical with Equation 93) for all $\lambda$:

$$\min_{\mathbf{u} \in \mathcal{U}} \qquad \lambda L(\mathbf{u}, \mathbf{o}; \mathbf{s}, \mathbf{r}) + \bar{J}(\mathbf{u}; \mathbf{o}, \mathbf{r}) \tag{88a}$$
$$\text{subject to} \qquad \bar{c}_i(\mathbf{u}; \mathbf{o}, \mathbf{r}) \geq 0, \quad i = 1, \ldots, \bar{n}_c \tag{88b}$$

**Remark 1.** The third statement concerns the continuity of $L(\mathbf{u}, \mathbf{o}; \mathbf{s}, \mathbf{r})$ with respect to $\mathbf{x}$ and $\mathbf{o}$ as a function of $(\mathbf{u}, \mathbf{o})$. This is different from the continuity of $L(\mathbf{u}^*(\mathbf{o}, \mathbf{r}), \mathbf{o}; \mathbf{s}, \mathbf{r})$ with respect to $\mathbf{o}$ as a function of $\mathbf{o}$, which does not generally hold since obtaining $\mathbf{u}^*$ may not be continuous.

Under Assumption 2, we can establish the desired properties of our approximated loss (continuous with respect to $\mathbf{o}$ and convergence to the real loss) in the following theorem. Before the proof, we define $L^*(\mathbf{o}; \mathbf{s}, \mathbf{r})$ as below:

$$L^*(\mathbf{o}; \mathbf{s}, \mathbf{r}) := \min_{\mathbf{u} \in S(\mathbf{o}, \mathbf{r})} L(\mathbf{u}, \mathbf{o}; \mathbf{s}, \mathbf{r}) \tag{89}$$

The minimum of $L$ is well-defined under Assumption 2 because $S_{\mathbf{z}}(\mathbf{o}, \mathbf{r})$ is compact[22] and $L$ is continuous with respect to $\mathbf{x}$. As noted earlier, direct calculation of Equation 89 is computationally hard. Instead, we prove that our approximated loss function $\tilde{L}(\mathbf{o}; \mathbf{s}, \mathbf{r}, \boldsymbol{\beta}, \lambda)$ approaches to $L^*(\mathbf{o}; \mathbf{s}, \mathbf{r})$ when $\boldsymbol{\beta}$ becomes sufficiently large and $\lambda$ becomes sufficiently small.

**Theorem 2.** When $\bar{J}(\mathbf{u}; \mathbf{o}, \mathbf{r}), \bar{\mathbf{c}}(\mathbf{u}; \mathbf{o}, \mathbf{r}), \mathcal{U}$, and $L(\mathbf{u}, \mathbf{o}; \mathbf{s}, \mathbf{r})$ satisfy Assumptions 1 and 2, for any $(\mathbf{o}, \mathbf{s}, \mathbf{r})$ and $\epsilon_2 > 0$, the following two properties hold:

1. $\tilde{L}(\mathbf{o}; \mathbf{s}, \mathbf{r}, \boldsymbol{\beta}, \lambda)$ is continuous with respect to the continuous part of $\mathbf{o}$ for any $\boldsymbol{\beta} > \mathbf{0}$ and $\lambda > 0$.

2. There exist $\lambda_0(\mathbf{o}, \mathbf{s}, \mathbf{r}, \epsilon_2)$ and $\boldsymbol{\beta}_0(\mathbf{o}, \mathbf{s}, \mathbf{r}, \epsilon_2, \lambda)$ such that for any $\lambda < \lambda_0$ and $\boldsymbol{\beta} > \boldsymbol{\beta}_0$, we have:

$$|\tilde{L}(\mathbf{o}; \mathbf{s}, \mathbf{r}, \boldsymbol{\beta}, \lambda) - L^*(\mathbf{o}; \mathbf{s}, \mathbf{r})| < \epsilon_2 \tag{90}$$

---

[22] $S_{\mathbf{z}}(\mathbf{o}, \mathbf{r})$ is the intersection of closed sets (since each set satisfying a constraint or optimality condition is closed, being the inverse image of a closed set) and is contained in the compact set $\mathcal{U}$, therefore it is compact.

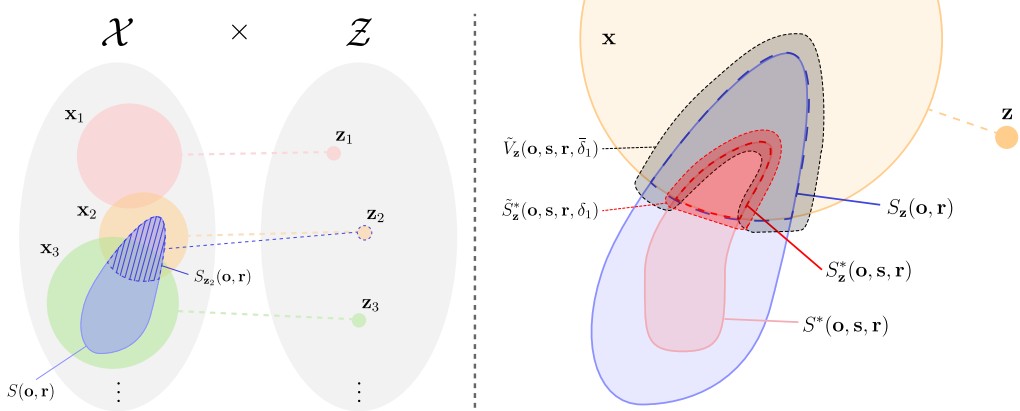

Figure 4: **Conceptual illustration for Theorem 2.**

*Proof.* At first, for any $\boldsymbol{\beta} > \mathbf{0}$ and $\lambda > 0$, both:

$$\min_{\mathbf{u} \in \mathcal{U}} \left( \lambda L(\mathbf{u}, \mathbf{o}; \mathbf{s}, \mathbf{r}) + \bar{J}(\mathbf{u}; \mathbf{o}, \mathbf{r}) + \boldsymbol{\beta}^\top \min(\bar{\mathbf{c}}(\mathbf{u}; \mathbf{o}, \mathbf{r}), \mathbf{0}) \right) \tag{91}$$

and

$$\min_{\mathbf{u} \in \mathcal{U}} \left( \bar{J}(\mathbf{u}; \mathbf{o}, \mathbf{r}) + \boldsymbol{\beta}^\top \min(\bar{\mathbf{c}}(\mathbf{u}; \mathbf{o}, \mathbf{r}), \mathbf{0}) \right) \tag{92}$$

are continuous with respect to the continuous part of $\mathbf{o}$ since they are the minimum of a continuous function with respect to $\mathbf{x}$, $\mathbf{z}$, and the continuous part of $\mathbf{o}$ for a fixed compact set. Thus, $\tilde{L}(\mathbf{o}; \mathbf{s}, \mathbf{r}, \boldsymbol{\beta}, \lambda)$ is continuous with respect to the continuous part of $\mathbf{o}$.

We define a *preferred optimal solution* of the optimization stage Equation 13 as an optimal solution of Equation 89, and denote the set of preferred optimal solutions as $S^*(\mathbf{o}, \mathbf{s}, \mathbf{r})$. Moreover, we can define an *integrated optimization layer* as Equation 93:

$$\min_{\mathbf{u} \in \mathcal{U}} \qquad \lambda L(\mathbf{u}, \mathbf{o}; \mathbf{s}, \mathbf{r}) + \bar{J}(\mathbf{u}; \mathbf{o}, \mathbf{r}) \tag{93a}$$

$$\text{subject to} \qquad \bar{c}_i(\mathbf{u}; \mathbf{o}, \mathbf{r}) \geq 0, \quad i = 1, \ldots, \bar{n}_c \tag{93b}$$

The proof of the second statement consists of two steps. The first step establishes that when $\lambda$ is sufficiently small, the minimum of $\bar{J} + \lambda L$ approaches the minimum of $\bar{J}$ plus the minimum of $L$ restricted to the optimal solution set of $\bar{J}$. Specifically, for any $\epsilon_2 > 0$, we find $\lambda_0(\mathbf{o}, \mathbf{s}, \mathbf{r}, \epsilon_2) > 0$ such that

$$\left| \lambda L(\mathbf{u}^*, \mathbf{o}; \mathbf{s}, \mathbf{r}) + \bar{J}(\mathbf{x}^*, \mathbf{z}^*; \mathbf{o}, \mathbf{r}) - \lambda L^*(\mathbf{o}; \mathbf{s}, \mathbf{r}) - \bar{J}^*(\mathbf{o}; \mathbf{r}) \right| < \frac{\lambda \epsilon_2}{3} \tag{94}$$

holds for all $\lambda < \lambda_0(\mathbf{o}, \mathbf{s}, \mathbf{r}, \epsilon_2)$ where $\bar{J}^*(\mathbf{o}; \mathbf{r})$ is the optimal value of the original optimization stage (Equation 13), $(\mathbf{x}^*, \mathbf{z}^*)$ is an optimal solution of the integrated optimization layer (Equation 93), and $L^*(\mathbf{o}; \mathbf{s}, \mathbf{r})$ is the minimum value of $L$ over the optimal solution set that minimizes $\bar{J}$. The second step addresses the constraint handling through penalty parameters. If constraints were absent, the problem would reduce to Equation 93a without the penalty terms, which can be solved by subtracting $\bar{J}^*$ and dividing by $\lambda$. We find $\boldsymbol{\beta}_0(\mathbf{o}, \mathbf{r}, \lambda, \mathbf{s}, \epsilon_2)$ such that the minimum values of the first and second minimization problems in Equation 87 are within distance $\lambda \epsilon_2 / 3$ of the minimum values of Equations 93 and 13, respectively.

**Step 1.** Since $L$ is continuous with respect to $\mathbf{x}$ and $\mathcal{U}_{\mathbf{z}}$ is compact for all $\mathbf{z}$, the function $L$ is uniformly continuous on $\mathcal{U}_{\mathbf{z}}$ for each $\mathbf{z}$. Therefore, we can choose $\delta_1(\mathbf{o}, \mathbf{s}, \mathbf{r}, \epsilon_2) > 0$ such that for all $\mathbf{x}_1, \mathbf{x}_2$ and $\mathbf{z}$, if $\|\mathbf{x}_1 - \mathbf{x}_2\| < \delta_1$, then

$$\left| L(\mathbf{x}_1, \mathbf{z}, \mathbf{o}; \mathbf{s}, \mathbf{r}) - L(\mathbf{x}_2, \mathbf{z}, \mathbf{o}; \mathbf{s}, \mathbf{r}) \right| < \frac{\epsilon_2}{3} \tag{95}$$

We define $S_{\mathbf{z}}^*(\mathbf{o}, \mathbf{s}, \mathbf{r})$ as:

$$S_{\mathbf{z}}^*(\mathbf{o}, \mathbf{s}, \mathbf{r}) := \{\mathbf{x} : \mathbf{u} \in S^*(\mathbf{o}, \mathbf{s}, \mathbf{r})\} \tag{96}$$

We fix a discrete variable $\mathbf{z}$ and denote $\tilde{S}_{\mathbf{z}}^*(\mathbf{o}, \mathbf{s}, \mathbf{r}, \delta_1)$ as the union of $\delta_1$-balls centered on elements of $S_{\mathbf{z}}^*(\mathbf{o}, \mathbf{s}, \mathbf{r})$. Then, by the same reason as explained in Proposition 5, $\mathcal{U}_{\mathbf{z}} \setminus (\tilde{S}_{\mathbf{z}}^*(\mathbf{o}, \mathbf{s}, \mathbf{r}, \delta_1) \times \{\mathbf{z}\})$ is compact. Since $L$ is continuous with respect to $\mathbf{x}$, we can find the minimum value of $L$ in the intersection of the two compact sets since $S(\mathbf{o}, \mathbf{r})$ is compact:

$$S_{\mathbf{z}}(\mathbf{o}, \mathbf{r}) \setminus (\tilde{S}_{\mathbf{z}}^*(\mathbf{o}, \mathbf{s}, \mathbf{r}, \delta_1) \times \{\mathbf{z}\}) = (\mathcal{U}_{\mathbf{z}} \cap S(\mathbf{o}, \mathbf{r})) \setminus (\tilde{S}_{\mathbf{z}}^*(\mathbf{o}, \mathbf{s}, \mathbf{r}, \delta_1) \times \{\mathbf{z}\}) \tag{97}$$

Let this minimum value as $l_{\mathbf{z}}(\mathbf{o}, \mathbf{s}, \mathbf{r}, \delta_1)$, and define $l(\mathbf{o}, \mathbf{s}, \mathbf{r}, \delta_1)$ as $l(\mathbf{o}, \mathbf{s}, \mathbf{r}, \delta_1) = \min_{\mathbf{z}} l(\mathbf{o}, \mathbf{s}, \mathbf{r}, \delta_1, \mathbf{z})$. Since we subtracted $\tilde{S}_{\mathbf{z}}^* \times \{\mathbf{z}\}$, this minimum value should be larger than the optimal value $L^*(\mathbf{o}; \mathbf{s}, \mathbf{r})$. By uniform continuity, we can find $\bar{\delta}_1$ such that for all $\mathbf{x}_1, \mathbf{x}_2$ and $\mathbf{z}$, if $||\mathbf{x}_1 - \mathbf{x}_2|| < \bar{\delta}_1$, then

$$|L(\mathbf{x}_1, \mathbf{z}, \mathbf{o}; \mathbf{s}, \mathbf{r}) - L(\mathbf{x}_2, \mathbf{z}, \mathbf{o}; \mathbf{s}, \mathbf{r})| < l(\mathbf{o}, \mathbf{s}, \mathbf{r}, \delta_1) - L^*(\mathbf{o}; \mathbf{s}, \mathbf{r}) \tag{98}$$

We define $\tilde{V}_{\mathbf{z}}(\mathbf{o}, \mathbf{s}, \mathbf{r}, \bar{\delta}_1)$ as the union of $\bar{\delta}_1$-balls centered on elements of $S_{\mathbf{z}}(\mathbf{o}, \mathbf{r}) \setminus (\tilde{S}_{\mathbf{z}}^*(\mathbf{o}, \mathbf{s}, \mathbf{r}, \delta_1) \times \{\mathbf{z}\})$. By the definition of $\bar{\delta}_1$, every element in $\tilde{V}_{\mathbf{z}}(\mathbf{o}, \mathbf{s}, \mathbf{r}, \bar{\delta}_1)$ has an $L$ value strictly greater than $L^*(\mathbf{o}; \mathbf{s}, \mathbf{r})$. Note that our preferred optimal solutions, which are feasible for both the original optimization problem and our integrated optimization layer (Equation 93), achieve the optimal values $\bar{J}^*(\mathbf{o}; \mathbf{r})$ and $L^*(\mathbf{o}; \mathbf{s}, \mathbf{r})$ for the functions $\bar{J}$ and $L$, respectively.

Let's consider any feasible solution in $\tilde{V}_{\mathbf{z}}(\mathbf{o}, \mathbf{s}, \mathbf{r}, \bar{\delta}_1)$. For such elements, the $\bar{J}$ value cannot be smaller than $\bar{J}^*(\mathbf{o}, \mathbf{r})$ due to the optimality of $\bar{J}^*$, while the $L$ value is strictly larger than $L^*(\mathbf{o}; \mathbf{s}, \mathbf{r})$ by the defining property of $\tilde{V}_{\mathbf{z}}$. Therefore, any element in $\tilde{V}_{\mathbf{z}}(\mathbf{o}, \mathbf{s}, \mathbf{r}, \bar{\delta}_1)$ has an objective value $\lambda L + \bar{J} > \lambda L^*(\mathbf{o}; \mathbf{s}, \mathbf{r}) + \bar{J}^*(\mathbf{o}, \mathbf{r})$ with strict inequality, making it impossible for such elements to be optimal solutions of the integrated optimization layer (Equation 93), regardless of $\lambda$.

It is clear that $\mathcal{U}_{\mathbf{z}} \setminus (\tilde{V}_{\mathbf{z}}(\mathbf{o}, \mathbf{s}, \mathbf{r}, \bar{\delta}_1) \times \{\mathbf{z}\})$ is compact. Therefore, $(\mathcal{U}_{\mathbf{z}} \setminus (\tilde{S}_{\mathbf{z}}^*(\mathbf{o}, \mathbf{s}, \mathbf{r}, \delta_1) \times \{\mathbf{z}\})) \setminus (\tilde{V}_{\mathbf{z}}(\mathbf{o}, \mathbf{s}, \mathbf{r}, \bar{\delta}_1) \times \{\mathbf{z}\})$ is also compact. We define the set of feasible solutions as $F(\mathbf{o}, \mathbf{r})$. Since all constraints are continuous with respect to $\mathbf{x}$, $F(\mathbf{o}, \mathbf{r})$ is an intersection of constraint-satisfaction regions of each constraint. Since each constraint includes equality, the constraint-satisfaction region is a continuous inverse image of a closed set and is thus closed. Then, $F(\mathbf{o}, \mathbf{r})$ is also closed and $\mathcal{U}_{\mathbf{z}} \cap F(\mathbf{o}, \mathbf{r})$ is compact for all $\mathbf{z}$.

Next, we can find the minimum value of $\bar{J}$ in the intersection of the two compact sets:

$$((\mathcal{U}_{\mathbf{z}} \cap F(\mathbf{o}, \mathbf{r})) \setminus (\tilde{S}_{\mathbf{z}}^*(\mathbf{o}, \mathbf{s}, \mathbf{r}, \delta_1) \times \{\mathbf{z}\})) \setminus (\tilde{V}_{\mathbf{z}}(\mathbf{o}, \mathbf{s}, \mathbf{r}, \bar{\delta}_1) \times \{\mathbf{z}\}) \tag{99}$$

Let $q_0(\mathbf{z}, \mathbf{o}, \mathbf{s}, \mathbf{r}, \delta_1)$ denote the minimum value (Note that $\bar{\delta}_1$ is a function of $\mathbf{z}, \mathbf{o}, \mathbf{s}, \mathbf{r}, \delta_1$). We can also find the maximum value $\bar{l}(\mathbf{z}, \mathbf{o}, \mathbf{s}, \mathbf{r}, \delta_1)$ and the minimum value $\underline{l}(\mathbf{z}, \mathbf{o}, \mathbf{s}, \mathbf{r}, \delta_1)$ of $L(\mathbf{u}, \mathbf{o}; \mathbf{s}, \mathbf{r})$ in $\mathcal{U}_{\mathbf{z}} \cap F(\mathbf{o}, \mathbf{r})$ because $L(\mathbf{u}, \mathbf{o}; \mathbf{s}, \mathbf{r})$ is continuous with respect to $\mathbf{x}$. Thus, when we define $\underline{\lambda}_{\mathbf{z}}(\mathbf{o}, \mathbf{s}, \mathbf{r}, \epsilon_2)$ as:

$$\underline{\lambda}_{\mathbf{z}}(\mathbf{o}, \mathbf{s}, \mathbf{r}, \epsilon_2) := \frac{q_0(\mathbf{z}, \mathbf{o}, \mathbf{s}, \mathbf{r}, \delta_1) - \bar{J}^*(\mathbf{o}; \mathbf{r})}{\bar{l}(\mathbf{z}, \mathbf{o}, \mathbf{s}, \mathbf{r}, \delta_1) - \underline{l}(\mathbf{z}, \mathbf{o}, \mathbf{s}, \mathbf{r}, \delta_1)} \tag{100}$$

and $\underline{\lambda}(\mathbf{o}, \mathbf{s}, \mathbf{r}, \epsilon_2)$ as:

$$\underline{\lambda}(\mathbf{o}, \mathbf{s}, \mathbf{r}, \epsilon_2) := \min_{\mathbf{z} \in Proj_{\mathcal{Z}}(\mathcal{U})} \underline{\lambda}_{\mathbf{z}}(\mathbf{o}, \mathbf{s}, \mathbf{r}, \epsilon_2) \tag{101}$$

for any positive $\lambda < \underline{\lambda}(\mathbf{o}, \mathbf{s}, \mathbf{r}, \epsilon_2)$, there is no solution of the integrated optimization problem Equation 93 in $((\mathcal{U}_{\mathbf{z}} \cap F(\mathbf{o}, \mathbf{r})) \setminus (\tilde{S}_{\mathbf{z}}^*(\mathbf{o}, \mathbf{s}, \mathbf{r}, \delta_1) \times \{\mathbf{z}\})) \setminus (\tilde{V}_{\mathbf{z}}(\mathbf{o}, \mathbf{s}, \mathbf{r}, \bar{\delta}_1) \times \{\mathbf{z}\})$. Since any element of $\tilde{V}_{\mathbf{z}}(\mathbf{o}, \mathbf{s}, \mathbf{r}, \bar{\delta}_1)$ cannot be an optimal solution of Equation 93, all optimal solutions of Equation 93 are elements of $\tilde{S}_{\mathbf{z}}^*(\mathbf{o}, \mathbf{s}, \mathbf{r}, \delta_1)$ for some $\mathbf{z}$. That is, all of them are at most in distance $\delta_1$ from a preferred optimal solution of the original optimization stage (Equation 13). Thus, we can obtain the inequalities below. Inequalities below are based on triangle inequalities and hold for any $\lambda$, including negative numbers.

$$\min_{\mathbf{u} \in F(\mathbf{o}, \mathbf{r})} (\lambda L(\mathbf{u}, \mathbf{o}; \mathbf{s}, \mathbf{r}) + \bar{J}(\mathbf{u}; \mathbf{o}, \mathbf{r})) = \min_{\mathbf{u} \in \tilde{S}^*_{\mathbf{z}} \times \{\mathbf{z}\}} (\lambda L(\mathbf{u}, \mathbf{o}; \mathbf{s}, \mathbf{r}) + \bar{J}(\mathbf{u}; \mathbf{o}, \mathbf{r}))$$

$$\geq \min_{\mathbf{u} \in \tilde{S}^*_{\mathbf{z}} \times \{\mathbf{z}\}} \lambda L(\mathbf{u}, \mathbf{o}; \mathbf{s}, \mathbf{r}) + \min_{\mathbf{u} \in \tilde{S}^*_{\mathbf{z}} \times \{\mathbf{z}\}} \bar{J}(\mathbf{u}; \mathbf{o}, \mathbf{r}) \quad (102)$$

$$= \min_{\mathbf{u} \in \tilde{S}^*_{\mathbf{z}} \times \{\mathbf{z}\}} \lambda L(\mathbf{u}, \mathbf{o}; \mathbf{s}, \mathbf{r}) + \bar{J}^*(\mathbf{o}, \mathbf{r})$$

$$\min_{\mathbf{u} \in F(\mathbf{o}, \mathbf{r})} (\lambda L(\mathbf{u}, \mathbf{o}; \mathbf{s}, \mathbf{r}) + \bar{J}(\mathbf{u}; \mathbf{o}, \mathbf{r})) = \min_{\mathbf{u} \in \tilde{S}^*_{\mathbf{z}} \times \{\mathbf{z}\}} (\lambda L(\mathbf{u}, \mathbf{o}; \mathbf{s}, \mathbf{r}) + \bar{J}(\mathbf{u}; \mathbf{o}, \mathbf{r}))$$

$$= - \min_{\mathbf{u} \in \tilde{S}^*_{\mathbf{z}} \times \{\mathbf{z}\}} (-\lambda L(\mathbf{u}, \mathbf{o}; \mathbf{s}, \mathbf{r}))$$

$$+ \min_{\mathbf{u} \in \tilde{S}^*_{\mathbf{z}} \times \{\mathbf{z}\}} (-\lambda L(\mathbf{u}, \mathbf{o}; \mathbf{s}, \mathbf{r})) + \min_{\mathbf{u} \in \tilde{S}^*_{\mathbf{z}} \times \{\mathbf{z}\}} (\lambda L(\mathbf{u}, \mathbf{o}; \mathbf{s}, \mathbf{r}) + \bar{J}(\mathbf{u}; \mathbf{o}, \mathbf{r}))$$

$$= \max_{\mathbf{u} \in \tilde{S}^*_{\mathbf{z}} \times \{\mathbf{z}\}} \lambda L(\mathbf{u}, \mathbf{o}; \mathbf{s}, \mathbf{r})$$

$$+ \min_{\mathbf{u} \in \tilde{S}^*_{\mathbf{z}} \times \{\mathbf{z}\}} (-\lambda L(\mathbf{u}, \mathbf{o}; \mathbf{s}, \mathbf{r})) + \min_{\mathbf{u} \in \tilde{S}^*_{\mathbf{z}} \times \{\mathbf{z}\}} (\lambda L(\mathbf{u}, \mathbf{o}; \mathbf{s}, \mathbf{r}) + \bar{J}(\mathbf{u}; \mathbf{o}, \mathbf{r}))$$

$$\leq \max_{\mathbf{u} \in \tilde{S}^*_{\mathbf{z}} \times \{\mathbf{z}\}} \lambda L(\mathbf{u}, \mathbf{o}; \mathbf{s}, \mathbf{r}) + \min_{\mathbf{u} \in \tilde{S}^*_{\mathbf{z}} \times \{\mathbf{z}\}} \bar{J}(\mathbf{u}; \mathbf{o}, \mathbf{r})$$

$$= \max_{\mathbf{u} \in \tilde{S}^*_{\mathbf{z}} \times \{\mathbf{z}\}} \lambda L(\mathbf{u}, \mathbf{o}; \mathbf{s}, \mathbf{r}) + \bar{J}^*(\mathbf{o}, \mathbf{r})$$

$$(103)$$

In conclusion, we have established that for any $\lambda < \underline{\lambda}(\mathbf{o}, \mathbf{s}, \mathbf{r}, \epsilon_2)$, all optimal solutions of the integrated optimization layer (Equation 93) must lie in $\tilde{S}^*_{\mathbf{z}}(\mathbf{o}, \mathbf{s}, \mathbf{r}, \delta_1)$ for some $\mathbf{z}$. From our inequalities:

$$\min_{\mathbf{u} \in \tilde{S}^*_{\mathbf{z}} \times \{\mathbf{z}\}} \lambda L(\mathbf{u}) + \bar{J}^*(\mathbf{o}, \mathbf{r}) < \min_{\mathbf{u} \in F(\mathbf{o}, \mathbf{r})} (\lambda L(\mathbf{u}) + \bar{J}(\mathbf{u}; \mathbf{o}, \mathbf{r})) < \max_{\mathbf{u} \in \tilde{S}^*_{\mathbf{z}} \times \{\mathbf{z}\}} \lambda L(\mathbf{u}) + \bar{J}^*(\mathbf{o}, \mathbf{r}) \quad (104)$$

Since every element $\mathbf{u} \in \tilde{S}^*_{\mathbf{z}}(\mathbf{o}, \mathbf{s}, \mathbf{r}, \delta_1) \times \{\mathbf{z}\}$ is within distance $\delta_1$ of some preferred optimal solution, and our choice of $\delta_1$ ensures that $|L(\mathbf{u}, \mathbf{o}; \mathbf{s}, \mathbf{r}) - L^*(\mathbf{o}; \mathbf{s}, \mathbf{r})| < \epsilon_2/3$, we obtain:

$$\max_{\mathbf{u} \in \tilde{S}^*_{\mathbf{z}} \times \{\mathbf{z}\}} \lambda L(\mathbf{u}, \mathbf{o}; \mathbf{s}, \mathbf{r}) < \lambda \left( L^*(\mathbf{o}; \mathbf{s}, \mathbf{r}) + \frac{\epsilon_2}{3} \right) \quad (105)$$

Subtracting $\lambda L^*(\mathbf{o}; \mathbf{s}, \mathbf{r}) + \bar{J}^*(\mathbf{o}, \mathbf{r})$ from all parts of our sandwich inequality, we conclude that any optimal solution $\mathbf{u}^*$ of Equation 93 satisfies:

$$|\lambda L(\mathbf{u}^*, \mathbf{o}; \mathbf{s}, \mathbf{r}) + \bar{J}(\mathbf{x}^*, \mathbf{z}^*; \mathbf{o}, \mathbf{r}) - \lambda L^*(\mathbf{o}; \mathbf{s}, \mathbf{r}) - \bar{J}^*(\mathbf{o}; \mathbf{r})| < \frac{\lambda \epsilon_2}{3} \quad (106)$$

Setting $\lambda_0(\mathbf{o}, \mathbf{s}, \mathbf{r}, \epsilon_2) = \underline{\lambda}(\mathbf{o}, \mathbf{s}, \mathbf{r}, \epsilon_2)$ completes Step 1.

**Step 2.** Now, we fix $\lambda$ and find $\beta_0$ that makes the minimum value of the first and the second minimization problem of Equation 87 have distance less than $\lambda \epsilon_2/3$ from the minimal value of their constrained (unconverted) versions. By Assumptions 1 and 2, both $\lambda L(\mathbf{u}, \mathbf{o}; \mathbf{s}, \mathbf{r}) + \bar{J}(\mathbf{u}; \mathbf{o}, \mathbf{r})$ and $\bar{J}(\mathbf{u}; \mathbf{o}, \mathbf{r})$ are continuous with respect to $\mathbf{x}$, and our domain with respect to $\mathbf{x}$ can be restricted to a compact set for any $\mathbf{z}$. Then, these two functions are uniformly continuous, so we can find $\delta_2(\mathbf{o}, \mathbf{s}, \mathbf{r}, \epsilon_2, \lambda)$ and $\delta_3(\mathbf{o}, \mathbf{r}, \epsilon_2, \lambda)$ such that for all $\mathbf{x}_1, \mathbf{x}_2$ and $\mathbf{z}$:

- if $\|\mathbf{x}_1 - \mathbf{x}_2\| < \delta_2(\mathbf{o}, \mathbf{s}, \mathbf{r}, \epsilon_2, \lambda)$, then

$$|\lambda L(\mathbf{x}_1, \mathbf{z}, \mathbf{o}; \mathbf{s}, \mathbf{r}) + \bar{J}(\mathbf{x}_1, \mathbf{z}; \mathbf{o}, \mathbf{r}) - \lambda L(\mathbf{x}_2, \mathbf{z}, \mathbf{o}; \mathbf{s}, \mathbf{r}) - \bar{J}(\mathbf{x}_2, \mathbf{z}; \mathbf{o}, \mathbf{r})| < \frac{\lambda \epsilon_2}{3} \quad (107)$$

- if $\|\mathbf{x}_1 - \mathbf{x}_2\| < \delta_3(\mathbf{o}, \mathbf{r}, \epsilon_2, \lambda)$, then

$$|\bar{J}(\mathbf{x}_1, \mathbf{z}; \mathbf{o}, \mathbf{r}) - \bar{J}(\mathbf{x}_2, \mathbf{z}; \mathbf{o}, \mathbf{r})| < \frac{\lambda \epsilon_2}{3} \quad (108)$$

Since we can treat $\lambda L(\mathbf{u}, \mathbf{o}; \mathbf{s}, \mathbf{r}) + \bar{J}(\mathbf{u}; \mathbf{o}, \mathbf{r})$ as another $\bar{J}$ and it satisfies Assumption 1 (as $\bar{c}$ remains the same and still satisfies Assumption 1), we can apply Proposition 5 to both Equation 93 and Equation 13.

Thus, we can choose

- $\underline{\boldsymbol{\beta}}_1(\mathbf{o}, \mathbf{s}, \mathbf{r}, \epsilon_2, \lambda)$ such that for any $\boldsymbol{\beta} > \underline{\boldsymbol{\beta}}_1$, the optimal solution of the first minimization problem in Equation 87 lies within distance $\delta_2(\mathbf{o}, \mathbf{s}, \mathbf{r}, \epsilon_2, \lambda)$ of some optimal solution of Equation 93.

- $\underline{\boldsymbol{\beta}}_2(\mathbf{o}, \mathbf{r}, \epsilon_2)$ such that for any $\boldsymbol{\beta} > \underline{\boldsymbol{\beta}}_2(\mathbf{u}, \mathbf{o}, \mathbf{r}, \epsilon_2)$, the optimal solution of the second minimization problem in Equation 87 lies within distance $\delta_3(\mathbf{o}, \mathbf{r}, \epsilon_2)$ of some optimal solution of Equation 13.

Finally, when we set $\boldsymbol{\beta}_0$ as:

$$\boldsymbol{\beta}_0(\mathbf{o}, \mathbf{r}, \lambda, \mathbf{s}, \epsilon_2) = \max(\underline{\boldsymbol{\beta}}_1(\mathbf{o}, \mathbf{s}, \mathbf{r}, \epsilon_2, \lambda), \underline{\boldsymbol{\beta}}_2(\mathbf{o}, \mathbf{r}, \epsilon_2)) \tag{109}$$

all $\boldsymbol{\beta} > \boldsymbol{\beta}_0(\mathbf{o}, \mathbf{r}, \lambda, \mathbf{s}, \epsilon_2)$ satisfy the following:

$$
\begin{aligned}
&\lambda|\tilde{L}(\mathbf{o}; \mathbf{s}, \mathbf{r}, \boldsymbol{\beta}, \lambda) - L^*(\mathbf{o}; \mathbf{s}, \mathbf{r})| \\
&= \left| \min_{\mathbf{u} \in \mathcal{U}} \left( \lambda L(\mathbf{u}, \mathbf{o}; \mathbf{s}, \mathbf{r}) + \bar{J}(\mathbf{u}; \mathbf{o}, \mathbf{r}) - \boldsymbol{\beta}^\top \min(\bar{\mathbf{c}}(\mathbf{u}; \mathbf{o}, \mathbf{r}), \mathbf{0}) \right) \right. \\
&\quad \left. - \min_{\mathbf{u} \in \mathcal{U}} \left( \bar{J}(\mathbf{u}; \mathbf{o}, \mathbf{r}) - \boldsymbol{\beta}^\top \min(\bar{\mathbf{c}}(\mathbf{u}; \mathbf{o}, \mathbf{r}), \mathbf{0}) \right) - \lambda L^*(\mathbf{o}; \mathbf{s}, \mathbf{r}) \right| \\
&= \left| \left( \min_{\mathbf{u} \in \mathcal{U}} \left( \lambda L(\mathbf{u}, \mathbf{o}; \mathbf{s}, \mathbf{r}) + \bar{J}(\mathbf{u}; \mathbf{o}, \mathbf{r}) - \boldsymbol{\beta}^\top \min(\bar{\mathbf{c}}(\mathbf{u}; \mathbf{o}, \mathbf{r}), \mathbf{0}) \right) \right. \right. \\
&\quad \left. - \lambda L(\mathbf{u}^*, \mathbf{o}; \mathbf{s}, \mathbf{r}) - \bar{J}(\mathbf{x}^*, \mathbf{z}^*; \mathbf{o}, \mathbf{r}) \right) \\
&\quad - \left( \min_{\mathbf{u} \in \mathcal{U}} \left( \bar{J}(\mathbf{u}; \mathbf{o}, \mathbf{r}) - \boldsymbol{\beta}^\top \min(\bar{\mathbf{c}}(\mathbf{u}; \mathbf{o}, \mathbf{r}), \mathbf{0}) \right) - \bar{J}^*(\mathbf{o}, \mathbf{r}) \right) \\
&\quad \left. + \left( \lambda L(\mathbf{u}^*, \mathbf{o}; \mathbf{s}, \mathbf{r}) + \bar{J}(\mathbf{x}^*, \mathbf{z}^*; \mathbf{o}, \mathbf{r}) - \lambda L^*(\mathbf{o}; \mathbf{s}, \mathbf{r}) - \bar{J}^*(\mathbf{o}, \mathbf{r}) \right) \right| \\[2mm]
&\leq \left| \min_{\mathbf{u} \in \mathcal{U}} \left( \lambda L(\mathbf{u}, \mathbf{o}; \mathbf{s}, \mathbf{r}) + \bar{J}(\mathbf{u}; \mathbf{o}, \mathbf{r}) - \boldsymbol{\beta}^\top \min(\bar{\mathbf{c}}(\mathbf{u}; \mathbf{o}, \mathbf{r}), \mathbf{0}) \right) \right. \\
&\quad \left. - \lambda L(\mathbf{u}^*, \mathbf{o}; \mathbf{s}, \mathbf{r}) - \bar{J}(\mathbf{x}^*, \mathbf{z}^*; \mathbf{o}, \mathbf{r}) \right| \\
&\quad + \left| \min_{\mathbf{u} \in \mathcal{U}} \left( \bar{J}(\mathbf{u}; \mathbf{o}, \mathbf{r}) - \boldsymbol{\beta}^\top \min(\bar{\mathbf{c}}(\mathbf{u}; \mathbf{o}, \mathbf{r}), \mathbf{0}) \right) - \bar{J}^*(\mathbf{o}, \mathbf{r}) \right| \\
&\quad + \left| \lambda L(\mathbf{u}^*, \mathbf{o}; \mathbf{s}, \mathbf{r}) + \bar{J}(\mathbf{x}^*, \mathbf{z}^*; \mathbf{o}, \mathbf{r}) - \lambda L^*(\mathbf{o}; \mathbf{s}, \mathbf{r}) - \bar{J}^*(\mathbf{o}, \mathbf{r}) \right| \\
&< \frac{\lambda \epsilon_2}{3} + \frac{\lambda \epsilon_2}{3} + \frac{\lambda \epsilon_2}{3} = \lambda \epsilon_2
\end{aligned}
\tag{110}
$$

Note that $\mathbf{u}^*$ is an optimal solution of Equation 93 and $\bar{J}^*(\mathbf{o}; \mathbf{r})$ is the optimal value of the original optimization stage Equation 13. The first equality is from the definition of the approximated loss function Equation 87, followed by the first inequality from the triangle inequality. The second inequality is from the result of Step 1 and the definition of $\delta_2, \delta_3$. Therefore, the statement is proven.

$\square$

Therefore, we can use $\tilde{L}(\mathbf{o}; \mathbf{s}, \mathbf{r}, \boldsymbol{\beta}, \lambda)$ as our loss function for training since it is continuous with respect to the continuous part of $\mathbf{o}$ and approaches to $L^*(\mathbf{o}; \mathbf{s}, \mathbf{r})$ when $\lambda \to 0$ and $\boldsymbol{\beta} \to \infty$.

# E  COMPUTATION OF GRADIENT

To train the information processing module (*i.e.*, to update the parameters to minimize our approximate loss function), we need to obtain the gradients of the approximated loss function with respect to the model parameters. The link between the information processing module and the approximate loss function is the information processing module output $\mathbf{o} \in \mathcal{O} \subset R^{n_{oc}} \times N_0^{n_{od}}$ that has $n_{oc}$ continuous elements and $n_{od}$ discrete elements. Let $\mathbf{o}_c \in \mathcal{O}_c \subset R^{n_{oc}}$ and $\mathbf{o}_d \in \mathcal{O}_d \subset N_0^{n_{od}}$ denote the continuous and discrete part, respectively. Since we cannot backpropagate gradients through the discrete part, we must handle them in an alternative way. We present the method for calculating the gradients for the real output vector first, and then present the method for calculating the gradients regarding the outputs from internal test data.

## E.1  REAL OUTPUT

In this subsection, we deal with how to compute the gradient for the real output vector (including discrete components). First, we present how to address the effect of the infinitesimal change of parameters on the approximate loss function through the discrete part to train the information processing module. We cannot directly define the gradient with respect to the discrete elements. However, these discrete elements are generally computed as a result of rounding or classification. For example, many classification models compute some real number associated with each class, and the class with the largest value can be regarded as the classification output. Thus, since continuous values are propagated between layers in information processing modules, discrete outputs can be considered as a sole function of some continuous interim results of the information processing module.

We need to obtain the *virtual partial derivative* with respect to the continuous interim results to train the information processing module. Although this is not the real gradient, this should be useful to update the model parameters to potentially result in a smaller approximate loss function. Let

$$\mathbf{o}_d = (o_{d1}, o_{d2}, \ldots, o_{dn_{od}}) \in \mathcal{O}_{d1} \times \mathcal{O}_{d2} \times \ldots \mathcal{O}_{dn_{od}} \tag{111}$$

with $o_{di} = g_i(f(\mathbf{y}_i; \mathbf{w}_i))$, $f(\mathbf{y}_i; \mathbf{w}_i) \in R^{m_i}$ and $g_i : \mathbb{R}^{m_i} \to \mathcal{O}_{di}$. Since $g_i(f(\mathbf{y}_i; \mathbf{w}_i))$ is clearly discontinuous by definition and thus $\tilde{L}(\mathbf{o}; \mathbf{s}, \mathbf{r}, \boldsymbol{\beta}, \lambda)$ is also discontinuous with respect to $\mathbf{w}_i$, we need to obtain the *softened approximated loss function* that is continuous and differentiable with respect to $f(\mathbf{y}_i; \mathbf{w}_i)$. Note that we assume $f(\mathbf{y}_i; \mathbf{w}_i)$ does not share elements with each other. If not, we can treat it as if there are two copies of one variable, and then the effects of them will be added when we back-propagate the (approximate) gradient.

We construct the *softened approximated loss function* (Equation 112) for each $f(\mathbf{y}_i; \mathbf{w}_i)$ as an expectation of $\tilde{L}$ with respect to the stochastic selection of $o_{di}$ via softened probability distribution $p(o_{di}; f(\mathbf{y}_i; \mathbf{w}_i))$ for $g_i(f(\mathbf{y}_i; \mathbf{w}_i))$. The softened probability distribution $p(o_{di}; f(\mathbf{y}_i; \mathbf{w}_i))$ can be set by various methods, but it needs to be continuous and partial-differentiable with respect to $f(\mathbf{y}_i; \mathbf{w}_i)$ for any $o_{di} \in \mathcal{O}_{di}$. One example can be setting $o_{di} = g_i(f(\mathbf{y}_i; \mathbf{w}_i) + \boldsymbol{\mu})$ for $\boldsymbol{\mu}$ following standard normal distribution, *i.e.*, $\mu_j \sim \mathcal{N}(0, 1)$, $j = 1, \ldots, m_i$. Another example can be softmax. Note that $\mathbf{o}_{d-i}$ denotes all elements of $\mathbf{o}_d$ other than the $i$-th element.

$$EL_i(f(\mathbf{y}_i; \mathbf{w}_i); \mathbf{o}_c, \mathbf{o}_{d-i}, \mathbf{s}, \mathbf{r}, \boldsymbol{\beta}, \lambda) = \sum_{o_{di} \in \mathcal{O}_{di}} p(o_{di}; f(\mathbf{y}_i; \mathbf{w}_i)) \tilde{L}(\mathbf{o}_c, o_{di}, \mathbf{o}_{d-i}; \mathbf{s}, \mathbf{r}, \boldsymbol{\beta}, \lambda) \tag{112}$$

Since $p(o_{di}; \mathbf{w}_i)$ is continuous and partial-differentiable with respect to $f(\mathbf{y}_i; \mathbf{w}_i)$, we can well-define a *virtual partial derivative* with respect to $f(\mathbf{y}_i; \mathbf{w}_i)$ as follows.

$$
\begin{aligned}
VPD_{di} &:= \frac{\partial}{\partial \mathbf{w}_i} EL_i(f(\mathbf{y}_i; \mathbf{w}_i); \mathbf{o}_c, \mathbf{o}_{d-i}, \mathbf{s}, \mathbf{r}, \boldsymbol{\beta}, \lambda) \\
&= \sum_{o_{di} \in \mathcal{O}_{di}} \frac{\partial p(o_{di}; f(\mathbf{y}_i; \mathbf{w}_i))}{\partial f(\mathbf{y}_i; \mathbf{w}_i)} \tilde{L}(\mathbf{o}_c, o_{di}, \mathbf{o}_{d-i}; \mathbf{s}, \mathbf{r}, \boldsymbol{\beta}, \lambda)
\end{aligned}
\tag{113}
$$

Gradient-descent method using this virtual partial derivative (Equation 113) decreases the probability of high $\tilde{L}$ and increases the probability of low $\tilde{L}$. This enables the information processing module to

produce output $\mathbf{o}$, which results in a lower approximate loss function more frequently. Thus, we can use the virtual partial derivative Equation 113 for training the information processing module as a substitute for real gradients.

Now, we deal with the continuous elements. Theorem 2 guarantees only continuity, not differentiability of $\tilde{L}(\mathbf{o}; \mathbf{s}, \mathbf{r}, \boldsymbol{\beta}, \lambda)$. Thus, we need to obtain the *virtual partial derivative* of $\tilde{L}$, which can be used as real gradients for training when it is not differentiable. Similar to the definition of gradient, we can define our *virtual partial derivative* as Equation 114. Note that $\mathbf{o}_{c-i}$ denotes all elements of $\mathbf{o}_c$ other than the $i$th element, and we can arbitrarily choose sufficiently small $\rho$.

$$VPD_{ci} := \frac{1}{\rho o_{ci}}(\tilde{L}(o_{ci} + \rho o_{ci}, \mathbf{o}_{c-i}, \mathbf{o}_d; \mathbf{s}, \mathbf{r}, \boldsymbol{\beta}, \lambda) - \tilde{L}(\mathbf{o}; \mathbf{s}, \mathbf{r}, \boldsymbol{\beta}, \lambda)) \tag{114}$$

In some cases, we can obtain the exact gradient by differentiating $\tilde{L}(\mathbf{o}; \mathbf{s}, \mathbf{r}, \boldsymbol{\beta}, \lambda)$. For simplicity, we let

$$P(\mathbf{u}, \mathbf{o}; \mathbf{s}, \mathbf{r}) := \lambda L(\mathbf{u}, \mathbf{o}; \mathbf{s}, \mathbf{r}) + \bar{J}(\mathbf{u}; \mathbf{o}, \mathbf{r}) - \boldsymbol{\beta}^\top \min(\bar{\mathbf{c}}(\mathbf{u}; \mathbf{o}, \mathbf{r}), \mathbf{0}) \tag{115}$$

and

$$Q(\mathbf{u}, \mathbf{o}; \mathbf{r}) := \bar{J}(\mathbf{u}; \mathbf{o}, \mathbf{r}) - \boldsymbol{\beta}^\top \min(\bar{\mathbf{c}}(\mathbf{u}; \mathbf{o}, \mathbf{r}), \mathbf{0}) \tag{116}$$

Then, a set of conditions sufficient for the analytical calculation of the gradient is presented in Condition 4.

**Condition 4.** For a given $\mathbf{s}, \mathbf{r}$, and $\mathbf{o}_0$, there exists $\epsilon_4(\mathbf{s}, \mathbf{r}, \mathbf{o}_0) > 0$ that makes the following statements satisfy in neighborhood $B(\mathbf{o}_{c0}, \epsilon_4) \times \{\mathbf{o}_{d0}\} \subset \mathcal{O}$.

1. $P(\mathbf{u}, \mathbf{o}; \mathbf{s}, \mathbf{r})$ and $Q(\mathbf{u}, \mathbf{o}; \mathbf{r})$ have gradients with respect to $\mathbf{o}_c$ at $\mathbf{o}_c = \mathbf{o}_{c0}$.

2. $P(\mathbf{u}, \mathbf{o}; \mathbf{s}, \mathbf{r})$ and $Q(\mathbf{u}, \mathbf{o}; \mathbf{r})$ have only one minimum $(\mathbf{x}_p(\mathbf{o}), \mathbf{z}_p(\mathbf{o})) \in \mathcal{U}$ and $(\mathbf{x}_q(\mathbf{o}), \mathbf{z}_q(\mathbf{o})) \in \mathcal{U}$, respectively. That is,

$$\{(\mathbf{x}_p, \mathbf{z}_p)\} = \arg\min_{\mathbf{u} \in \mathcal{U}} \left(\lambda L(\mathbf{u}, \mathbf{o}; \mathbf{s}, \mathbf{r}) + \bar{J}(\mathbf{u}; \mathbf{o}, \mathbf{r}) - \boldsymbol{\beta}^\top \min(\bar{\mathbf{c}}(\mathbf{u}; \mathbf{o}, \mathbf{r}), \mathbf{0})\right) \tag{117}$$

$$\{(\mathbf{x}_q, \mathbf{z}_q)\} = \arg\min_{\mathbf{u} \in \mathcal{U}} \left(\bar{J}(\mathbf{u}; \mathbf{o}, \mathbf{r}) - \boldsymbol{\beta}^\top \min(\bar{\mathbf{c}}(\mathbf{u}; \mathbf{o}, \mathbf{r}), \mathbf{0})\right) \tag{118}$$

3. $\mathbf{x}_p(\mathbf{o})$ and $\mathbf{x}_q(\mathbf{o})$ have gradients with respect to $\mathbf{o}_c$ at $\mathbf{o}_0$.

4. $\mathbf{z}_p(\mathbf{o})$ and $\mathbf{z}_q(\mathbf{o})$ are constant.

5. $(\mathbf{x}_p(\mathbf{o}_0), \mathbf{z}_p(\mathbf{o}_0))$ and $(\mathbf{x}_q(\mathbf{o}_0), \mathbf{z}_q(\mathbf{o}_0))$ are in the interior of $\mathcal{U}$.

Then, we derive the exact gradient of $\tilde{L}(\mathbf{o}; \mathbf{s}, \mathbf{r}, \boldsymbol{\beta}, \lambda)$ with respect to $\mathbf{o}$ under Condition 4 in Theorem 6.

**Proposition 6.** Under Assumption 1, Assumption 2, and Condition 4, the gradient of $\tilde{L}(\mathbf{o}; \mathbf{s}, \mathbf{r}, \boldsymbol{\beta}, \lambda)$ with respect to $\mathbf{o}_c$ at $\mathbf{o}_0$ can be calculated as follows.

$$\nabla_{\mathbf{o}_c} \tilde{L}(\mathbf{o}; \mathbf{s}, \mathbf{r}, \boldsymbol{\beta}, \lambda)(\mathbf{o}_0) = \frac{1}{\lambda}(\nabla_{\mathbf{o}_c} P(\mathbf{x}_p, \mathbf{o}_0) - \nabla_{\mathbf{o}_c} Q(\mathbf{x}_q, \mathbf{o}_0)) \tag{119}$$

*Proof.* Under Assumption 1, Assumption 2, and Condition 4, Equation 120 is a direct result of the chain rule. Note that the effect of the gradient of $\mathbf{z}$ with respect to $\mathbf{o}$ is 0 because of statement 4 of Condition 4.

$$\begin{aligned}\nabla_{\mathbf{o}_c} \tilde{L}(\mathbf{o}; \mathbf{s}, \mathbf{r}, \boldsymbol{\beta}, \lambda)(\mathbf{o}_0) = \frac{1}{\lambda}(&\nabla_{\mathbf{o}_c} P(\mathbf{x}_p, \mathbf{o}_0) + \nabla_{\mathbf{x}} P(\mathbf{x}_p, \mathbf{o}_0)\nabla_{\mathbf{o}_c}\mathbf{x}_p(\mathbf{o}_0) \\ &-\nabla_{\mathbf{o}_c} Q(\mathbf{x}_q, \mathbf{o}_0) - \nabla_{\mathbf{x}} Q(\mathbf{x}_q, \mathbf{o}_0)\nabla_{\mathbf{o}_c}\mathbf{x}_q(\mathbf{o}_0))\end{aligned} \tag{120}$$

Then, by the fifth statement of Condition 4, constraints $\mathbf{u} \in \mathcal{U}$ do not affect $\nabla_{\mathbf{o}_c} \mathbf{x}_p(\mathbf{o}_0)$ and $\nabla_{\mathbf{o}_c} \mathbf{x}_q(\mathbf{o}_0)$. Thus, as a sole minimality condition, we need to consider only stationarity with respect to $\mathbf{x}$, that is,

$$\nabla_{\mathbf{x}} P(\mathbf{x}_p, \mathbf{o}_0) = \nabla_{\mathbf{x}} Q(\mathbf{x}_q, \mathbf{o}_0) = 0 \tag{121}$$

Therefore, Equation 120 reduces to Equation 119. $\qquad\square$

Note that Condition 4 generally holds for problems we typically encounter. The reasons are as follows: Most of the basic functions we work with (polynomial, exponential, trigonometric, etc.) are differentiable, so statements 1 and 3 hold. The remaining statements are analogous to non-singularity conditions that hold almost everywhere. The key point of Proposition 6 is the elimination of gradients with respect to $\mathbf{x}$, and this is the direct result of minimization in the definition of $\tilde{L}$. Since we can choose an arbitrarily big $\mathcal{U}$ as long as it is bounded and the functions are well-defined, we can easily satisfy the fifth condition that removes the effect of constraints.

As the final part of this subsection, we summarize the computational cost of calculating the (virtual) gradient in terms of computational cost when solving the minimization problem in Proposition 7. This is important since we do not constrain the type of our optimization problem, and thus, the minimization can take significantly longer than other operations.

**Proposition 7.** When all $\mathcal{O}_{di}(1 \leq i \leq n_{od})$ are finite, the number of times we need to solve minimization problems (of $P$ or $Q$) to obtain virtual or exact gradients with respect to all elements of the information processing module output is as follows.

1. When we use Equation 114 to obtain virtual gradients with respect to continuous elements:

$$2 \times (n_{oc} + 1 + \sum_i^{n_{od}} |\mathcal{O}_{di}|) \tag{122}$$

2. When we use Equation 119 to obtain exact gradients with respect to continuous elements and $\nabla_{\mathbf{o}_c} P$ and $\nabla_{\mathbf{o}_c} Q$ are (locally) known analytically:

$$2 \times (1 + \sum_i^{n_{od}} |\mathcal{O}_{di}|) \tag{123}$$

*Proof.* By the definition of virtual partial derivative with respect to a discrete element (Equation 113), calculating it needs computation of $\tilde{L}$ for all $o_{di}$. Thus, the calculation of the virtual partial derivative with respect to all discrete elements requires $2 \times \sum_i^{n_{od}} |\mathcal{O}_{di}|$ times of minimization since one computation of $\tilde{L}$ needs two minimizations.

Meanwhile, Equation 114 needs a single computation of $\tilde{L}$ with the non-deviated value and one additional computation of $\tilde{L}$ for each continuous element of $\mathbf{o}$, which means that $2 \times (n_{oc} + 1)$ times of minimization are needed. As a result, $2 \times (n_{oc} + 1 + \sum_i^{n_{od}} |\mathcal{O}_{di}|)$ times of minimization are needed to find (virtual) gradients for the whole output when we use Equation 114.

In contrast, computation of Equation 119 requires only two times of minimization when we already know the analytical expression of $\nabla_{\mathbf{o}_c} P$ and $\nabla_{\mathbf{o}_c} Q$ locally. This provides partial derivatives for all continuous elements of $\mathbf{o}$ simultaneously. Therefore, we need only $2 \times (1 + \sum_i^{n_{od}} |\mathcal{O}_{di}|)$ times of minimization. $\qquad\square$

### E.2 GRADIENTS FOR INTERNAL TEST DATA

In this subsection, we present techniques for computing the (approximate) gradient of the approximate loss function using conservative testing procedures for training the chance-constrained method within the framework. For simplicity, we define $\mathbf{N}^\xi$ as $(N^{+\xi}, N^{-\xi})$. By starting from the definition equation 113, we can calculate the virtual partial gradient for the chance-constrained part (with conservative testing technique) as

$$VPD_{cr}^{\xi} = \sum_{k}^{n_t} \sum_{\{\mathbf{N}_{\mathbf{s}_{cr}=\bar{\mathbf{s}}_i, \mathbf{o}_{cr}=\bar{\mathbf{o}}_j}^{\xi}\}} \frac{\partial Pr^{\xi}(\{\mathbf{N}_{\mathbf{s}_{cr}=\bar{\mathbf{s}}_i, \mathbf{o}_{cr}=\bar{\mathbf{o}}_j}^{\xi}\})}{\partial f(\mathbf{y}_{cr}^{t_k}; \mathbf{w}_{cr})} \tilde{L}(\{\mathbf{N}_{\mathbf{s}_{cr}=\bar{\mathbf{s}}_i, \mathbf{o}_{cr}=\bar{\mathbf{o}}_j}^{\xi}\}, \mathbf{o}_{ncr}; \mathbf{s}, \mathbf{r}, \boldsymbol{\beta}, \lambda)$$

(124)

when $Pr^{\xi}(\{\mathbf{N}_{\mathbf{s}_{cr}=\bar{\mathbf{s}}_i, \mathbf{o}_{cr}=\bar{\mathbf{o}}_j}^{\xi}\})$ is defined as the probability of $\{\mathbf{N}_{\mathbf{s}_{cr}=\bar{\mathbf{s}}_i, \mathbf{o}_{cr}=\bar{\mathbf{o}}_j}^{\xi}\}$ under $p^{+\xi}(\bar{\mathbf{o}}_j; f(\mathbf{y}_{cr}^{t_k}; \mathbf{w}_{cr}))$. Here, we stochastically assign $\mathbf{1}^{+\xi}(\mathbf{o}_{cr}^{t_k}, \bar{\mathbf{o}}_j)$ and $\mathbf{1}^{-\xi}(\mathbf{o}_{cr}^{t_k}, \bar{\mathbf{o}}_j)$ at the same time, that is, $\mathbf{1}^{-\xi}(\mathbf{o}_{cr}^{t_k}, \bar{\mathbf{o}}_j)$ implies $\mathbf{1}^{+\xi}(\mathbf{o}_{cr}^{t_k}, \bar{\mathbf{o}}_j)$ and the latter solely occurs with probability $p^{+\xi}(\bar{\mathbf{o}}_j; f_{cr}(\mathbf{y}_{cr}^{t_k}; \mathbf{w}_{cr})) - p^{-\xi}(\bar{\mathbf{o}}_j; f_{cr}(\mathbf{y}_{cr}^{t_k}; \mathbf{w}_{cr}))$.

Then, by using the chain rule and defining $\{\mathbf{N}_{\mathbf{s}_{cr}=\bar{\mathbf{s}}_i, \mathbf{o}_{cr}=\bar{\mathbf{o}}_j}^{\xi, -k}\}$ as the number of (internal) test cases for each environment state and output without $k$ th test case, we can convert it as

$$VPD_{cr}^{\xi} = \sum_{k}^{n_t} \sum_{\{\mathbf{N}_{\mathbf{s}_{cr}=\mathbf{s}_{cr}^{t_k}, \mathbf{o}_{cr}=\bar{\mathbf{o}}_j}^{\xi}\}} \frac{\partial Pr^{\xi}(\{\mathbf{N}_{\mathbf{s}_{cr}=\mathbf{s}_{cr}^{t_k}, \mathbf{o}_{cr}=\bar{\mathbf{o}}_j}^{\xi}\})}{\partial f(\mathbf{y}_{cr}^{t_k}; \mathbf{w}_{cr})} E(\tilde{L}; \{\mathbf{N}_{\mathbf{s}_{cr}=\mathbf{s}_{cr}^{t_k}, \mathbf{o}_{cr}=\bar{\mathbf{o}}_j}^{\xi}\})$$

$$= \sum_{k}^{n_t} ( \sum_{\{\mathbf{N}_{\mathbf{s}_{cr}=\mathbf{s}_{cr}^{t_k}, \mathbf{o}_{cr}=\bar{\mathbf{o}}_j}^{\xi, -k}\}} \sum_{l} \frac{\partial p^{-\xi}(\bar{\mathbf{o}}_l; f(\mathbf{y}_{cr}^{t_k}; \mathbf{w}_{cr}))}{\partial f(\mathbf{y}_{cr}^{t_k}; \mathbf{w}_{cr})} Pr^{\xi}(\{\mathbf{N}_{\mathbf{s}_{cr}=\mathbf{s}_{cr}^{t_k}, \mathbf{o}_{cr}=\bar{\mathbf{o}}_j}^{\xi, -k}\})$$

$$(E(\tilde{L}; \{\mathbf{N}_{\mathbf{s}_{cr}=\mathbf{s}_{cr}^{t_k}, \mathbf{o}_{cr}=\bar{\mathbf{o}}_1}^{\xi, -k}, \cdots, \mathbf{N}_{\mathbf{s}_{cr}=\mathbf{s}_{cr}^{t_k}, \mathbf{o}_{cr}=\bar{\mathbf{o}}_{l-1}}^{\xi, -k}, \mathbf{N}_{\mathbf{s}_{cr}=\mathbf{s}_{cr}^{t_k}, \mathbf{o}_{cr}=\bar{\mathbf{o}}_l}^{\xi, -k} + (1,1),$$

$$\mathbf{N}_{\mathbf{s}_{cr}=\mathbf{s}_{cr}^{t_k}, \mathbf{o}_{cr}=\bar{\mathbf{o}}_{l+1}}^{\xi, -k}, \cdots, \mathbf{N}_{\mathbf{s}_{cr}=\mathbf{s}_{cr}^{t_k}, \mathbf{o}_{cr}=\bar{\mathbf{o}}_{N_o}}^{\xi, -k}\}) - E(\tilde{L}; \{\mathbf{N}_{\mathbf{s}_{cr}=\mathbf{s}_{cr}^{t_k}, \mathbf{o}_{cr}=\bar{\mathbf{o}}_j}^{\xi, -k}\})))$$

$$+ \sum_{\{\mathbf{N}_{\mathbf{s}_{cr}=\mathbf{s}_{cr}^{t_k}, \mathbf{o}_{cr}=\bar{\mathbf{o}}_j}^{\xi, -k}\}} \sum_{l} (\frac{\partial p^{+\xi}(\bar{\mathbf{o}}_l; f(\mathbf{y}_{cr}^{t_k}; \mathbf{w}_{cr}))}{\partial f(\mathbf{y}_{cr}^{t_k}; \mathbf{w}_{cr})} - \frac{\partial p^{-\xi}(\bar{\mathbf{o}}_l; f(\mathbf{y}_{cr}^{t_k}; \mathbf{w}_{cr}))}{\partial f(\mathbf{y}_{cr}^{t_k}; \mathbf{w}_{cr})}) Pr^{\xi}(\{\mathbf{N}_{\mathbf{s}_{cr}=\mathbf{s}_{cr}^{t_k}, \mathbf{o}_{cr}=\bar{\mathbf{o}}_j}^{\xi, -k}\})$$

$$(E(\tilde{L}; \{\mathbf{N}_{\mathbf{s}_{cr}=\mathbf{s}_{cr}^{t_k}, \mathbf{o}_{cr}=\bar{\mathbf{o}}_1}^{\xi, -k}, \cdots, \mathbf{N}_{\mathbf{s}_{cr}=\mathbf{s}_{cr}^{t_k}, \mathbf{o}_{cr}=\bar{\mathbf{o}}_{l-1}}^{\xi, -k}, \mathbf{N}_{\mathbf{s}_{cr}=\mathbf{s}_{cr}^{t_k}, \mathbf{o}_{cr}=\bar{\mathbf{o}}_l}^{\xi, -k} + (1,0),$$

$$\mathbf{N}_{\mathbf{s}_{cr}=\mathbf{s}_{cr}^{t_k}, \mathbf{o}_{cr}=\bar{\mathbf{o}}_{l+1}}^{\xi, -k}, \cdots, \mathbf{N}_{\mathbf{s}_{cr}=\mathbf{s}_{cr}^{t_k}, \mathbf{o}_{cr}=\bar{\mathbf{o}}_{N_o}}^{\xi, -k}\}) - E(\tilde{L}; \{\mathbf{N}_{\mathbf{s}_{cr}=\mathbf{s}_{cr}^{t_k}, \mathbf{o}_{cr}=\bar{\mathbf{o}}_j}^{\xi, -k}\}))))$$

$$\sum_{\{\mathbf{N}_{\mathbf{s}_{cr}=\mathbf{s}_{cr}^{t_k}, \mathbf{o}_{cr}=\bar{\mathbf{o}}_j}^{\xi, -k}\}} \sum_{l} (-\frac{\partial p^{+\xi}(\bar{\mathbf{o}}_l; f(\mathbf{y}_{cr}^{t_k}; \mathbf{w}_{cr}))}{\partial f(\mathbf{y}_{cr}^{t_k}; \mathbf{w}_{cr})}) Pr^{\xi}(\{\mathbf{N}_{\mathbf{s}_{cr}=\mathbf{s}_{cr}^{t_k}, \mathbf{o}_{cr}=\bar{\mathbf{o}}_j}^{\xi, -k}\})$$

$$(E(\tilde{L}; \{\mathbf{N}_{\mathbf{s}_{cr}=\mathbf{s}_{cr}^{t_k}, \mathbf{o}_{cr}=\bar{\mathbf{o}}_1}^{\xi, -k}, \cdots, \mathbf{N}_{\mathbf{s}_{cr}=\mathbf{s}_{cr}^{t_k}, \mathbf{o}_{cr}=\bar{\mathbf{o}}_{l-1}}^{\xi, -k}, \mathbf{N}_{\mathbf{s}_{cr}=\mathbf{s}_{cr}^{t_k}, \mathbf{o}_{cr}=\bar{\mathbf{o}}_l}^{\xi, -k} + (0,0),$$

$$\mathbf{N}_{\mathbf{s}_{cr}=\mathbf{s}_{cr}^{t_k}, \mathbf{o}_{cr}=\bar{\mathbf{o}}_{l+1}}^{\xi, -k}, \cdots, \mathbf{N}_{\mathbf{s}_{cr}=\mathbf{s}_{cr}^{t_k}, \mathbf{o}_{cr}=\bar{\mathbf{o}}_{N_o}}^{\xi, -k}\}) - E(\tilde{L}; \{\mathbf{N}_{\mathbf{s}_{cr}=\mathbf{s}_{cr}^{t_k}, \mathbf{o}_{cr}=\bar{\mathbf{o}}_j}^{\xi, -k}\}))))$$

(125)

when $E(\tilde{L}; \{\mathbf{N}_{\mathbf{s}_{cr}=\mathbf{s}_{cr}^{t_k}, \mathbf{o}_{cr}=\bar{\mathbf{o}}_j}^{\xi}\})$ is the expectation of $\tilde{L}$ under $\{\mathbf{N}_{\mathbf{s}_{cr}=\mathbf{s}_{cr}^{t_k}, \mathbf{o}_{cr}=\bar{\mathbf{o}}_j}^{\xi}\}$ (with probabilistic $\{\mathbf{N}_{\mathbf{s}_{cr}=\bar{\mathbf{s}}_i, \mathbf{o}_{cr}=\bar{\mathbf{o}}_j}^{\xi}\}$ for $\bar{\mathbf{s}}_i \neq \mathbf{s}_{cr}^{t_k}$). We can approximate this by fixing the outputs for other internal test data (thus, considering only the output of specific internal test data as stochastic) in computing expectations (thus, collapsing expectations into deterministic values). Further details about the technique are provided with empirical examples.

As an alternative way, we can make a continuous approximate function of $\tilde{L}$ for continuous:

$$\tilde{N}_{\mathbf{s}_{cr}=\bar{\mathbf{s}}_i, \mathbf{o}_{cr}=\bar{\mathbf{o}}_j}^{+\xi} := \sum_{k=1}^{n_t} \mathbf{1}(\mathbf{s}_{cr}^{t_k}, \bar{\mathbf{s}}_i) p^{+\xi}(\bar{\mathbf{o}}_l; f(\mathbf{y}_{cr}^{t_k}; \mathbf{w}_{cr}))$$

(126)

and

$$\tilde{N}_{\mathbf{s}_{cr}=\bar{\mathbf{s}}_i, \mathbf{o}_{cr}=\bar{\mathbf{o}}_j}^{-\xi} := \sum_{k=1}^{n_t} \mathbf{1}(\mathbf{s}_{cr}^{t_k}, \bar{\mathbf{s}}_i) p^{-\xi}(\bar{\mathbf{o}}_l; f(\mathbf{y}_{cr}^{t_k}; \mathbf{w}_{cr}))$$

(127)

# F SCALING LAW

To mathematically deal with the scaling law, we review the concept of $\zeta$-coverage introduced in (Kim et al., 2025).

**Definition 3.** [Modified from Definition 1 in (Kim et al., 2025)]

For $\zeta > 0$, we call a dataset $D$ is $\zeta$-coverage if and only if $D$ satisfies the following:

$$\bigcup_{\omega \in D} B(\omega, \zeta) = \Omega, \tag{128}$$

where $B(\omega, \zeta)$ is an open ball with radius $\zeta$ centered at $\omega$ and $\Omega$ is a compact metric space of potential inputs.

Note that $\zeta$-coverage is a weaker version of $\zeta$-informative (from Section C.1), considering only when $X$ is the full set. Now, we construct the safe set based on the internal test data with this concept. Our main method for identifying the safe set is as follows:

We employ internal test data in the product of the input space and the action space (denoted as $\Omega^*$). Then, we consider a radius $\zeta$ that makes the internal test data be $\zeta$-coverage, which is the condition that the union of $\zeta$-balls centered on the data contains the whole product space. Next, when we classify all data within the $\zeta$-range from internal test data as safe, our safe set classification can serve as an estimate of the real safe set. The probability of Type I errors (misclassifying unsafe actions as safe) and Type II errors (misclassifying safe actions as unsafe) is upper-bounded by the difference of the probability of the real safe set and the $\zeta$-inflated real safe set, and the difference of the probability of the real unsafe set and the $\zeta$-inflated real unsafe set.

When we use more internal test data, we can use a smaller $\zeta$ and the probability of a set difference becomes smaller. Conversely, when we decide the probability bound, we can decide the appropriate $\zeta$ to ensure it (with some condition of the real safe set). Then, we can compute the expectation of the required number of internal test data to ensure $\zeta$-coverage with such $\zeta$. With the process, under the following conditions, we can obtain the following theorem.

**Condition 5.** 1. $\Omega^*$ is a compact metric space with endowed probability measure $P^*$ that is defined on a Borel $\sigma$-algebra.

2. There exists a well-defined safe set $S$ that includes all pairs of input and action that satisfy the constraint. That is, the safe and unsafe pairs of input and action can be completely distinguishable in principle.

3. The safe set $S$ and its complement $\Omega^* \setminus S$ have finite boundary probability measure for dimension $d - 1$, that is,

$$\limsup_{r \to 0^+} \frac{P^*(\bigcup_{\omega \in S} B(\omega, r) \setminus S)}{r}, \quad \limsup_{r \to 0^+} \frac{P^*(\bigcup_{\omega \in \Omega^* \setminus S} B(\omega, r) \setminus (\Omega^* \setminus S))}{r} < \infty \tag{129}$$

**Condition 6.** All open balls with radius $r$ in the input space $\Omega^*$ have a probability measure of at least $\kappa r^d$ with constant $\kappa$ and the dimension of the input space $d$.

**Theorem 3.** *Under Conditions 5 and 6, when we assume that we can define the safety classification model as an arbitrary partition of $\Omega^*$, as $p$ decreases gradually, the expected number of required internal test data $N_{reqit}$ to upper-bound both Type I errors and Type II errors with $p$ as*

$$N_{reqit} \leq A p^{-2d} \tag{130}$$

*where $A$ is a constant and $d$ is the dimension of the input space.*

*Proof.* For simplicity, let $t_1, \ldots, t_{N_{sit}} (\in S)$ as safe internal test data and $t_{N_{sit}+1}, \ldots t_{N_{sit}+N_{usit}} (\in \Omega^* \setminus S)$ as unsafe internal test data ($N_{sit} + N_{usit} = n_t$). For $\zeta$ that makes the internal test data $t_1, \ldots, t_{n_t}$ be $\zeta$-coverage of $\Omega^*$, we classify pairs of input and action in $\bigcup_{i=1}^{n_{sit}} B(t_i, \zeta)$ as safe.

**Part 1**: Computation of appropriate $\zeta$ to upper bound the error probabilities.

In this part, we compute the required $\zeta$ to upper bound the error probabilities to $p$. Type I and Type II error probabilities are

$$\frac{P^*(\bigcup_{i=1}^{n_{sit}} B(t_i, \zeta) \setminus S)}{P^*(\Omega^* \setminus S)}, \frac{P^*(S \setminus \bigcup_{i=1}^{n_{sit}} B(t_i, \zeta))}{P^*(S)} \tag{131}$$

respectively. Based on the $\zeta$-coverage assumption, we have

$$\frac{P^* \left( S \setminus \bigcup_{i=1}^{n_{sit}} B(t_i, \zeta) \right)}{P^*(S)} \leq \frac{P^* \left( S \setminus \left( \Omega^* \setminus \left( \bigcup_{i=N_{sit}+1}^{N_{sit}+N_{usit}} B(t_i, \zeta) \right) \right) \right)}{P^*(S)}$$

$$= \frac{P^* \left( S \cap \bigcup_{i=N_{sit}+1}^{N_{sit}+N_{usit}} B(t_i, \zeta) \right)}{P^*(S)} \tag{132}$$

Since unsafe internal test data is not included in the safe set, we can obtain

$$\frac{P^* \left( S \cap \left( \bigcup_{i=N_{sit}+1}^{N_{sit}+N_{usit}} B(t_i, \zeta) \right) \right)}{P^*(S)} \leq \frac{P^* \left( S \cap \left( \bigcup_{\omega \in \Omega^* \setminus S} B(\omega, \zeta) \right) \right)}{P^*(S)}$$

$$= \frac{P^* \left( \left( \bigcup_{\omega \in \Omega^* \setminus S} B(\omega, \zeta) \right) \setminus (\Omega^* \setminus S) \right)}{P^*(S)} \tag{133}$$

Conversely, since all safe internal test data is included in $S$, we can obtain

$$\frac{P^* \left( \left( \bigcup_{i=1}^{n_{sit}} B(t_i, \zeta) \right) \setminus S \right)}{P^* (\Omega^* \setminus S)} \leq \frac{P^* \left( \left( \bigcup_{\omega \in S} B(\omega, \zeta) \right) \setminus S \right)}{P^* (\Omega^* \setminus S)} \tag{134}$$

By Condition 5, we have $(k_s, r_s)$ and $(k_{us}, r_{us})$ that satisfy

$$\forall r < r_s, \quad P^* \left( \bigcup_{\omega \in S} B(\omega, \zeta) \setminus S \right) < k_s r_s$$

$$\forall r < r_{us}, \quad P^* \left( \bigcup_{\omega \in \Omega^* \setminus S} B(\omega, \zeta) \setminus (\Omega^* \setminus S) \right) < k_{us} r_{us} \tag{135}$$

respectively. Now, when $\zeta$ satisfies

$$\zeta \leq \min \left( r_s, r_{us}, \frac{p P^*(S)}{k_{us}}, \frac{p P^*(\Omega^* \setminus S)}{k_s} \right), \tag{136}$$

both Type I error and Type II error probabilities are lower than or equal to $p$.

**Part 2**: Computation of the expected number of required internal test data to achieve $\zeta$-coverage with such $\zeta$.

In this part, we compute how much internal test data is needed to achieve $\zeta$-coverage with

$$\zeta := \min \left( r_s, r_{us}, \frac{p P^*(S)}{k_{us}}, \frac{p P^*(\Omega^* \setminus S)}{k_s} \right) \tag{137}$$

Since $\Omega^*$ is compact, we can obtain a finite subcover of the following open cover:

$$\left\{ B\left(\omega, \frac{\varsigma}{4}\right) \middle| \omega \in \Omega^* \right\} \tag{138}$$

Let this finite subcover as:

$$\left\{ B\left(\omega_1, \frac{\varsigma}{4}\right), \cdots, B\left(\omega_{n_b}, \frac{\varsigma}{4}\right) \right\} \tag{139}$$

and inductively define $H_i$ as $H_i := \phi$ if $B\left(\omega_i, \frac{\varsigma}{4}\right) \subset \bigcup_{j=1}^{i-1} H_j$, otherwise a $\frac{\varsigma}{4}$-ball centered somewhere in $B\left(\omega_i, \frac{\varsigma}{4}\right) \setminus \bigcup_{j=1}^{i-1} H_j$. Let $\{\omega'_1, \cdots, \omega'_{n_h}\}$ as the set of centers of $H_i$s which is not $\phi$. It is clear that

$$\bigcup_i B\left(\omega'_i, \frac{\varsigma}{2}\right) \supset \bigcup_i B\left(\omega_i, \frac{\varsigma}{4}\right) \supset \Omega^* \tag{140}$$

and the distance between $\omega'_i$ and $\omega'_j$ for any $i \neq j$ is at least $\varsigma/4$ since we define each $\omega'_i$ outside from $\bigcup_{j=1}^{i-1} H_j$.

Considering the $\varsigma/8$-balls centered on each $\omega'_i$, the balls cannot overlap and each ball has a probability measure of at least $\kappa\varsigma^d/8^d$ based on Condition 6. Since the sum of the probabilities of non-overlapping sets cannot exceed 1, we can obtain

$$n_h \leq \frac{8^d}{\kappa\varsigma^d} \tag{141}$$

Additionally, based on Condition 6, $B(\omega'_i, \frac{\varsigma}{2})$ has a probability measure of at least $\kappa\varsigma^d/2^d$. Then, when we sample $n_t$ internal test data, each of the $B(\omega'_i, \frac{\varsigma}{2})$s has at least one sampled element with probability at least $1 - (1 - \kappa\varsigma^d/2^d)^{n_t}$. By the union bound, for probability at least $1 - n_h(1 - \kappa\varsigma^d/2^d)^{n_t}$, all open balls in the subcover have at least one internal test data. Considering that the radius of the open balls is $\varsigma/2$, all elements in the subcover are subsets of $\bigcup_{i=1}^{n_t} B(t_i, \varsigma)$, and thus, this internal test data is $\varsigma$-coverage.

Now, we can compute the expected number of the internal test data to be $\varsigma$-coverage as follows (with a constant $A$).

$$
\begin{aligned}
N_{reqit} &\leq \sum_{n_t} n_t Pr_{n_t}(\{t_1, \ldots, t_{n_t}\}: \varsigma\text{-coverage}, \{t_1, \ldots, t_{N_{t-1}}\}: \text{not } \varsigma\text{-coverage}) \\
&= \sum_{n_t} Pr_{n_t}(\{t_1, \ldots, t_{n_t}\}: \text{not } \varsigma\text{-coverage}) \\
&= \sum_{n_t} n_h \left(1 - \frac{\kappa\varsigma^d}{2^d}\right)^{n_t} \\
&= \frac{2^d n^h}{\kappa\varsigma^d} \\
&\leq \frac{16^d n^h}{\kappa^2\varsigma^{2d}} \\
&= \frac{16^d}{\kappa^2 \min(r_s, r_{us}, \frac{pP^*(S)}{k_{us}})^{2d}} \\
&\leq Ap^{-2d}
\end{aligned}
\tag{142}
$$

□

# G    BIAS CORRECTION TO TAILOR TO USER-GIVEN THRESHOLD IN UTILIZATION

Our framework enables users to set or change a threshold that differs from the one used during training. For this purpose, in this section, we present how our framework can handle different thresholds than the one used in training, by adding a term named 'bias' in the model output logit.

In the utilization (inference) stage, the computed posterior of outputs $p^{\xi}(\mathbf{s}_{cr} = \bar{\mathbf{s}}_i | \mathbf{o}_{cr} = \bar{\mathbf{o}}_j)$ is discrete since the number of possible constraint-related environment states and outputs is finite by assumption. If we fix the perception procedure, the system based on our framework can only tackle user-given thresholds in a step-like manner. For instance, if the threshold is lower than $\min_{i,j} p^{\xi}(\mathbf{s}_{cr} = \bar{\mathbf{s}}_i | \mathbf{o}_{cr} = \bar{\mathbf{o}}_j)$, we should satisfy all constraints included in chance-constraints. This is inefficient because we cannot smoothly adjust the system according to the user-given threshold.

Alternatively, in the utilization stage, we modify our information processing module by adding a constant vector $\mathbf{v}$ to the final layer output (*i.e.,* the logit value before post-processing steps):

$$f'(\mathbf{y}; \mathbf{w}_{cr}) = f(\mathbf{y}; \mathbf{w}_{cr}) + \mathbf{v} \tag{143}$$

Then, we can treat $f'(\mathbf{y}; \mathbf{w}_{cr})$ as the final layer output of a new information processing module and obtain new outputs $\mathbf{o}'_{cr}$. Note that $\mathbf{v}$ can be any constant vector that has the same dimension as $f(\mathbf{y}; \mathbf{w}_{cr})$. The whole theory we discuss through this document holds for any $f'(\mathbf{y}; \mathbf{w}_{cr})$ and $\mathbf{o}'_{cr}$. Thus, we may adjust the information processing module output to obtain better performance.

At the beginning of the utilization stage, we run inference for the internal test data first. Then, we can find $\mathbf{v}$ that makes the computed posterior $p^{\xi}(\mathbf{s}_{cr} = \bar{\mathbf{s}}_i | \mathbf{o}'_{cr} = \bar{\mathbf{o}}_j)$ be desired values. For example, when there is one environment state $\bar{\mathbf{s}}$ and the output is $\bar{\mathbf{o}}_1$, we can set $\mathbf{v}$ that makes $p^{\xi}(\mathbf{s}_{cr} = \bar{\mathbf{s}} | \mathbf{o}'_{cr} = \bar{\mathbf{o}}_1)$ the same as the threshold $r_t$ (if such $\mathbf{v}$ exists) to avoid the constraint associated with $\bar{\mathbf{s}}$ as much as possible through $\bar{\mathbf{o}}_1$. We compute $\mathbf{v}$ whenever a different threshold is assigned during utilization, thereby adjusting our model according to the threshold given by the user.

Implementation details we used for this process are provided in each experiment (Section H, I, J).

## H  PRODUCTION PLANNING WITH DEMAND PREDICTION

As a first example, we apply our framework to production planning with demand prediction. We address the combined challenging problem of optimizing production decisions based on predicted demand. In this example, we use the OMEN HP 45L Gaming Desktop GT22-3000t PC equipped with Intel Core Ultra9 285K and NVIDIA GeForce RTX 4090 GPU (personally replaced from NVIDIA GeForce RTX 4070 Super). We use Gurobi (Gurobi Optimization, 2024) for our optimization.

### H.1  PROBLEM SETUP.

We compute production quantities $\mathbf{u} \in \mathbb{R}^4$ based on the demand data for prior $24$ time steps $\mathbf{y}$ that maximize revenue depending on unknown current demand $\mathbf{s} \in \mathbb{R}^4$ while satisfying material constraints and halting production when current demand is low ($s_i < 3$), and otherwise optimized to maximize revenue. Considering that market price is influenced by supply and demand, the revenue is formulated as $\sum_{i=1}^{4}(p_i - k_i(u_i - s_i)) \cdot u_i$ where standard price $(p_1, p_2, p_3, p_4)$ and price sensitivity parameters $(k_1, k_2, k_3, k_4)$ are given constants.

The system faces two types of constraints. First, deterministic material limitations are formulated as $A\mathbf{u} + |\mathbf{u}| \leq \mathbf{b}$, where $A$ and $\mathbf{b}$ are fixed matrices and vectors representing material consumption rates and availability limits. The element-wise absolute value $|\mathbf{u}|$ creates robustness against uncertainties in the consumption rates, handling worst-case scenarios where actual consumption might deviate from nominal values $A$. This robust formulation can be expressed as a second-order cone constraint. Second, we introduce uncertain constraints: when the demand for the $i$-th product is too low ($s_i < 3$, where demand is normalized to $[0, 10]$), production should cease ($u_i = 0$) due to inefficient distribution networks.

### H.2  DATA PREPARATION

We obtain New York regional hourly electricity demand data for 2020-2023 from U.S. Energy Information Administration (Administration, 2020-2023) and New York regional global horizontal irradiance (GHI), relative humidity, and temperature data for 2020-2023 from the National Solar Radiation Data Base (NSRDB) of the National Renewable Energy Laboratory (NREL) (Sengupta et al., 2018). Then, we normalize the data to $[0, 10]$. We set the demand of the four products as normalized values by assuming the products are related to weather or electricity. Thus, we use demand series for the four products, which are normalized to $[0, 10]$, as our ground truth.

We use data for 2020 as our training data, data for 2021 as our internal test data for training, data for 2022 as our internal test data for validation, and data for 2023 as our validation data. (Thus, using a separate dataset for internal test data for training and validation, since the number of internal test data is too small to completely overcome data leakage by conservative testing with a reasonable $\xi$.) Due to the limited number of data, we cannot run a scaling-law experiment for this application.

### H.3  IMPLEMENTED METHODS

For all methods except the mean-variance method, we use the following flow: For each product, we adopt a 1-layer LSTM (Hochreiter & Schmidhuber, 1997) model with hidden layer size $64$ and a subsequent fully-connected layer with output size depending on the method. We put demand data for $24$ prior time steps as the input $\mathbf{y}$. Then, we process the final outputs for $4$ products and compute the production decision for them. In the training phase, we compute the gradient to improve the revenue and back-propagate the networks. In back propagation, we use the Adam optimizer (Kingma & Ba, 2015).

Details for each method are elaborated in each sub-section.

### H.3.1  PROPOSED METHOD

Figure 5 presents the structure of the method on the proposed framework. In this example, for each product, we combine the AI model and the safety classification model, and predict demand and whether demand is less than 3 by LSTM, based on the demand series for the last 24 time steps. Our LSTM-based classifier for $i$-th product has 3 dimensional output whose first element is estimate of

demand $o_{ncr,i}$ ($o_{ncr} = (o_{ncr,1}, o_{ncr,2}, o_{ncr,3}, o_{ncr,4})$ is an estimate of $s = (s_1, s_2, s_3, s_4)$ and other two elements for binary classifier for whether $s_i < 3$ to obtain $o_{cr} = (o_{cr,1}, o_{cr,2}, o_{cr,3}, o_{cr,4}) \in \{0,1\}^4$ when $o_{cr,i} = 1$ indicates $s_i < 3$. Separation of the estimation part and classification part is helpful for tailoring to each threshold because the estimated value does not indicate the probability of the demand to be lower than 3. Then, we put $u_i = 0$ or $u_i \geq 0$ when the (upper-bound of) posterior probability of $s_i < 3$ conditional to the (conservatively tested) classification part output is larger or smaller than the user-given threshold, respectively. Finally, we solve the optimization problem and obtain the optimal action.

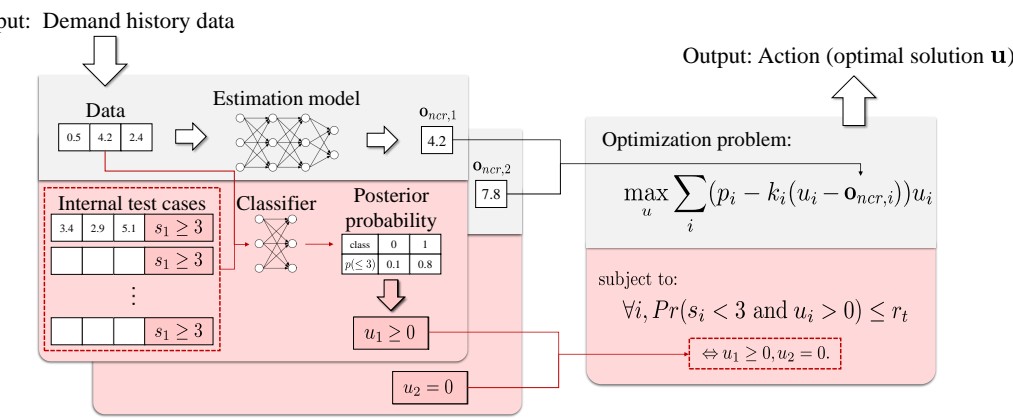

Figure 5: System structure based on the proposed framework for demand prediction-based production decision.

The problem contains both a deterministic material constraint and an uncertain constraint. In particular, the uncertain constraint encodes that when the demand is low ($s_i < 3$), production should cease ($u_i = 0$). We limit the probability of violating this constraint below a user-specified threshold $r_t$. We set $\mathbf{u} = 0$ when the constraint is unsatisfied, and otherwise maximize revenue $\sum_{i=1}^{4}(p_i - k_i(u_i - s_i)) \cdot u_i$, where standard price $p_i$ and price sensitivity $k_i$ are constants.

Now, the problem can be formulated as follows.

$$\min_{\mathbf{u} \in [0,10]^4} \quad \bar{J}(\mathbf{u}; \mathbf{o_{ncr}}) := -\sum_{i=1}^{4}(p_i - k_i(u_i - o_{ncr,i}))u_i \tag{144a}$$

$$c_i := r_t - Pr(s_i < 3 \text{ and } u_i > 0 | o_{cr,i}) \geq 0, i = 1, 2, 3, 4 \tag{144b}$$

$$(c_5, \ldots, c_9) := \bar{A}\mathbf{u} + \mathbf{1}|u| \leq \mathbf{b} \tag{144c}$$

Note that demand state $\mathbf{s}$ is unknown and should be predicted by last 24 time steps of demand data and other parameters $\mathbf{p}, \mathbf{k}, \bar{A}, \mathbf{b}, r_t$ are generated randomly based on a seed and fixed in one experiment (we run 100 experiments with varying seeds).

At the beginning of each epoch, we run the forward call for the model (LSTM+fc network) and obtain the 3-dimensional outputs for each product. Note that we use full batch (8760 data at once) and the posterior is only computed once since it depends only on the internal test data (shared within a batch). Then we construct the `normal table`, which is useful for calculating the posterior probability of the current demand to be lower than 3 for each product. The table is structured as *(number of labels: 2) × (number of classes: 2) × 2 (+$\xi$ and −$\xi$)*. For $i$-th label and $j$-th class, the corresponding entries for each +$\xi$ axis and −$\xi$ axis indicate $N^{+\xi}_{\mathbf{s}_{cr}=\bar{\mathbf{s}}_i, o_{cr}=\bar{\mathbf{o}}_j}$ and $N^{-\xi}_{\mathbf{s}_{cr}=\bar{\mathbf{s}}_i, o_{cr}=\bar{\mathbf{o}}_j}$, respectively. The following procedure illustrates how the table is updated for each internal test data during implementation[23]:

1. For $j$-th class, if adding $\xi$ to the logit of a class results in that class having the highest value among all logits, the corresponding label × $j$-th class × +$\xi$ entry is incremented by 1.

---

[23]Note that $\xi$ is inflated by $\sqrt{2}$ in experiment settings for practical reasons.

2. For $j$-th class, if subtracting $\xi$ results in that class having the highest value among all logits, the corresponding label $\times$ $j$-th class $\times$ $-\xi$ entry is incremented by 1.

After processing all internal test data, we calculate the posterior probabilities based on the completed normal table. The following procedure illustrates the computation of the posterior probability for the $j$-th class:

1. The denominator is calculated using the equation below. The value $N^{-\xi}_{\mathbf{s}_{cr}=\bar{\mathbf{s}}_k,\mathbf{o}_{cr}=\bar{\mathbf{o}}_j}$ is precomputed and stored in the $k$-th label $\times$ $j$-th class $\times$ $-\xi$ entry of the normal table.

$$\sum_k p^{-\xi}(\mathbf{o}_{cr}=\bar{\mathbf{o}}_j|\mathbf{s}_{cr}=\bar{\mathbf{s}}_k) \times \text{prior} = \sum_k \frac{N^{-\xi}_{\mathbf{s}_{cr}=\bar{\mathbf{s}}_k,\mathbf{o}_{cr}=\bar{\mathbf{o}}_j}}{N_{\mathbf{s}_{cr}=\bar{\mathbf{s}}_k}} \times \text{prior} \tag{145}$$

2. The numerator of the $i$-th label posterior probability is implemented based on the following equation. Similar to denominator calculation, we retreive $N^{+\xi}_{\mathbf{s}_{cr}=\bar{\mathbf{s}}_i,\mathbf{o}_{cr}=\bar{\mathbf{o}}_j}$ from $i$-th label $\times$ $j$-th class $\times$ $+\xi$ entry in the normal table.

$$\text{(For $i$-th label)} \quad p^{+\xi}(\mathbf{o}_{cr}=\bar{\mathbf{o}}_j|\mathbf{s}_{cr}=\bar{\mathbf{s}}_i) \times \text{prior} = \frac{N^{+\xi}_{\mathbf{s}_{cr}=\bar{\mathbf{s}}_i,\mathbf{o}_{cr}=\bar{\mathbf{o}}_j}}{N_{\mathbf{s}_{cr}=\bar{\mathbf{s}}_i}} \times \text{prior} \tag{146}$$

Note that we set the prior as the portion of the demand under 3 of the data for $2020-2023$. Moreover, for efficient training, we schedule $\xi$ linearly, starting from 0 and progressively increasing it to the desired value throughout training. The computed posterior probabilities are stored in an array with the structure *(number of classes) $\times$ (number of labels)*.

Then, we compute the posterior as follows:

$$p^\xi(\mathbf{s}_{cr,i}=1|\mathbf{o}_{cr,i}=o) = \frac{\frac{N^{+\xi}_{\mathbf{s}_{cr,i}=1,\mathbf{o}_{cr,i}=o}}{N_{\mathbf{s}_{cr,i}=1}} \times \text{prior}(\mathbf{s}_{cr,i}=1)}{\frac{N^{-\xi}_{\mathbf{s}_{cr,i}=0,\mathbf{o}_{cr,i}=o}}{N_{\mathbf{s}_{cr,i}=0}} \times \text{prior}(\mathbf{s}_{cr,i}=0) + \frac{N^{-\xi}_{\mathbf{s}_{cr,i}=1,\mathbf{o}_{cr,i}=o}}{N_{\mathbf{s}_{cr,i}=1}} \times \text{prior}(\mathbf{s}_{cr,i}=1)} \tag{147}$$

Next, we compute the action (production decision) and the approximated loss function. In principle (and in validation), we allow production of a product $u_i > 0$ only if the posterior is less than or equal to the threshold ($p^\xi(\mathbf{s}_{cr,i}=1|\mathbf{o}_{cr,i}=o) \leq r_{t,i}$). However, in training, to expedite training of the $\mathbf{o}_{ncr}$ part by ensuring nontrivial action to be obtained, we arbitrarily set our virtual threshold as the lower posterior between two output classes, and thus, produce a product if it is classified as safer (low risk for $s_i < 3$) class. Instead, to reduce the posterior of the safer class, we append an additional term to our loss, which is naturally defined as the negative of the total revenue, penalizing the posterior of class 0 (in this example, to reduce the instability, we arbitrary set class 0 to be induced to be the safer class) higher than the given threshold. Moreover, we also add a small term to guide $\mathbf{o}_{ncr}$ (implemented directly in the gradient calculation code). Thus, our loss function is finally defined as ($\beta = 1000$)

$$-\sum_{i=1}^4 (p_i - k_i(u_i - s_i))u_i + \sum_{i=1}^4 \beta \times \log\left(\max(\frac{p^\xi(s_{cr,i}=1|o_{cr,1}=0)}{r_{t,i}},1)\right) + 0.00005\sum_{i=1}^4 (o_{ncr,i}-s_i)^2. \tag{148}$$

Based on this loss function, the approximate loss function is obtained and implemented as follows ($\lambda = 0.005$, note that the latter two terms are not functions of the optimal action $\mathbf{u}$):

$$\tilde{L} = \frac{(\bar{J} - \lambda\sum_{i=1}^4 (p_i - k_i(u_i - s_i))u_i)^* - \bar{J}^*}{\lambda} + \sum_{i=1}^4 \beta \times \log\left(\max(\frac{p^\xi(s_{cr,i}=1|o_{cr,1}=0)}{r_{t,i}},1)\right)$$
$$+ 0.00005\sum_{i=1}^4 (o_{ncr,i}-s_i)^2 \tag{149}$$

Since there are 8760 data (hours in a year) in the training dataset, we use the final loss and revenue value as the sum of all hours. Note that we use variable name $i$ for the hour and variable name $l$ for products, apart from the notation in this document.

We calculate the gradient for the model based on the approximated loss value. This process is divided into three parts: the gradient for the $\mathbf{o}_{ncr}$, the gradient for the $\mathbf{o}_{cr}$, and the gradient for internal test data $\mathbf{o}_{cr}^{t_k}$. We can compute the first one as

$$\frac{\partial \tilde{L}}{\partial o_{ncr,i}} = -k_i(\tilde{u}_i^* - u_i^*) + 0.0001(s_i - o_{ncr,i}) \tag{150}$$

when $\tilde{u}_i^*$ and $u_i^*$ are the minimizer of $(\bar{J} - \lambda \sum_{i=1}^4 (p_i - k_i(u_i - s_i))u_i)$ and $\bar{J}$, respectively. The virtual partial derivative with respect to the logit for the classification $f(\mathbf{y}_i; \mathbf{w}_i)$ is defined as follows:

$$VPD_{cr,i} := \sum_{o_{cr,i}} \frac{\partial p(o_{cr,i}; f(\mathbf{y}_i; \mathbf{w}_i))}{\partial f(\mathbf{y}_i; \mathbf{w}_i)} \tilde{L}(o_{cr,i}, \mathbf{o}_{d-i}; \mathbf{s}, \mathbf{r}, \boldsymbol{\beta}, \lambda) \tag{151}$$

To compute this, we assign all possible classes to the $i$-th output and calculate the approximated loss value, $\tilde{L}(o_{cr,i}, \mathbf{o}_{d-i}; \mathbf{s}, \mathbf{r}, \boldsymbol{\beta}, \lambda)$, using pre-computed posterior probabilities. These values are stored in an array structured as (number of dense action candidates) $\times$ (number of classes). The final result is obtained by multiplying the approximated loss values by the partial derivatives of the softmax function for each class.

Next, for the internal test data, we calculate gradients by modifying entries in the normal table. More specifically, we generate new normal tables by modifying the $i$-th label $\times$ $j$-th class $\times$ $(+\xi$ or $-\xi)$ entry. The following describes the new normal tables and the corresponding approximated loss values used in the implementation:

- `plusoneone_approxloss[i][j]`: Approximated loss based on a new normal table where 1 is added to both the $(i, j, -\xi)$ and $(i, j, +\xi)$ entries.

- `plusone_minusxi_approxloss[i][j]`: Approximated loss based on a new normal table where 1 is added only to the $(i, j, -\xi)$ entry.

- `plusone_plusxi_approxloss[i][j]`: Approximated loss based on a new normal table where 1 is added only to the $(i, j, +\xi)$ entry.

- `minusoneone_approxloss[i][j]`: Approximated loss based on a new normal table where 1 is subtracted from both the $(i, j, -\xi)$ and $(i, j, +\xi)$ entries. If either entry is non-positive in the original table, the approximated loss remains unchanged.

- `minusone_minusxi_approxloss[i][j]`: Approximated loss based on a new normal table where 1 is subtracted from the $(i, j, -\xi)$ entry. If the entry is non-positive in the original table, the approximated loss remains unchanged.

- `minusone_plusxi_approxloss[i][j]`: Approximated loss based on a new normal table where 1 is subtracted from the $(i, j, +\xi)$ entry. If the entry is non-positive in the original table, the approximated loss remains unchanged.

Then, based on those approximated values, we can calculate the gradients for the $j$-th internal test data by multiplying the partial derivative of the softmax function. Since we use a full batch and the posterior does not vary over data within a batch, we compute this value for only the first data in the batch. Moreover, due to the large number of internal test data, we divide the gradient by the number of internal test data.

The loss function for the classification network is computed through element-wise multiplication of the optimization phase gradients (clipped into between $-10^6$ and $10^6$) and the logits. Additionally, a regularization term, which is 10% of the sum of the squared logits, is added to the final loss.

In validation of the proposed method, since we train the proposed method with only one threshold and validate it with various thresholds, we introduce a bias (see Section G for theoretical details)

by adding or subtracting a specific value to or from the logit output of the classification network, aligning the network output with the desired threshold. The following procedure outlines the abstract process for computing the bias corresponding to each $r_t$.

1. Identify the safe class as the one that minimizes the posterior probability for the unsafe label, and record its corresponding posterior value.

2. If the posterior value is smaller than $r_t$, add an appropriate unit value to the logit of the safe class for all internal test data. Otherwise, subtract an appropriate unit value from the logits.

3. Recalculate the posterior probability. If the updated posterior still deviates significantly from $r_t$, repeat step 2.

4. Save the final bias value.

Next, we generate the logits from the classification network by inputting the concatenation of the current observation and dense action candidates. Note that, during validation, the final bias value is added to or subtracted from the logit corresponding to the safe class.

### H.3.2 MEAN_VAR

Let $\mu_i$ and $\sigma_i$ be the mean and the standard deviation for the demand data of the last 24 time steps, respectively. Then, we solve the following problem to obtain our action (production decision).

$$\min_{\mathbf{u} \in [0,10]^4} \quad \bar{J}(\mathbf{u}; \mu) := -\sum_{i=1}^{4} (p_i - k_i(u_i - \mu_i))u_i \tag{152a}$$

$$c_i := \left( \begin{cases} 0, & \text{if } \mu_i - r_{t,i}\sigma_i < 3 \\ \infty, & \text{otherwise} \end{cases} \right) - u_i \geq 0, i = 1, 2, 3, 4 \tag{152b}$$

$$(c_5, \ldots, c_9) := \bar{A}\mathbf{u} + \mathbf{1}|u| \leq \mathbf{b} \tag{152c}$$

Note that $r_{t,i}$ in this method is merely a coefficient for the standard deviation rather than a threshold. Due to the deterministic nature of this method, there is no training phase required.

### H.3.3 TWOSTAGE

In this method, for each product, we obtain a 1 one-dimensional output, $o_{ncr,i}$, which is the estimated demand, from the model. Then, we compute our production decision based on the estimated demand by solving:

$$\min_{\mathbf{u} \in [0,10]^4} \quad \bar{J}(\mathbf{u}; \mathbf{o_{ncr}}) := -\sum_{i=1}^{4} (p_i - k_i(u_i - o_{ncr,i}))u_i \tag{153a}$$

$$c_i := \left( \begin{cases} 0, & \text{if } o_{ncr,i} < 3 \\ \infty, & \text{otherwise} \end{cases} \right) - u_i \geq 0, i = 1, 2, 3, 4 \tag{153b}$$

$$(c_5, \ldots, c_9) := \bar{A}\mathbf{u} + \mathbf{1}|u| \leq \mathbf{b}. \tag{153c}$$

This is the traditional method, which trains the estimator first and then runs optimization based on the estimation. In the training phase, we train the model with the following two-sided loss and gradient (We penalize the overestimation more according to the user parameter ($r_{t,i}$, not threshold) to reduce the constraint violation):

$$\tilde{L} := 0.5(s_i - o_{ncr,i})^2 \times \left( \begin{cases} 1, & \text{if } o_{ncr,i} < s_i \\ 1 + r_{t,i}, & \text{otherwise} \end{cases} \right) \tag{154}$$

$$\frac{\partial \tilde{L}}{\partial o_{ncr,i}} := (s_i - o_{ncr,i}) \times \left( \begin{cases} 1, & \text{if } o_{ncr,i} < s_i \\ 1 + r_{t,i}, & \text{otherwise} \end{cases} \right) \tag{155}$$

### H.3.4 END-TO-END

In this method, for each product, we directly obtain a 1 one-dimensional output to use as our final production decision, $u_i := o_{ncr,i}$, from the model. In the training phase, we train the model with the following loss (user parameter $r_{t,i}$ is not the threshold but used to penalize the violation of the uncertain constraint)

$$
\tilde{L} := L := \bar{J}(\mathbf{u}, \mathbf{s}) + \beta \left( \sum_i \begin{cases} u_i^2, & \text{if } u_i < 0 \\ 0, & \text{otherwise} \end{cases} + \sum_j (\bar{A}_j \mathbf{u} + |u| - b_j)^+ \right) \\
+ \sum_i r_{t,i} \left( \begin{cases} u_i^2, & \text{if } s_i < 3 \text{ and } u_i > 0 \\ 0, & \text{otherwise} \end{cases} \right)
$$
(156)

We compute the gradient with respect to $\mathbf{u}$ ($= \mathbf{o}_{ncr}$) by computing the difference as follows ($\lambda = 0.005$):

$$
\frac{\partial L}{\partial u_i} \simeq \frac{(L(u_i + \lambda) - L(u_i))}{\lambda}
$$
(157)

### H.4 CODE STRUCTURE AND EXECUTION

We implement each method as a C++ project with a Python API (Foundation, 2024), primarily defined in `User_api.h` and `User_api.cpp`. Through these extensions, the project is linked to the neural network components and several supporting Python functions defined in `NN_function.py`.

We implement data pre-processing ("or_data"), parallel execution ("or_run"), and result post-processing ("or_results") code as separate projects, along with the projects for each method. We run 100 experiments with different random seeds through the execution code and report the average and standard error of the results through the result postprocessing code. In each experiment, we run the methods with various $r_t$ values (we use the same $r_{t,i}$ for all products, and thus, call it $r_t$.) presented in the following table. We train each method (except Mean_Var) for 500 epochs and validate it.

Table 1: Table of $r_t$ values used in the product planning experiment.

| Method | Proposed | | Mean_Var | | Twostage | | End-to-end | |
|--------|----------|-----|----------|-----|----------|-----|------------|-----|
| Phase | Train | Val | Train | Val | Train | Val | Train | Val |
| $r_t$ | 0.001 | 1.0 | – | 10 | 0 | 0 | 0 | 0 |
| | | 0.5 | | 1 | 1 | 1 | 1 | 1 |
| | | 0.2 | | 0 | 3 | 3 | 3 | 3 |
| | | 0.1 | | -0.3 | 10 | 10 | 10 | 10 |
| | | 0.05 | | -0.6 | 30 | 30 | 30 | 30 |
| | | 0.02 | | -0.9 | 100 | 100 | 100 | 100 |
| | | 0.01 | | -1.2 | 300 | 300 | 300 | 300 |
| | | 0.005 | | -1.5 | 1000 | 1000 | 1000 | 1000 |
| | | 0.002 | | -1.8 | 3000 | 3000 | 3000 | 3000 |
| | | 0.001 | | -2.1 | 10000 | 10000 | 10000 | 10000 |

## H.5 RESULTS

Figure 6-A illustrates the performance versus constraint violation trade-off. We use $r_t = 0.001$ for training and $0.001 - 1.0$ for validation. Our method achieves significantly higher revenue than baseline approaches, particularly at low violation percentages. Figure 6-B shows that the constraint violation percentage is lower than the threshold $r_t$, confirming our safety guarantee.

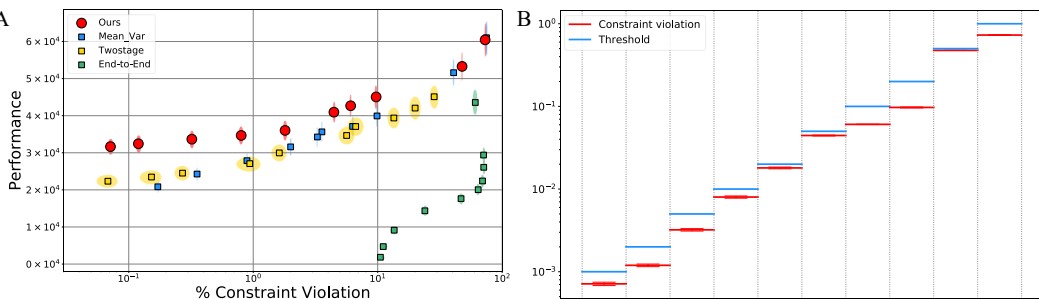

Figure 6: **Production Planning.** (A) Our method achieves significantly higher venue than baselines. $x$-axis is the percentage of constraint violation cases where production continues despite low demands, and $y$-axis is total revenue. (B) Our method achieves constraint violations lower than the designated thresholds.

## I   Reinforcement learning in Safetygym

We use the `ASUS ESC8000A-E12` equipped with two `AMD EPYC 9554 64-core` processors for the whole procedure in this application.

### I.1   Environment

We use the Safety Gym environments (Ray et al., 2019) to evaluate a goal task (`Safexp-PointGoal1-v0`), where the robot's primary objective is to navigate toward a series of goal positions while avoiding hazards. All default settings (OpenAI, 2019b;a) are retained, except that the environment resets when the robot enters a hazard zone that incurs a cost of 1, as this is treated as a failure. `Point` robots receive a 60-dimensional vector as the observation from the environment and have a two-dimensional action with range $[-1, 1]$: one is the force applied to translational motion, and the other is the rotation velocity.

### I.2   Pretraining PPO agents

We use the provided code (Jayant, 2022) to pre-train the PPO agent and the PPO-Lagrangian agent in our experiments. Additionally, we modify the provided code to implement the PPO-Barrier agent (Yang et al., 2023) by referencing the original implementation (Yang, 2023). Each agent is trained using 30 CPU cores with parallelization via mpi4py, as implemented in the provided code. Each core executes $10^3$ steps per epoch, leading to a total of $3 \cdot 10^4$ steps per epoch. For our framework, we use $10^4$-epoch checkpoints as pretrained agents. Note that both $10^4$-epoch checkpoints and $3 \cdot 10^4$-epoch checkpoints are used as baselines for each agent.

### I.3   Collecting the internal test data

Before training our framework, we collect the internal test data for each agent using the $10^4$-epoch checkpoints. One important modification to the original checkpoint is that we reset the standard deviation of the pretrained policy. This is because pretraining for $10^4$ epochs often results in an overly small standard deviation in the action distribution, reducing its plasticity and adaptability for our framework. Therefore, the standard deviation is manually set to $\exp(-2)$ at the start of framework training. Consequently, internal test data are also collected using PPO agents with this adjusted standard deviation.

Each internal test data is composed of a concatenated observation, action, and safety label. A safety label is assigned as 0 for safe cases and 1 for unsafe cases. Unsafe cases occur when a failure arises during 60 steps. Otherwise, the case is labeled as safe. To avoid biased data collection, internal test data are collected every 10 steps.

A total of $10^7$ internal test data is collected per pretraining checkpoint: $5 \cdot 10^6$ for safe cases and $5 \cdot 10^6$ for unsafe cases. This dataset is used when initiating training for our framework. Once training of our framework begins, new internal test data are queued to reflect the updated policy of each agent. The total number of internal test data remains fixed at $10^7$, with older data being replaced by new entries.

Collection of internal test data for PPO includes $3 - 4 \cdot 10^8$ total environment interactions. For PPO-Lagrangian, it includes $4 - 5 \cdot 10^8$ total environment interactions. Thus, internal test data collection has a similar number of environment interactions with $10^4 - 1.7 \cdot 10^4$ additional epochs with PPO or PPO-Lagrangian training. This suggests the PPO and PPO-Lagrangian agents that are trained for $3 \cdot 10^4$ epochs are a fair comparison with the proposed method from the perspective of data usage.

### I.4   Training

The main training is carried out by multiple `Python` processes using `PyTorch`. Each process performs $10^3$ steps per epoch, and with 30 CPU cores running in parallel, this results in a total of $3 \cdot 10^4$ steps per epoch. The training runs for $10^3$ epochs in total. The training process of a single step consists of three phases:

1. Data batch preparation (`Python`)

2. Conservative testing & Optimization phase (`C++`)

3. Executing final action and Training (`Python`)

`Python` and `C++` processes communicate via shared memory, with semaphores used to prevent race conditions. Both shared memory and semaphores are implemented using `sysv_ipc`.

A $10^4$ epoch pretrained PPO agent is loaded at the beginning of the training. Only the standard deviation of the actor network is reset to a predefined value. Additionally, our framework employs a fully connected classification network with a single hidden layer consisting of 128 input nodes and 128 output nodes. Table 2 provides a detailed description of the classifier architecture and parameters. Note that both the pretraining PPO agents and the classification network are synchronized across all 30 `Python` processes.

Table 2: Detailed parameters for the classifier

| Section | Implementation | Parameter | Value |
|---------|----------------|-----------|-------|
| Classifier layer 1 | `torch.nn.Linear` | `in_features` | `observation_dim` |
| | | `out_features` | 128 |
| Layer 1 activation | `torch.nn.functional.relu` | – | – |
| Classifier layer 2 | `torch.nn.Linear` | `in_features` | 128 |
| | | `out_features` | 128 |
| Layer 2 activation | `torch.nn.functional.relu` | – | – |
| Classifier layer 3 | `torch.nn.Linear` | `in_features` | 128 |
| | | `out_features` | `class_dim` |
| Optimizer | `torch.optim.AdamW` | `lr` | 1e-4 |
| | | `weight_decay` | 1e-6 |
| | | `amsgrad` | True |
| Learning Rate Scheduler | `torch.optim.lr_scheduler` `.LinearLR` | `start_factor` | 1 |
| | | `end_factor` | 1e-3 |
| | | `total_iters` | 17100000 |
| Scheduler with Warmup | `ignite.handlers` `.param_scheduler` | `warmup_start_value` | 0 |
| | | `warmup_duration` | 900000 |

Internal test data are loaded into shared memory via `mpi4py`, enabling all `Python` processes to access the same data. As the new internal test data are generated, they are queued in shared memory, and the index of this queue is continuously updated in an additional shared memory.

### I.4.1 DATA BATCH PREPARATION

At the beginning of each epoch, the environment provides an observation (*i.e.*, the current state). We extract the mean and standard deviation of the action distribution based on the given observation. The action will be determined later during the optimization phase.

The data batch for the next step includes not only the mean and standard deviation but also additional components derived from the observation. We extract two components from the observation: a 16-dimensional vector representing the robot's proximity to hazards and a 1-dimensional vector indicating the robot's translation velocity. The proximity vector helps determine whether the robot is near a hazard. If the robot seems to be in a hazard zone, a *default action* is assigned to guide it out of the cost zone using information from the velocity component.

To complete the data batch, we also require logits generated by the classification network. To generate these logits, we prepare action candidates that appropriately cover the action space by discretizing each dimension within the range $[-3, 3]$ at intervals of $1.6 \cdot 10^{-2}$. We refer to these as *base action candidates*. However, using base candidates in every phase of training can slow down the overall training process. One major reason is that the classification network needs to generate logits for all action candidates with a current observation, and the gradients for these logits must be computed and used to update the network. To address this computational burden, we introduce *dense action candidates*. For action dimensions measured in velocity units, we discretize the range $[-1, 1]$ at intervals of $1.6 \cdot 10^{-2}$, just considering the possible actions on our environment. For dimensions measured in force units, we simplify discretization by using only the values $\{-1, 1\}$. By adopting

dense action candidates, we maintain the essential characteristics of base action candidates while minimizing gradient calculations and other costly computations.

### I.4.2 CONSERVATIVE TESTING & OPTIMIZATION PHASE

This part begins by retrieving the final data batch from shared memory. This batch contains key elements such as the mean, standard deviation, classifier logits, and safety labels of the internal test data. As part of pre-processing, all logits are divided by 2 as the parameter for softmax.

The first step in this phase is to construct the *normal table*, which is useful for calculating the posterior probability of each action candidate. The table is structured as *(number of labels) × (number of classes) × 2 (+ξ and −ξ)*. For $i$-th label and $j$-th class, the corresponding entries for each $+\xi$ axis and $-\xi$ axis indicate $N^{+\xi}_{\mathbf{s}_{cr}=\bar{\mathbf{s}}_i, \mathbf{o}_{cr}=\bar{\mathbf{o}}_j}$ and $N^{-\xi}_{\mathbf{s}_{cr}=\bar{\mathbf{s}}_i, \mathbf{o}_{cr}=\bar{\mathbf{o}}_j}$, respectively. The following procedure illustrates how the table is updated for each internal test data during implementation[24]:

1. For $j$-th class, if adding $\xi$ to the logit of a class results in that class having the highest value among all logits, the corresponding label × $j$-th class × $+\xi$ entry is incremented by 1.

2. For $j$-th class, if subtracting $\xi$ results in that class having the highest value among all logits, the corresponding label × $j$-th class × $-\xi$ entry is incremented by 1.

After processing all internal test data, we calculate the posterior probabilities based on the completed normal table. The following procedure illustrates the computation of the posterior probability for the $j$-th class:

1. The denominator is calculated using the equation below. The value $N^{-\xi}_{\mathbf{s}_{cr}=\bar{\mathbf{s}}_k, \mathbf{o}_{cr}=\bar{\mathbf{o}}_j}$ is precomputed and stored in the $k$-th label × $j$-th class × $-\xi$ entry of the normal table.

$$\sum_k p^{-\xi}(\mathbf{o}_{cr} = \bar{\mathbf{o}}_j | \mathbf{s}_{cr} = \bar{\mathbf{s}}_k) \cdot \text{prior} = \sum_k \frac{N^{-\xi}_{\mathbf{s}_{cr}=\bar{\mathbf{s}}_k, \mathbf{o}_{cr}=\bar{\mathbf{o}}_j}}{N_{\mathbf{s}_{cr}=\bar{\mathbf{s}}_k}} \cdot \text{prior} \tag{158}$$

2. The numerator of the $i$-th label posterior probability is implemented based on the following equation. Similar to denominator calculation, we retrieve $N^{+\xi}_{\mathbf{s}_{cr}=\bar{\mathbf{s}}_i, \mathbf{o}_{cr}=\bar{\mathbf{o}}_j}$ from $i$-th label × $j$-th class × $+\xi$ entry in the normal table.

$$\text{(For } i\text{-th label)} \quad p^{+\xi}(\mathbf{o}_{cr} = \bar{\mathbf{o}}_j | \mathbf{s}_{cr} = \bar{\mathbf{s}}_i) \cdot \text{prior} = \frac{N^{+\xi}_{\mathbf{s}_{cr}=\bar{\mathbf{s}}_i, \mathbf{o}_{cr}=\bar{\mathbf{o}}_j}}{N_{\mathbf{s}_{cr}=\bar{\mathbf{s}}_i}} \cdot \text{prior} \tag{159}$$

Note that we set the prior as 0.5 for both the safe label and unsafe label. The computed posterior probabilities are stored in an array with the structure *(number of classes) × (number of labels)*.

Next, we compute the approximated loss function. The procedure begins by calculating the following intermediate value ($\beta$=3):

$$\bar{L} = \beta \cdot \ln(\max(\frac{p^\xi(\mathbf{s}_{cr} = 1 | \mathbf{o}_{cr} = o)}{threshold}, 1)) \tag{160}$$

where $o$ denotes one of the classes assigned by the classifier to the current observation concatenated with the dense action candidates. For efficient training, we schedule $\xi$ linearly, starting from 0 and progressively increasing it to the desired value throughout training. The goal is to select the action whose corresponding class minimizes $\bar{L}$. The final approximated loss function is defined as

$$\tilde{L} = \bar{L} + 5 \cdot p - 0.0015 \cdot n_o \tag{161}$$

where $p$ is the minimum posterior probability for $\mathbf{s}_{cr} = 0$ among all classes and $n_o$ is the number of action candidates associated with class $o$. Note that the term $p$ is included in the final loss to suppress excessive type 2 error, thereby encouraging a broader range of safe actions to be classified into the

---

[24]Note that $\xi$ is inflated by $\sqrt{2}$ in experiment settings for practical reasons.

appropriate class. To prevent large loss values from destabilizing training, we initialize a *default minimum loss* $L_d = 30$ and update it over time using:

$$L_d = 0.99999 \cdot L_d + 0.00001 \cdot (\bar{L} + \frac{30}{\bar{L}+1}) \tag{162}$$

If no class yields a $\bar{L}$ below the current $L_d$, or if the proximity vector indicates that the robot is in a hazard zone, the loss is set to the default minimum loss $L_d$, prompting the robot to perform the default action.

We reintroduce base action candidates to refine the optimization process further. These candidates are used to build a conditional distribution over actions associated with the class that minimizes the loss. Using the mean and standard deviation from the data batch, we calculate the probability of each filtered action and sample one final action from this distribution.

We calculate the gradient for the classification network based on the approximated loss value. This process is divided into two main parts: the gradient for the general output and the gradient for sampled internal test data. We first describe the calculation of the general output gradient, followed by the method used for internal test data.

The classification network generates a general output by feeding a concatenation of the duplicated current observation and dense action candidates as input. Note that the virtual partial derivative with respect to the logit of the concatenation of the current observation and $i$-th action in the candidates, $f(\mathbf{y}_i; \mathbf{w}_i)$, is defined as follows:

$$VPD_{cr,i} := \sum_{o_{cr,i}} \frac{\partial p(o_{cr,i}; f(\mathbf{y}_i; \mathbf{w}_i))}{\partial f(\mathbf{y}_i; \mathbf{w}_i)} \tilde{L} \tag{163}$$

To compute this, we assign all possible classes to the $i$-th output and calculate the approximated loss value $\tilde{L}$ using precomputed posterior probabilities. These values are stored in an array of shape *(number of dense action candidates)* $\times$ *(number of classes)*. The final gradient is obtained by multiplying the approximated loss values by the partial derivatives of the softmax function for each class. This resulting value is then clipped between $-5 \cdot 10^{-4}$ and $5 \cdot 10^{-4}$.

Next, for the sampled internal test data, we calculate gradients by modifying entries in the normal table. More specifically, we generate new normal tables by modifying the $i$-th label $\times$ $j$-th class $\times$ $(+\xi$ or $-\xi)$ entry. The following describes the new normal tables and the corresponding approximated loss values used in the implementation:

- `plusoneone_approxloss[i][j]`: Approximated loss based on a new normal table where 1 is added to both the $(i, j, -\xi)$ and $(i, j, +\xi)$ entries.

- `plusone_minusxi_approxloss[i][j]`: Approximated loss based on a new normal table where 1 is added only to the $(i, j, -\xi)$ entry.

- `plusone_plusxi_approxloss[i][j]`: Approximated loss based on a new normal table where 1 is added only to the $(i, j, +\xi)$ entry.

- `minusoneone_approxloss[i][j]`: Approximated loss based on a new normal table where 1 is subtracted from both the $(i, j, -\xi)$ and $(i, j, +\xi)$ entries. If either entry is non-positive in the original table, the approximated loss remains unchanged.

- `minusone_minusxi_approxloss[i][j]`: Approximated loss based on a new normal table where 1 is subtracted from the $(i, j, -\xi)$ entry. If the entry is non-positive in the original table, the approximated loss remains unchanged.

- `minusone_plusxi_approxloss[i][j]`: Approximated loss based on a new normal table where 1 is subtracted from the $(i, j, +\xi)$ entry. If the entry is non-positive in the original table, the approximated loss remains unchanged.

Then, based on those approximated values, we can calculate the gradients for $j$-th internal test data by multiplying the partial derivative of the softmax function.

Finally, the selected action and the gradients for all classifier logits are written to shared memory and passed to the next phase.

### I.4.3 Executing final action and training

The loss function for the classification network is computed through element-wise multiplication of the optimization phase gradients and the logits. In addition, a regularization term, which is the average of the squared logits, is added to the final loss. Figure 7 illustrates an example of the gradient and logit values for a single training case.

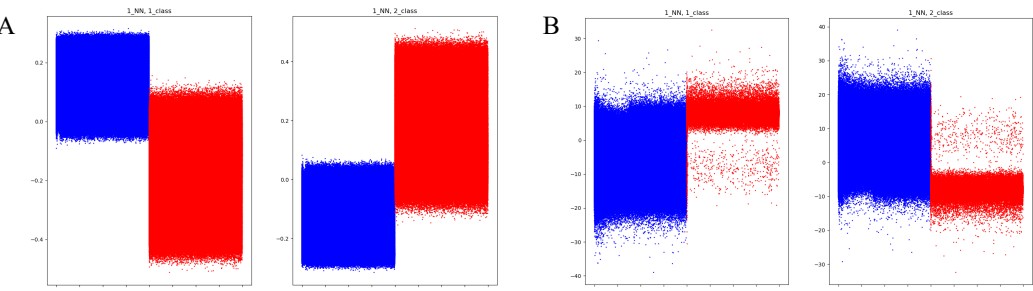

Figure 7: Gradients and logit values for the internal test data. Internal test data labeled as safe are shown in blue, while those labeled as unsafe are shown in red. (A) Initial gradient of the safety classification network for a single training case. (B) Final logit values of the safety classification network for the same training case.

At the end of each step, the environment is updated by executing the final action. This action is also used to update the PPO agents during training.

### I.5 Validation

#### I.5.1 Baseline validation

This section presents the baseline validation process. We begin by loading the checkpoint of pretrained PPO agents in evaluation mode within the default environment. Each agent is evaluated over $10^4$ epochs[25], with $10^3$ steps per epoch, without parallelization. We manually record three metrics: reward, cost, and action.

In addition, using the pretrained PPO agents and a safety classification network trained with cross-entropy loss, we validate our framework to serve as a baseline. We use $\xi = 10^{-2}$ and ten values of the threshold $r_t$: $10^{-5}, 10^{-4}, 10^{-3}, 10^{-2}, 10^{-1}, 0.5 - 10^{-2}, 0.5 - 10^{-3}, 0.5 - 10^{-4}, 0.5 - 10^{-5}$, and $1 + 10^{-5}$. The prior probabilities $(\mathbf{s}_{cr} = 0, \mathbf{s}_{cr} = 1)$ used for this validation are set to $(150, 850)$ for the PPO agent and $(90, 910)$ for the PPO-Lagrangian agent with a cost limit of 1.5. The implementation details closely follow those described in the next section.

#### I.5.2 Validation utilizing our framework

To validate an agent trained by our framework, we load a checkpoint containing the agent, the queued internal test data, and the trained classification network. Similar to the baseline validation, we record reward, cost, and action, while also recording the constraint violation metric. Specifically, we update the numerator and denominator for constraint violation during each epoch and store their ratio. If the robot enters a hazard, we increase the denominator by the number of steps taken after resetting the environment and add up to 60 steps to the numerator, counting only the steps within the previous 60-step trajectory. Note that steps where the robot executes the default action are excluded from both the numerator and the denominator.

---

[25]In few cases in baseline, due to technical issue during the experiment, the number of epochs are slightly lower (between $9 \cdot 10^3$ and $10^4$), but the effect in results is negligible.

Next, we configure the $\xi$ and $r_t$ values. We set $\xi = 10^{-2}$ and use the same ten $r_t$ values as in the baseline validation, which employs a pretrained safety classification model trained with cross-entropy loss. The agent is evaluated over $10^4$ epochs with parallelization, and each `Python` process is executed for a specific $\xi$-$r_t$ pair. As a result, the number of CPU cores running parallel `Python` processes equals the product of the numbers of the two variables. Similar to the training phase in our framework, each `Python` process is linked to a corresponding C++ executable. These processes communicate via shared memory and semaphores using the `sysv_ipc` library. Note that we use the `NVIDIA GeForce RTX 4090` GPU to perform inference on the classification network in this process.

In contrast to the training phase, we construct the normal table and calculate the posterior probabilities only once at the beginning of the validation in the `Python` processes. This is feasible because we use a fixed set of $10^7$ internal test data, and the classification network now produces the same logit for identical inputs. Additionally, we introduce a bias (see Section

Note that we use the same prior probabilities as in the baseline validation, with an additional setting of $(30, 970)$ for the PPO-Lagrangian agent with a cost limit of $0.5$

The validation process of a single step consists of two phases:

1. Data batch preparation (`Python`)

2. Selecting the final action (`C++`) and executing (`Python`)

In the data patch preparation phase, similar to Section I.4.1, we obtain the mean and standard deviation of the action distribution from the PPO agents based on the current observation. We also extract a 16-dimensional hazard proximity vector and a 1-dimensional velocity vector. Next, we generate the logits from the classification network by inputting the concatenation of the current observation and dense action candidates. Note that, during validation, the final bias value is added to or subtracted from the logit corresponding to the safe class. Finally, to complete the data batch, we include the safe class that minimizes the posterior probability for the unsafe label, as determined during the bias calculation process.

In the next step, based on the logits in the data batch, we identify actions that correspond to the safe class among the dense action candidates. Then, using the base action candidates, we build a conditional distribution over the selected actions based on the previously obtained mean and standard deviation. We then sample the final action from this distribution. Note that if no actions match the safe class, or if the proximity vector indicates that the robot is currently in a hazard zone, the default action is selected. The final action is passed to the `Python` process via shared memory and executed by the PPO agent.

### I.6   SCALING LAW

We perform additional training and validation of both the PPO agent and the PPO-Lagrangian agent with a cost limit of $1.5$ to demonstrate the scaling law in this application. Each agent is trained not only with the full set of $10^7$ internal test data, as described in Section I.4, but also with varying amounts of internal test data under the same settings: $10^4$, $2 \cdot 10^4$, $5 \cdot 10^4$, $10^5$, $2 \cdot 10^5$, $5 \cdot 10^5$, $10^6$, $2 \cdot 10^6$, and $5 \cdot 10^6$.

The validation procedure for these agents is nearly identical to that described in Section I.5.2, except that the bias calculation process is omitted. In this experiment, the value of $\xi$ is scaled inversely with the amount of internal test data, and is set to $10^5$ divided by the number of internal test data used in each training. For instance, when validating with $10^7$ internal test data, we use $\xi = 10^{-2}$, which is consistent with the setting used in the previous validation. The results demonstrate the effect of internal test data quantity on performance, as measured by the combined metric of reward maximization and cost minimization, which is presented in the main paper.

## J  Natural language generation

In this section, we elaborate on the experimental setup, data preparation, training procedures, and validation methodology for the natural language generation task presented in the main paper. We have shared code related to the core components of our proposed safety framework, such as the safety classification model training, conservative testing, and the optimization stage. However, the code specifically for the language model training is not publicly released due to license-related considerations. To compensate for this, we believe that the methods for these language model training stages are sufficiently described throughout this supplementary material and the main paper to allow for the reproducibility of the overall experimental approach. In this experiment, we use an NVIDIA H100 Tensor Core GPU with 80GB SXM5 via 1x and 8x GPU instances of Lambda Lab (Labs, 2025).

### J.1  Problem setup

The primary objective of this experiment is to develop and assess a system that guides a large language model (LLM) to generate responses $\mathbf{u}$ that are not only helpful to the user's prompt $\mathbf{y}$ but also adhere to specified harmlessness constraints. The goal is to ensure the probability of producing a harmful response remains below a user-defined threshold, $r_t$. In this experimental setup, the 7B parameter reward model ($R_\phi$) and the 7B parameter cost model ($C_\psi$), which were previously fine-tuned in the SafeRLHF study (Dai et al., 2024), are utilized as proxies for human judgments on helpfulness and harmlessness, respectively. Consequently, these models are considered to provide the ground truth for the helpfulness and harmlessness objectives throughout our experiments.

Our proposed method utilizes pre-trained Open Pre-trained Transformer (OPT) models (Zhang et al., 2022) and the SafeRLHF human preference dataset (Dai et al., 2024). Specifically, OPT-1.3B serves as the base for the policy LLM, and OPT-350M is used for the safety classification model. The core of our framework involves a policy LLM $\pi$, equivalent to $f(\mathbf{y}; \mathbf{w}_{ncr})$ in our framework and based on OPT-1.3B. This policy LLM is fine-tuned using a PPO-Lag (PPO-Lagrangian) algorithm (Ray et al., 2019) following (Zhang et al., 2022) to generate multiple candidate answers ($o_{ncr}$), guided by the aforementioned 7B parameter reward model ($R_\phi$) and 7B parameter cost model from (Dai et al., 2024). A safety classification model, built upon OPT-350M with LoRA (Hu et al., 2022), then predicts the safety ($o_{cr}$) of these candidates and is trained within our framework. An optimization stage selects the final answer based on an objective function while ensuring the estimated probability of harm—determined through conservative testing with $o_{cr}$ and internal test data—is below $r_t$. The ground truth safety labels for the internal test data and for the final validation of responses are also provided by the 7B parameter cost model from (Dai et al., 2024).

To evaluate our framework, we compare it against two main baselines. The first approach involves using the PPO-Lag fine-tuned policy LLM with different safety constraint thresholds, which helps measure the benefit of our framework's additional safety layers compared to a standard constrained generation method. The second baseline is Rejection Sampling, where a conventionally trained safety classifier filters candidates from $\pi$. We utilize the generated internal test data when training the classifier to ensure a fair comparison. This comparison highlights the advantages of our framework's integrated optimization approach versus a simpler filtering mechanism. The performance of all methods is evaluated on "Helpfulness" (Mean Reward from the 7B $R_\phi$) and "Safety" (Unsafe Responses (%) based on the 7B $C_\psi$). These experiments aim to demonstrate our system's ability to effectively balance helpfulness and safety, particularly under strict safety requirements.

### J.2  Data preparation

#### J.2.1  Base Models and Supervised Fine-Tuning (SFT) Data

The experiment began with pre-trained Open Pre-trained Transformer (OPT) models (Zhang et al., 2022) with 350 million (OPT-350M) and 1.3 billion (OPT-1.3B) parameters. These models served as the foundation for subsequent fine-tuning stages. For the initial Supervised Fine-Tuning (SFT), we utilized the Alpaca dataset (Taori et al., 2023) for general instruction following, augmented with safe responses from the SafeRLHF dataset (Dai et al., 2024) to instill baseline safety awareness.

### J.2.2 PREFERENCE DATA FOR REWARD AND COST MODELS

The human preference dataset from the SafeRLHF study (Dai et al., 2024) was instrumental in the prior fine-tuning of the 7B parameter reward model ($R_\phi$) and the 7B parameter cost model ($C_\psi$), which we utilize for PPO-Lag policy training and evaluation.

Specifically, the SafeRLHF dataset includes a helpfulness-related portion, $\mathcal{D}_R = \{x^i, y^i_w, y^i_l\}^N_{i=1}$, where for a given prompt $x$, response $y_w$ is preferred over $y_l$ in terms of helpfulness. It also contains a harmlessness-related portion, $\mathcal{D}_C = \{x^j, y^j_w, y^j_l, s^j_w, s^j_l\}^M_{j=1}$. In $\mathcal{D}_C$, where $y_w$ denotes the more harmful response compared to $y_l$. This dataset further provides binary safety meta-labels $s(y) \in \{+1, -1\}$ for each response, indicating if it is harmful (+1) or harmless (-1), based on 14 predefined categories of harm.

### J.2.3 INTERNAL TEST DATA FOR SAFETY CLASSIFICATION MODEL

The internal test data, essential for training and validating the safety classification model within our framework, was generated through a specific pipeline[26]:

**Prompt Source**: A diverse set of prompts was utilized. These included prompts from the SafeRLHF dataset (Dai et al., 2024) and, additionally, prompts from the BeaverTails project (Ji et al., 2023) were used to generate responses. This combined set was chosen to cover a broad range of topics, including those with the potential to elicit unsafe or problematic responses, thereby ensuring the safety classifier is trained on relevant scenarios.

**Candidate Generation**: The OPT-1.3B model, after its fine-tuning as a policy $\pi$ using PPO-Lag (detailed in Section J.3), was employed to generate 16 diverse answer candidates for each prompt in the selected set. To achieve this, we utilized a diverse beam search strategy (Vijayakumar et al., 2018) where greedy decoding is performed within each beam group, and a diversity penalty is applied between groups. This method aims to produce individually coherent candidates (due to greedy decoding within groups) yet collectively diverse, providing a varied and plausible set of options for the safety classification model and optimization stage, thereby increasing the likelihood of finding an optimal safe and helpful answer. These 16 candidates per prompt constitute the $o_{ncr}$ (and by definition, $s_{ncr}$) part of the internal test data instances. The parameters for diverse beam search are detailed in Table 3.

Table 3: Beam Search Parameters for Candidate Generation

| Parameter | Value |
|---|---|
| num beams | 16 |
| num beams group | 16 |
| diversity penalty | 1.0 |
| repetition penalty | 1.0 |
| max length | 512 |
| no repeat ngram size | 2 |
| early stopping | False |

**Safety Labeling**: Each of the 16 generated candidates was then assigned a safety label ($s_{cr}$) by the 7B parameter cost model from (Dai et al., 2024). This larger cost model, itself trained on extensive human harmlessness preference data, acts as a robust proxy for human safety judgments. The scalar output of this 7B cost model (where positive values indicate higher predicted harm and negative values indicate predicted harmlessness) was thresholded (e.g., at zero) to create binary "harmful" / "harmless" labels for the $s_{cr}$ component of the internal test data. This curated dataset of (prompt, 16 candidates, 16 safety labels) was then partitioned as the internal test data for training and validation phases of the safety classification model within our framework.

---

[26]We use 688,908 internal test data in this experiment

### J.3 TRAINING

The multi-stage training process involved developing the policy LLM and the safety classification model integrated into our framework.

#### J.3.1 INITIAL SFT MODELS

We started with the supervised fine-tuned versions of the OPT-350M (SFT on Alpaca dataset (Taori et al., 2023)) and OPT-1.3B (SFT on Alpaca (Taori et al., 2023) and SafeRLHF datasets (Dai et al., 2024)) as our base LLMs, prepared as described in Section J.2.1.

#### J.3.2 PROVISION OF REWARD AND COST MODELS FROM PRIOR WORK

For guiding the PPO-Lag policy training, we utilized the 7B parameter reward model ($R_\phi$) and cost model ($C_\psi$) that were previously fine-tuned in the SafeRLHF study (Dai et al., 2024).

The 7B $R_\phi$ was fine-tuned in that prior work on the helpfulness preference dataset $\mathcal{D}_R$ (from the SafeRLHF dataset) to assign higher scalar scores to responses humans found more helpful, typically using a pairwise comparison loss (Equation (3) in (Dai et al., 2024)).

The 7B $C_\psi$ was fine-tuned in that prior work on the harmlessness preference dataset $\mathcal{D}_C$ (from the SafeRLHF dataset) using a specialized loss function (Equation (4) in (Dai et al., 2024)) that incorporates both preference rankings for harmlessness and absolute safety labels $s(y)$. This design enables the cost model $C_\psi(y, x)$ to output positive values for harmful responses and negative values for harmless ones, with magnitudes reflecting relative preference. We use these models, as developed in the prior work, directly.

#### J.3.3 POLICY (AI MODEL $\pi$) TRAINING WITH PPO-LAGRANGIAN

The SFT OPT-1.3B model (fine-tuned on Alpaca and SafeRLHF datasets) was further fine-tuned to serve as the policy $\pi$ (also denoted $f(y; w_{ncr})$ in the main paper). This training employed the Proximal Policy Optimization (PPO) algorithm with a Lagrangian method to handle constraints (Ray et al., 2019) following (Dai et al., 2024). For the PPO-Lag algorithm, separate reward and cost critic networks were utilized, both based on OPT-350M models. These critic models were initialized by first training them as reward and cost models, respectively, using the SafeRLHF dataset (Dai et al., 2024) as described in Section J.2.1, before their use as critics in PPO-Lag. The objective was to maximize the expected helpfulness rewards provided by the 7B $R_\phi$ while ensuring that the expected harmlessness costs from the 7B $C_\psi$ remained below a predefined budget $d$. This approach aims to maximize $\mathcal{J}_R(\theta) = \mathbb{E}[R_\phi(y, x)]$ subject to a constraint on the cost $\mathcal{J}_C(\theta) = \mathbb{E}[C_\psi(y, x)] + d \leq 0$. This constrained optimization is typically solved by addressing the Lagrangian dual problem: $\min_\theta \max_{\lambda \geq 0}[-\mathcal{J}_R(\theta) + \lambda \cdot \mathcal{J}_C(\theta)]$. The final output of this trained policy $\pi$ for a given prompt $y$ is a set of 16 candidate answers, referred to as $o_{ncr}$, generated using diverse beam search as described in Section J.2.3.

#### J.3.4 SAFETY CLASSIFICATION MODEL TRAINING (WITHIN OUR FRAMEWORK)

The safety classification model in our framework begins with the SFT OPT-350M architecture (fine-tuned as described in Section J.2.1). The model is then further fine-tuned using Low-Rank Adaptation (LoRA) (Hu et al., 2022) with the tailored gradients from our framework. Using a smaller model (OPT-350M) and then applying LoRA for the framework-specific adaptation helps prevent overfitting and allows for more stable training.

The model takes the original prompt $y$ from an internal test data instance, concatenated with the 16 candidate answers ($o_{ncr}$) also from that same instance, as input. It then outputs $o_{cr,pred} \in \{0, 1\}^{16}$, where each bit represents its binary classification (0 for harmless, 1 for harmful) for the corresponding candidate answer.

This safety classification model's LoRA weights are updated using the internal test data (which contains pre-generated sets of 16 candidates and their true safety labels $s_{cr}$). The policy $\pi$ is not used to generate candidates during this LoRA fine-tuning phase; instead, the candidates are fixed according to the internal test data. The LoRA weights are updated based on tailored gradients computed from

an approximate loss function $\tilde{L}$. This $\tilde{L}$ is designed to penalize the classifier if its predictions (when used to form the conservative posterior $p^\xi$, as described below) make it impossible to satisfy the overall safety constraint $p^\xi \leq r_t$. The exact formulation of $\tilde{L}$ is detailed in Section 4.1 of the main paper. For efficient training, we schedule $\xi$ linearly, starting from 0 and progressively increasing it to the desired value throughout the training process[27].

### J.3.5 TRAINING SIGNAL GENERATION FOR THE SAFETY CLASSIFICATION MODEL

During the training of the safety classification model, the training signal is derived as follows:

1. **Conservative Posterior Estimation**: For each training batch, the conservative posterior probability $p^\xi(s_{cr,u} = 1 | o_{cr} = o_{cr,pred})$ is estimated. This estimation uses the current safety classification model's predictions ($o_{cr,pred}$) on a set of 512 samples from the training portion of the internal test data. This $p^\xi$ quantifies the estimated risk associated with the classifier's predictions for a given output class $o_{cr,pred}$.

2. **Approximate Loss Calculation**: An approximate loss $\tilde{L}$ is calculated. This loss function is high if the conservative posterior $p^\xi$, corresponding to the "safest" possible classification the model could output for an action (e.g., predicting "harmless"), still exceeds the target safety threshold $r_t$. This signifies that, based on the classifier's current reliability as captured by $p^\xi$, the system would be forced to refuse action to maintain safety. Conversely, the loss is lower if $p^\xi \leq r_t$ is achievable for this "safest" classification. The loss $\tilde{L}$ is thus a function of $p^\xi$, which in turn depends on the classifier's predictions $o_{cr,pred}$ on the 512 samples used for posterior estimation. We also add a term to penalize type 2 error to suppress it, thereby encouraging a broader range of safe actions to be classified into the appropriate class, resulting in the final approximate loss as

$$\bar{J}(u) - \beta \left( \ln \left( \min \left( \frac{r_t}{p^\xi(\mathbf{s}_{cr} = 1 | \mathbf{o}_{cr} = 0)}, 1 \right) \right) - 0.5 p^\xi(\mathbf{s}_{cr} = 0 | \mathbf{o}_{cr} = 1) \right). \quad (164)$$

3. **Gradient Computation and Update**: Gradients of this approximate loss $\tilde{L}$ with respect to the classifier's predictions on the 512 internal test data samples ($\partial \tilde{L} / \partial o_{cr,pred}$) are computed. These gradients indicate how changes in the classifier's predictions would affect the conservative posterior and, consequently, the loss. These gradients are then used to update the LoRA weights of the safety classification model. The aim is to improve its reliability such that its predictions lead to posteriors that accurately reflect true safety and allow the system to meet its safety target $r_t$ by enabling the selection of an implicitly assumed "safe" action.

This process trains the safety classification model by assessing its impact on the feasibility of the downstream safety-constrained optimization, rather than by directly selecting from the 16 candidates of each individual training instance during this phase.

### J.4 HYPERPARAMETERS

Hyperparameters are provided in Table 6.

### J.5 VALIDATION

The validation process was designed to rigorously assess the effectiveness of our framework in enhancing safety while preserving helpfulness, in comparison to established baseline approaches.

### J.5.1 METRICS

Performance was evaluated along two primary dimensions: **Helpfulness**, measured as "Mean Reward" using the 7B parameter reward model ($R_\phi$) from (Dai et al., 2024), and **Safety**, quantified as "Unsafe Responses (%)" using the 7B parameter cost model ($C_\psi$) from (Dai et al., 2024) to determine if a response was harmful. The trade-off is depicted in Figure 4-A of the main paper.

---

[27]Note that $\xi$ is inflated by $\sqrt{2}$ in experiment settings for practical reasons.

Table 4: Hyper-parameters for PPO-Lag.

| Hyper-parameter | Value |
|---|---|
| epochs | 2 |
| max length | 512 |
| temperature | 1.0 |
| top p | 1 |
| num return sequences | 1 |
| repetition penalty | 1.0 |
| prompt batch size | 128 |
| train batch size | 128 |
| actor lr | $1.00 \times 10^{-5}$ |
| actor weight decay | 0.01 |
| actor lr scheduler type | cosine |
| actor lr warmup ratio | 0.03 |
| critic lr | $5.00 \times 10^{-6}$ |
| critic weight decay | 0.0 |
| critic lr scheduler type | cosine |
| critic lr warmup ratio | 0.03 |
| lambda init ($\lambda_0$) | 1.0 |
| lambda lr ($\alpha$) | 0.01 |
| kl coeff ($\beta$) | 0.05 |
| clip range ratio ($\epsilon$) | 0.2 |
| ptx coeff ($\gamma$) | 16.0 |
| bf16 | TRUE |
| tf32 | TRUE |

Table 5: Hyper-parameters of Reward and Cost Model Training for Initialization.

| Hyper-parameters | Reward | Cost |
|---|---|---|
| epochs | 2 | 2 |
| max length | 512 | 512 |
| train batch size | 64 | 64 |
| regularization | 0.001 | 0.001 |
| lr | $2.00 \times 10^{-5}$ | $2.00 \times 10^{-5}$ |
| lr scheduler type | cosine | cosine |
| lr warmup ratio | 0.03 | 0.03 |
| weight decay | 0.1 | 0.1 |
| bf16 | TRUE | TRUE |
| tf32 | TRUE | TRUE |

Table 6: Hyper-parameters for Safety Classification Training. We used different learning rates for each method due to different gradients.

| Hyper-parameters | Ours | RS - LoRA | RS |
|---|---|---|---|
| epochs | 10 | 10 | 10 |
| max length | 512 | 512 | 512 |
| train batch size | 192 | 192 | 192 |
| regularization | 0.000 | 0.001 | 0.001 |
| lr | $2.00 \times 10^{-4}$ | $1.00 \times 10^{-3}$ | $1.00 \times 10^{-6}$ |
| lr scheduler type | warmup | warmup | warmup |
| lr warmup ratio | 0.03 | 0.03 | 0.03 |
| weight decay | 0.1 | 0.1 | 0.1 |
| bf16 | TRUE | TRUE | TRUE |
| tf32 | TRUE | TRUE | TRUE |
| LoRA rank | 1 | 1 | – |
| xi | $\frac{1722270}{n_t}$ | – | – |
| xi scheduler | linear | – | – |

### J.5.2 OUR FRAMEWORK EVALUATION PROTOCOL

At the start of validation of our framework, since we train it with only one threshold and validate it with various thresholds, we introduce a bias (see Section G for theoretical details) by adding or subtracting a specific value to or from the logit output of the classification network, aligning the network output with the desired threshold. The following procedure outlines the abstract process for computing the bias corresponding to each $r_t$.

1. Identify the safe class as the one that minimizes the posterior probability for the unsafe label, and record its corresponding posterior value.

2. If the posterior value is smaller than $r_t$, add an appropriate unit value to the logit of the safe class for all internal test data. Otherwise, subtract an appropriate unit value from the logits.

3. Recalculate the posterior probability. If the updated posterior still deviates significantly from $r_t$, repeat step 2.

4. Save the final bias value.

The final bias value is added to or subtracted from the logit corresponding to the safe class.

The evaluation of our framework proceeded as follows for each prompt in the validation set:

1. The fixed, PPO-Lag trained policy $\pi$ (OPT-1.3B model) generated 16 answer candidates $o_{ncr}$ using diverse beam search.

2. The trained safety classification model (LoRA OPT-350M) predicted their safety classifications $o_{cr}$.

3. **Conservative Testing**: The posterior probability $p^\xi(s_{cr,u} = 1|o_{cr} = o)$ – that a candidate $u$ is unsafe given its classification $o$ – was computed. This utilized the conservative testing procedure outlined in Equations (3), (4), and (5) of the main paper, applied to the validation split of the internal test data. The parameter $\xi$ for conservative testing was set as $\xi = \frac{1722270}{n_t}$, where $n_t$ is the size of the internal test data used during training of the safety classifier.

4. **Optimization for Final Answer Selection**: The final answer $u$ was selected by solving the optimization problem:

$$\min_{u \sim \pi^d} \bar{J}(u) \quad \text{subject to} \quad \log p^\xi(s_{cr,u} = 1|o_{cr} = o) \leq \log r_t \qquad (165)$$

Here, $\pi^d = \{\mathcal{U}(\{o_{ncr,1}, ..., o_{ncr,16}\}), \hat{d}\}$ represents a uniform distribution over the 16 candidates and the default refusal answer $\hat{d}$ ("Sorry, I can't answer this question."). $\bar{J}(\hat{d})$ was assigned a positive value to penalize refusal, while $\bar{J}(u) = 0$ for the 16 candidates, effectively preferring a candidate if it meets the safety constraint. This entire procedure was repeated for various safety thresholds $r_t$ to map out the performance-safety curve.

### J.5.3 BASELINE METHODS FOR COMPARISON

The performance of our framework was compared against:

**PPO-Lag (Directly)**: This involved using the OPT-1.3B policy model $\pi$ directly after its fine-tuning with PPO-Lag, without the subsequent safety classification and optimization layers introduced by our framework. To plot its performance curve, different safety budgets $d$ (as defined in Section J.3.3) were used during its PPO-Lag training/fine-tuning, and the resulting models were evaluated. The final answer was generated by sampling from its output distribution (or by selecting the sequence with the highest probability). This corresponds to "PPOLag" in Figure 4-A of the main paper.

**Rejection Sampling**: This baseline also started with the 16 candidates generated by the PPO-Lag OPT-1.3B policy model $\pi$. Two versions of the safety classification model (OPT-350M), initialized from the same SFT OPT-350M model that was then fully fine-tuned using a cost modeling approach on the SafeRLHF dataset (as described in Section J.3.4 for our framework's classifier initialization), were trained for this baseline. These initialized models were then further trained on the same internal test data using a standard cross-entropy loss: one using LoRA and another with full fine-tuning. For each prompt, the 16 candidates were classified by one of these cross-entropy-trained classifiers. To plot its performance curve, different rejection thresholds were used based on the classifier's logit output for the "harmful" class. Candidates classified as "harmful" according to the current threshold were rejected. A final answer was then selected from the non-rejected candidates (e.g., the one with the highest helpfulness score from the 7B $R_\phi$ model (Dai et al., 2024), or chosen randomly). If all 16 candidates were rejected, the default refusal answer $\hat{d}$ was issued. Figure 4-A of the main paper refers to "Rejection sampling" (likely corresponding to the fully fine-tuned classifier) and "Rejection sampling-LoRA" (corresponding to the LoRA-tuned classifier).

### J.5.4 SAFETY GUARANTEE VERIFICATION

Figure 4-B of the main paper validates the probabilistic safety guarantee of our framework by plotting the actual percentage of unsafe responses against the target safety threshold $r_t$ used in our framework's optimization. This aims to demonstrate that the observed unsafe response rate is indeed at or below the specified $r_t$. In contrast, baseline methods like PPO-Lag (Directly) and Rejection Sampling do not offer such an *a priori* guarantee for a given $r_t$. For these baselines, the safety budget $d$ (for PPO-Lag) or the rejection threshold (for Rejection Sampling) must be tuned empirically during validation to achieve a desired safety level, rather than being able to target a specific $r_t$ directly beforehand. Our framework, however, is designed to satisfy the given $r_t$ through its optimization process.

### J.5.5 SCALING LAW ANALYSIS

The impact of the quantity of internal test data ($n_t$) on the safety-performance trade-off was investigated, with results shown in Figure 4-C of the main paper. The y-axis of this plot represents a combined metric, Reward - dB(Cost), plotted against $n_t$. The parameter $\xi$ for conservative testing was scaled inversely with the number of internal test data points for this analysis, specifically $\xi = \frac{2 \times 10^5}{n_t}$. This experiment aimed to validate the theoretical scaling law presented in Section 5 of the main paper empirically, which suggests that the safety-performance trade-off improves predictably with more internal test data.

# K  COMPUTATIONAL COST ANALYSIS

In this section, we present a comprehensive analysis of our computational costs.

## K.1  REINFORCEMENT LEARNING (SAFETYGYM)

We report the computation time for the complete training and inference process (validation experiment with 10M actions) in Tables 7 and 8. All experiments were conducted using $4\times$RTX 4090 GPUs.

We implement three variants of our approach to demonstrate the flexibility and scalability of our framework:

1. **Generalized**: Our implementation designed for broad applicability across diverse domains, serving as the foundation for our open-source release. This version prioritizes generalizability, ease of adaptation, and debugging capabilities, making it ideal for research and development purposes; however, it exhibits a slower execution speed[28].

2. **Efficient**: A task-specific optimized version that maintains identical functionality while achieving significant speed improvements through targeted code optimizations for this task. This variant provides a fair and direct comparison with (task-specific) baseline methods.

3. **Fast**: Building upon the Efficient version, this variant demonstrates the configurability of our framework. By reducing internal test data size, model complexity, and training epochs, we achieve substantial speedup at the cost of some performance degradation. This version illustrates how users can flexibly balance computational efficiency with task performance based on their specific deployment requirements and constraints. The performance of our Fast version is detailed in Figure 8.

Table 7: PPO Computational Cost Analysis

| | Baseline (PPO) | | Ours | | |
|---|---|---|---|---|---|
| | 10K epochs | 30K epochs | Generalized | Efficient | Fast |
| Training (s) | 48,462 | + 101,389 | + 109,279 | + 47,016 | + 8,907 |
| Inference (s) | 17,988 | | 116,358 | 34,785 | 34,753 |

Table 8: PPO-Lag Computational Cost Analysis

| | Baseline (PPO-Lag) | | Ours | | |
|---|---|---|---|---|---|
| | 10K epochs | 30K epochs | Generalized | Efficient | Fast |
| Training (s) | 45,332 | + 141,463 | + 111,014 | + 47,706 | + 9,142 |
| Inference (s) | 15,312 | | 106,113 | 37,915 | 34,630 |

Note that training times indicate additional time beyond the Baseline with 10K epochs, denoted by the + mark. Tables 7 and 8 demonstrate that our three variants (Generalized, Efficient, and Fast) offer distinct computational trade-offs tailored to different use cases.

Our framework achieves significantly enhanced performance compared to baselines with Generalized/Efficient versions, where the Efficient version requires only approximately 100% additional training time compared to the baseline. Remarkably, our Fast variant still outperforms the baseline with only 20% extra training time beyond the 10K epoch baseline, demonstrating the configurability of our approach.

---

[28]Note that the explanation in this paper follows this implementation

Regarding inference performance, our framework (Efficient version) exhibits approximately twice the inference time of PPO/PPO-Lag baselines due to additional time required for internal test data inference and bias correction (Section 4.3)—accounting for roughly 30% of the overhead—with the remaining difference mainly arising from interpreter-related processing. Notably, the additional time required for internal test data inference and bias correction constitutes a one-time cost that significantly improves model safety, making it a worthwhile investment for industry deployments.

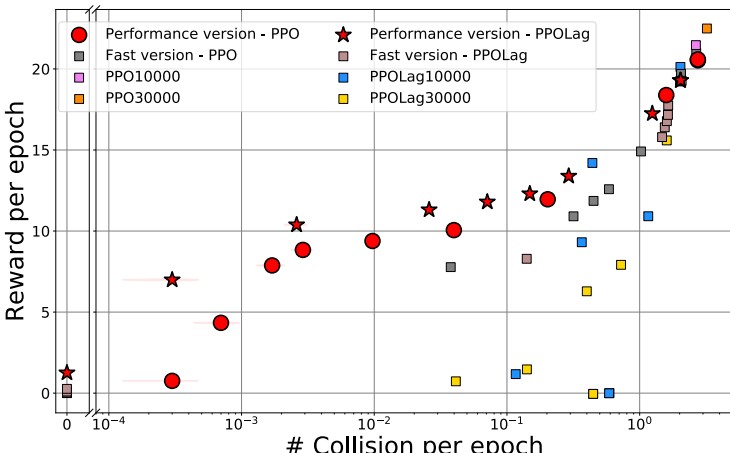

Figure 8: **Performance of Fast version.** Fast version still outperforms baselines, with only 20% extra training time.

## K.2   NATURAL LANGUAGE GENERATION

We report the computational time for the complete training and inference process (evaluated on 800 prompts) in Table 9. Most experiments were conducted using $4\times$RTX 4090 GPUs, while baseline measurements (marked with $\star$) were performed on $8\times$H100 GPUs.

Table 9: Natural Language Generation Computational Cost Analysis

|  | Baseline | Ablation (Rejection Sampling) | | Ours |
| --- | --- | --- | --- | --- |
|  | (OPT-1.3B) | LoRA X | LoRA O | LoRA O |
| Training (s) | 738$\star$ | + 23,089 | + 21,353 | + 23,333 |
| Inference (s) | 100$\star$ | 432 | 426 | 464 (118$\star$)$^{\dagger}$ |

$\star$ Measured on $8\times$H100. All others on $4\times$ RTX 4090.
$^{\dagger}$ Requires an additional 2,432 seconds (on $4\times$RTX 4090).

Most notably, **our method incurs only an 18% overhead in inference time** compared to the baseline fine-tuned OPT-1.3B model (118 seconds vs. 100 seconds). This minimal additional cost is particularly significant given the substantial performance improvements our method achieves.

An important consideration in our computational analysis is the additional time required for internal test data inference and bias correction (Section 4.3). For our configuration, this process *requires an additional* $2,432$ *seconds* on $4\times$ RTX 4090 GPUs.While this represents additional computational overhead, it is a one-time cost that significantly improves model safety, making it a worthwhile investment, especially for industry deployments.

## L    RELATED WORKS

AI safety has become a critical research priority as AI systems are increasingly deployed in high-stakes applications. Although we could not find approaches specifically designed and affirmed to ensure AI safety domain-agnostically, we introduce methods from related domains and discuss general safety frameworks.

**Constrained Reinforcement Learning.**    Safety in reinforcement learning has been extensively studied through constrained optimization approaches. Constrained Policy Optimization (CPO) (Achiam et al., 2017) introduced trust region methods for safe policy learning with probabilistic constraints. Proximal Policy Optimization-based methods have emerged as practical solutions, including PPO-Lag (Ray et al., 2019), which uses Lagrangian methods for constraint satisfaction, and PPO-Barrier (Yang et al., 2023), which employs neural barrier certificates. Other approaches include safe exploration methods (García & Fernández, 2015), temporal-logic shielding (Alshiekh et al., 2018), and reward shaping techniques for safety (Leike et al., 2017). While these methods can be adapted across domains, they lack unified theoretical frameworks for safety guarantees across diverse AI applications.

**Large Language Model Safety.**    The safety of large language models has become increasingly critical due to their widespread deployment (Wei et al., 2022). Constitutional AI (Bai et al., 2022) introduces self-critique mechanisms for harmless responses. Reinforcement Learning from Human Feedback (RLHF) (Ouyang et al., 2022) has become a standard approach for aligning language models with human preferences. Recent work includes red teaming approaches (Ganguli et al., 2022), instruction tuning for helpfulness (Li et al., 2024) and harmlessness (Ji et al., 2023), and adversarial training methods (Zou et al., 2023). Domain-specific safety measures include content filtering (Gehman et al., 2020) and prompt engineering for safety (Liu et al., 2024b). These approaches often require substantial domain-specific customization and engineering.

**General Safety Approaches.**    Rejection sampling (von Neumann, 1951) represents one of the few domain-general safety techniques, applied in both RL (Srinivasan et al., 2020) and language model contexts (Nakano et al., 2021). However, rejection sampling suffers from performance issues. Other general approaches include uncertainty quantification methods (Guo et al., 2017) and robustness techniques (Madry et al., 2018), but these focus on specific aspects of safety rather than providing comprehensive frameworks with mathematical guarantees.

**Scaling Laws and Safety.**    While scaling laws have been extensively studied for model performance (Kaplan et al., 2020; Hoffmann et al., 2022), theoretical relationships between data quantity and safety guarantees remain largely unexplored. Our work addresses this gap by establishing the first scaling law relating internal test data quantity to safety-performance trade-offs.

Our framework generalizes and improves upon these approaches by providing a unified mathematical foundation with provable safety guarantees. For example, our framework integrates with PPO-Lag (Ray et al., 2019) in our experiments. Unlike existing approaches that may require extensive domain-specific engineering, our method achieves safety through constrained optimization with chance constraints, providing theoretical guarantees while maintaining practical applicability across arbitrary AI models and domains.

## M    USE OF LARGE LANGUAGE MODELS

Large language models were used as writing assistance tools for grammar correction, sentence structure improvement, and style refinement throughout the manuscript. LLMs were also employed to assist in identifying and organizing relevant literature during the initial stages of the related work review. All factual content, research contributions, methodology, and scientific claims remain the original work of the authors.

