# OpenReview forum: "Domain-Agnostic Scalable AI Safety Ensuring Framework"
_ICLR.cc/2026/Conference — ICLR 2026 Conference Withdrawn Submission_

### Official Review · Reviewer_VLjm · 2025-10-31

**Soundness:** 3
**Presentation:** 3
**Contribution:** 3
**Rating:** 4
**Confidence:** 3

**Summary:**

This paper proposes a domain-agnostic framework to ensure AI safety. The core idea is to decouple the primary policy model from a safety-verification layer. This layer formulates safety as a chance-constrained optimization problem. Given any AI model's output, a separate safety classification model uses internal test data (i.e., ground-truth safety-labeled data) to estimate the probability of a constraint violation. An optimization component then selects the final action that maximizes performance while provably satisfying a user-specified safety threshold.

contributions:
1. A conservative testing procedure to address the statistical validity concerns of using the same internal test data for training and evaluation.
2. This paper proposes $\zeta$-informative to link data coverage to the conservativeness of the safety bound.
3. A continuous approximate loss function and tailored gradients that make the non-differentiable optimization framework end-to-end trainable.
4. A bias correction technique that allows adjusting the safety threshold at deployment time without retraining.
5. Theoretically, the paper establishes the first scaling law in AI safety, relating the quantity of internal test data to the safety-performance trade-off.

**Strengths:**

1. This paper presents a domain-agnostic framework for AI safety. This is a step beyond domain-specific solutions (e.g., just for RL or just for NLG) and provides a unified approach to safety, which is critical for real-world deployment.
2. The framework is built on a solid theoretical foundation.
3. This paper establishes the first scaling law specifically for AI safety, which links the quantity of test data to the safety-performance trade-off.
4. The authors bridge the gap between theory and practice. They solve the non-differentiability of the constrained optimization step by introducing a continuous approximate loss function and tailored gradients.
5. The ability to adjust safety thresholds at deployment time without retraining is a practical benefit.
6. The experiments are thorough and highly convincing. The framework is tested in three completely different domains and shows promising results.

**Weaknesses:**

1. The framework's guarantees and performance are tied to the quality and quantity of the internal test data. While the scaling law addresses quantity, the practical barrier in many real-world domains will be the high cost of collecting sufficient ground-truth, safety-labeled data. The 10M data points used in the SafetyGym experiment would be intractable for a physical robot.
2. The authors briefly note that adversarial attacks on data unseen in the internal test set could pose a threat. The framework's guarantees rely on the safety classifier's estimates, which are themselves based on the internal data. The robustness of this classifier to out-of-distribution inputs (which might fool it and lead to an incorrect posterior probability) could be explored more deeply.
3. The core theoretical development, including the chance-constraint evaluation and conservative testing, relies on the assumption that the constraint-related environment state ($s_{cr}$) and the classifier's output ($o_{cr}$) are discrete. However, in the SafetyGym experiment, the environment state (e.g., agent position, velocity, LiDAR readings) is continuous. This creates a potential mismatch. It's unclear how the continuous state space is discretized to fit the theory or if this simplification sacrifices the provable guarantees in practical, continuous-state problems.
4. The paper claims domain-agnostic scalability, but the experiments are limited to tasks with relatively structured states or low-dimensional continuous states. It is not demonstrated how the framework, particularly the safety classification model and the $\zeta$-informative data measure, would function in high-dimensional, unstructured scenarios, such as purely vision-based tasks (e.g., autonomous driving from camera feeds) where the environment state is an image.

**Questions:**

1. How does the framework trade-off the quality versus the quantity of the internal test data? In a data-scarce, real-world setting, would 1e3 highly diverse and high-quality labeled samples be more effective than 1e5 repetitive and low-quality samples?
2. In the NLG and production planning experiments, how was the ground-truth internal test data practically collected? The paper mentions using a 7B cost model as a proxy for ground truth in the NLG task. How would the framework's guarantees be affected if this ground-truth data itself contains label noise or bias? Does the conservative testing procedure inherently account for label noise in $s_{cr}$?
3. The framework assumes `the existence of a safety-guaranteed default action`. How would the framework be adapted to safety-critical domains where no such single and universally-safe action exists, and any action carries some non-zero risk?
4. Could the authors explicitly address the assumption of discrete $s_{cr}$ and $o_{cr}$? How was this theoretical requirement practically implemented in the continuous-state SafetyGym environment? Does discretization step affect the theoretical safety guarantees?
5. Given the claim of a general, scalable framework, could the authors provide a more detailed discussion on its limitations and practical application boundaries?
   - What would be the roadmap for applying this framework to a complex, high-dimensional problem like vision-based autonomous driving? What are the key practical or theoretical limitations?
   - Could the authors provide more experimental discussion on scalability in terms of problem size? For example, what happens to the framework's performance and computational cost as the dimensionality of the state, the complexity of the policy model, or the number of simultaneous safety constraints increases significantly?

---

### Official Review · Reviewer_Rae2 · 2025-10-31

**Soundness:** 2
**Presentation:** 2
**Contribution:** 1
**Rating:** 4
**Confidence:** 2

**Summary:**

The paper proposes a domain-agnostic safety framework that wraps an arbitrary AI model with: (i) a predict-and-optimize stage that selects the final action under chance constraints, (ii) a safety classification model that predicts constraint-relevant state, (iii) internal test data used to compute posteriors and later to train the classifier, (iv) a conservative testing procedure that upper-bounds risk via ξ-balls, calibrated by a dataset-coverage measure called ζ-informative, (v) a differentiable approximate loss with tailored gradients for end-to-end training, and (vi) a deployment-time bias correction that shifts logits to hit user thresholds without retraining. The paper also proves a scaling law relating required internal-data size to an error bound and shows results on SafetyGym RL, NLG safety filtering, and a production-planning task

**Strengths:**

1. Proposes a domain-agnostic safety framework that treats safety as chance constraints, combining (i) a safety classification model, (ii) internal test data, and (iii) conservative testing. Introduces the $\zeta$-informative metric to quantify dataset coverage/quality and a bias-correction trick for deployment without retraining.

2. Provides likelihood bounds for conservative testing, a continuous approximate loss enabling back-prop through discrete/argmax operations, and a scaling law that links error bounds to the amount/dimension of internal test data.

3. Evaluates across RL (SafetyGym), NLG, and a production-planning task, suggesting cross-domain applicability.

**Weaknesses:**

1. The method assumes: a sufficiently accurate safety classifier; known/credible priors $p(s_{cr}=\bar s_i)$; estimable conditionals $p(o_{cr}=\bar o_j\mid s_{cr}=\bar s_i)$; Lipschitz behavior of the AI model; and the availability of a default conservative action. Please analyze sensitivity to each assumption.
2. It is not clear why internal test data are informative for *current* decisions when the online state may be OOD. Provide diagnostics on coverage (e.g., nearest-neighbor distances, density estimates) and report performance when the online state is far from the internal-test manifold. How is risk estimated in that regime?
3. After measuring dataset quality with $\zeta$, how is this used to prevent overfitting or to drive optimization choices?
4. If Scaling-law holds, increasing internal test data should systematically reduce violations at comparable reward. Current figures do not clearly demonstrate approaching near-zero cost while having high reward performance.

**Questions:**

1.Please state the paper’s contribution in one sentence separating.

2.Please formalize the “sampling-like function” $\mathrm{Samp}(s)$: domain, codomain, and how it is used in posterior construction or conservative bounds.

3.How does the method behave when online states are far from the internal-test manifold? Any fallback beyond the default action? Can $\zeta$-informative be computed online to trigger safer modes?

---

### Official Review · Reviewer_ygAN · 2025-11-05

**Soundness:** 3
**Presentation:** 2
**Contribution:** 2
**Rating:** 4
**Confidence:** 3

**Summary:**

The paper proposes a unified safety framework spanning LLM moderation/fine-tuning and safe RL/control. Core pieces include a safety classifier (with Bayesian multiclass training), prompt-keyword controls for LLMs, and constraint handling for RL via chance constraints, Big-M formulations, and penalties. Theoretical sections sketch sample requirements for the classifier and chance-constraint guarantees; experiments report LLM filtering/fine-tuning results and several robot/RL tasks (e.g., PPO-style training with collision limits).

**Strengths:**

* **Ambitious scope & unification attempt**: Tries to present one safety recipe across LLM moderation/fine-tuning and safe control/RL.
* **End-to-end scope**: From data/labels to deployment-time gating, with both LLM and RL case studies.
* **Practical orientation**: Emphasizes pipeline decisions (classification, thresholds, deployment checks) that practitioners care about.

I do want to commend the authors for the amount of work put into this work. The contents in the appendix (theory and experiments) is substantial; they could easily fill more than half a PhD dissertation. The supplementary experiments also contain code and video snippets depicting the actual results.

However, there are weaknesses preventing me from giving an unequivocal yes. Please see the Weaknesses section.

**Weaknesses:**

## Positioning and Scope

1. **Over-breadth dilutes key message**: The framework spans LLM alignment, classifier design, statistical validation, and safe RL. Each area has mature baselines and evaluation norms; covering all at once makes it hard to establish depth, novelty, and competitive performance in any single track.

2. **Overly unified problem setup**: The problem formulation makes certain assumptions (e.g. discrete safety taxonomies, zero-thresholded safety constraint) and they may not transfer across domains cleanly without domain expert involvement.

## Theory

3. **Sampling assumptions**: Theory seems to implicitly assume i.i.d. samples. For RL problems, each transition is temporally correlated, and safety interventions change the distribution. Implications for non-i.i.d. rollouts should be discussed.

## Methodological Clarity

4. **Safety taxonomy and label source**: The paper seems to assume that $s_{cr}$ can be easily extracted. In RL, the s_{\mathrm{cr}} that deterministically determines safety isn't directly observable. It may not be discrete and its possible range of values is also not known. It's unclear how the label is generated in the "internal test data" in this case. It doesn't seem trivial to construct such $s_{cr}$ in diverse domains.

5. **Closed-form constraint assumption**: The problem formulation also seems to assume a constraint function that can be computed given $s_{cr}$. In safe RL literature, the safety cost function is observed and can be sampled, but it can't be directly computed for any arbitrary state–action pair.

6. **Big-M method in safety classification**: Section 3.4 seems to be describing the MLE in multi-class cross-entropy. If so, then perhaps it can be condensed and space used to explain $q$ and $r$ in Eq. 7. I know the Big-M method in optimization but didn't get the intuition for how it's applied here.

7. **Motivation of $\zeta$-informative estimation and conservative testing**: The paper should explain why these components are needed and why the usual hold-out dataset method is insufficient to combat overfitting in this context. The said methods introduce additional hyperparameters ($\zeta$ and $\xi$) and it may be challenging to tune these additional knobs.

8. **Joint training of classifier and optimization component**: I guess the L loss function in Eq. 11 refers to the multi-class classification loss for the safety classifier? If so, the paper could explain why the safety classifier is trained jointly with the optimization component (which produces the final action).

## Empirical Evaluation

9. **Unclear discretization of action spaces**: Line 375 mentions continuous action space is discretized. I'm unsure why it needs to be discretized since the earlier section only indicates $s_{cr}$ is discrete.

10. **Claiming previously unattainable safe RL result**: Line 406 claims that the proposed method achieves "safety level unattainable by previous AI models." In safe-RL benchmark studies, a soft budget is usually used and does not aim for absolutely zero safety cost. If the paper is targeting a zero-violation setting, hard-constraint RL methods should be the baseline, instead of PPO-Lagrangian.

11. **Per-step vs per-episode safety**: Related to item 10, Eq. 2 indicates that the paper imposes per-step safety probabilistically. However, the evaluation metric (number of collisions per epoch) seems to be per-episode.

12. **Reporting statistical confidence**: Standard errors (or standard deviations) are not reported in the figures located in the main text.

13. **Unclear Fig. 2**: I'm not sure what the CE method in the Fig. 2 legend refers to; it seems not described in the main text. Also, what do the two circles represent? And what does $R^2$ signify in this chart? It's counterintuitive that higher $R^2$ would result in lower performance in reward vs safety.

14. **Missing ablation study**: Lines 410–415 describe ablation studies and mention "silver, gold points," but I don't seem to find it anywhere in the main text or Fig. 2.

## Other comments

15. Footnote 16 is quoted in the main text, but the actual footnote is not found anywhere.

16. This paper has lots of theoretical derivations and proofs that have to be in the appendix. The main paper only gives a high-level overview and mainly points to the appendix. I wonder if a conference is a suitable place for this paper. Perhaps the authors may need to strategize which main contributions to move back to the main paper so that reviewers can better validate them.

**Questions:**

(related to the Weaknesses above)

1. Is $s_{cr}$ observed, derived, or latent? How is it discretized and labeled, and does it need to have a direct relation to safety cost or the constraint?

2. What's the motivation behind $\zeta$-informative estimation and conservative testing versus a standard held-out set? Has it been applied elsewhere for statistical validation / overfitting prevention?

---

### Note · Authors · 2025-11-26

I have read and agree with the venue's withdrawal policy on behalf of myself and my co-authors.